# On the Optimal Time Complexities in Decentralized Stochastic Asynchronous Optimization

**Alexander Tyurin**
KAUST,[*] AIRI,[†] Skoltech[‡]

**Peter Richtárik**
KAUST[*]

## Abstract

We consider the decentralized stochastic asynchronous optimization setup, where many workers asynchronously calculate stochastic gradients and asynchronously communicate with each other using edges in a multigraph. For both homogeneous and heterogeneous setups, we prove new time complexity lower bounds under the assumption that computation and communication speeds are bounded. We develop a new nearly optimal method, Fragile SGD, and a new optimal method, Amelie SGD, that converge under arbitrary heterogeneous computation and communication speeds and match our lower bounds (up to a logarithmic factor in the homogeneous setting). Our time complexities are new, nearly optimal, and provably improve all previous asynchronous/synchronous stochastic methods in the decentralized setup.

## 1   Introduction

We consider the smooth nonconvex optimization problem

$$\min_{x \in \mathbb{R}^d} \left\{ f(x) := \mathbb{E}_{\xi \sim \mathcal{D}_\xi} \left[ f(x; \xi) \right] \right\}, \tag{1}$$

where $f : \mathbb{R}^d \times \mathbb{S}_\xi \to \mathbb{R}$, and $\mathcal{D}_\xi$ is a distribution on a non-empty set $\mathbb{S}_\xi$. For a given $\varepsilon > 0$, we want to find a possibly random point $\bar{x}$, called an $\varepsilon$–stationary point, such that $\mathbb{E}[\|\nabla f(\bar{x})\|^2] \leq \varepsilon$. We analyze the heterogeneous setup and the convex setup with smooth and non-smooth functions in Sections C and D.

### 1.1   Decentralized setup with times

We investigate the following decentralized asynchronous setup. Assume that we have $n$ workers/nodes with the associated computation times $\{h_i\}$, and communications times $\{\rho_{i \to j}\}$. It takes less or equal to $h_i \in [0, \infty]$ seconds to compute a stochastic gradient by the $i^{\text{th}}$ node, and less or equal $\rho_{i \to j} \in [0, \infty]$ seconds to send *directly* a vector $v \in \mathbb{R}^d$ from the $i^{\text{th}}$ node to the $j^{\text{th}}$ node (it is possible that $h_i = \infty$ and $\rho_{i \to j} = \infty$). All computations and communications can be done asynchronously and in parallel. We would like to emphasize that $h_i \in [0, \infty]$ and $\rho_{i \to j} \in [0, \infty]$ are only upper bounds, and the real and effective computation and communication times can be arbitrarily heterogeneous and random. For simplicity of presentation, we assume the upper bounds are static; however, in Section 5.5, we explain that our result can be trivially extended to the case when the upper bounds are dynamic.

We consider any *weighted directed multigraph* parameterized by a vector $h \in \mathbb{R}^n$ such that $h_i \in [0, \infty]$, and a matrix of distances $\{\rho_{i \to j}\}_{i,j} \in \mathbb{R}^{n \times n}$ such that $\rho_{i \to j} \in [0, \infty]$ for all $i, j \in [n]$ and

---
[*] King Abdullah University of Science and Technology, Thuwal, Saudi Arabia

[†] AIRI, Moscow, Russia

[‡] Skolkovo Institute of Science and Technology, Moscow, Russia

38th Conference on Neural Information Processing Systems (NeurIPS 2024).

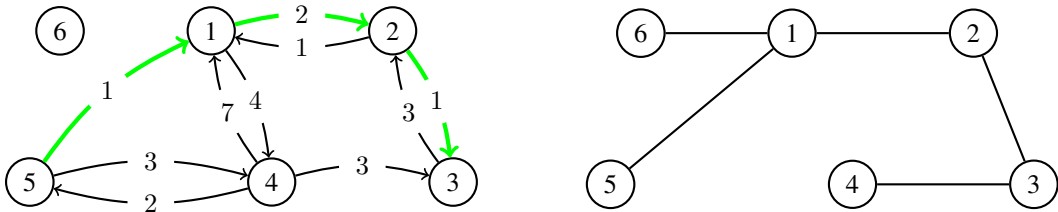

Figure 1: *On the left:* an example of a multigraph with $n = 6$. The edges with $\rho_{i \to j} = \infty$ are omitted. The shortest distance between nodes 5 and 3 is $\tau_{5 \to 3} = \rho_{5 \to 1} + \rho_{1 \to 2} + \rho_{2 \to 3}$. Note that $\rho_{5 \to 3} = \infty$. *On the right:* an example of a spanning tree that illustrates the shortest paths from every node to node 3. The shortest distance between nodes 6 and 3 is $\tau_{6 \to 3} = \infty$ because $\rho_{6 \to i} = \infty$ for all $i \neq 6$.

$\rho_{i \to i} = 0$ for all $i \in [n]$. Every worker $i$ is connected to any other worker $j$ with two edges $i \to j$ and $j \to i$. For this setup, it would be convenient to define *the distance of the shortest path from worker $i$ to worker $j$* :

$$\tau_{i \to j} := \min_{\text{path} \in P_{i \to j}} \sum_{(u,v) \in \text{path}} \rho_{u \to v} \in [0, \infty], \tag{2}$$

where $P_{i \to j} := \big\{ [(k_1, k_2), \ldots, (k_m, k_{m+1})] \,\big|\, \forall m \in \mathbb{N} \, \forall p \in [m+1] \, \forall k_p \in [n],$
$$k_1 = i, k_{m+1} = j, \forall j \in \{2, \ldots, m\} \, k_{j-1} \neq k_j \neq k_{j+1} \big\}$$

is the set of all possible paths without loops from worker $i$ to worker $j$ for all $i, j \in [n]$. One can easily show that the triangle inequality $\tau_{i \to j} \leq \tau_{i \to k} + \tau_{k \to j}$ holds for all $i, j, k \in [n]$. Note that $\tau_{i \to j} \leq \rho_{i \to j}$ for all $i, j \in [n]$. It is important to distinguish $\tau_{i \to j}$ and $\rho_{i \to j}$ because it is possible that $\tau_{i \to j} < \rho_{i \to j} = \infty$ if workers $i$ and $j$ are connected by an edge $\rho_{i \to j} = \infty$, and there is a path through other workers (see Fig. 1).

We work with the following standard assumption from smooth nonconvex stochastic optimization literature.

**Assumption 1.** *$f$ is differentiable and $L$–smooth, i.e., $\|\nabla f(x) - \nabla f(y)\| \leq L \|x - y\| \, \forall x, y \in \mathbb{R}^d$.*

**Assumption 2.** *There exist $f^* \in \mathbb{R}$ such that $f(x) \geq f^*$ for all $x \in \mathbb{R}^d$.*

**Assumption 3.** *For all $x \in \mathbb{R}^d$, stochastic gradients $\nabla f(x; \xi)$ are unbiased and $\sigma^2$-variance-bounded, i.e., $\mathbb{E}_\xi[\nabla f(x; \xi)] = \nabla f(x)$ and $\mathbb{E}_\xi[\|\nabla f(x; \xi) - \nabla f(x)\|^2] \leq \sigma^2$, where $\sigma^2 \geq 0$. We also assume that computation and communication times are statistically independent of stochastic gradients.*

## 2 Previous Results

### 2.1 Time complexity with one worker

For the case when $n = 1$ and $\tau_{1 \to 1} = 0$, convergence rates and time complexities of problem (1) are well-understood. It is well-known that the stochastic gradient method (SGD), i.e., $x^{k+1} = x^k - \gamma \nabla f(x^k; \xi^k)$, where $\{\xi^k\}$ are i.i.d. from $\mathcal{D}_\xi$, has the optimal *oracle complexity* $\Theta(L\Delta/\varepsilon + \sigma^2 L\Delta/\varepsilon^2)$ (Ghadimi and Lan, 2013; Arjevani et al., 2022). Assuming that the computation time of one stochastic gradient is bounded by $h_1$, we can conclude that the optimal time complexity is

$$T_{\text{single}}^{\tau=0} := \Theta \left( h_1 \times \left( \frac{L\Delta}{\varepsilon} + \frac{\sigma^2 L\Delta}{\varepsilon^2} \right) \right) \tag{3}$$

seconds in the worst case.

### 2.2 Parallel optimization without communication costs

Assume that $n > 1$ and $\tau_{i \to j} = 0$ for all $i, j \in [n]$, and computation times of stochastic stochastic gradients are arbitrarily heterogeneous. The simplest baseline method in this setup is Minibatch SGD,

Table 1: **Homogeneous Case** (1). The time complexities to get an $\varepsilon$-stationary point in the nonconvex setting. We assume that $\tau_{i \to j} = \tau_{j \to i}$ for all $i, j \in [n]$ in this table. Abbr.: $\sigma^2$ is defined as $\mathbb{E}_\xi[\|\nabla f(x; \xi) - \nabla f(x)\|^2] \leq \sigma^2$ for all $x \in \mathbb{R}^d$, $L$ is a smoothness constant of $f$, $\Delta := f(x^0) - f^*$.

| Method | The Worst-Case Time Complexity Guarantees | Comment |
|---|---|---|
| Minibatch SGD | $\max\left\{\max\limits_{i,j \in [n]} \tau_{i \to j}, \max\limits_{i \in [n]} h_i\right\}\left(\frac{L\Delta}{\varepsilon} + \frac{\sigma^2 L\Delta}{n\varepsilon^2}\right)$ | Suboptimal since, for instance, it "linearly"[(c)] depends on $\max\limits_{i \in [n]} h_i$ |
| SWIFT (Bornstein et al., 2023) | —[(b)] | Suboptimal since, for instance, it "linearly"[(c)] depends on $\max\limits_{i \in [n]} h_i$ |
| Asynchronous SGD (Even et al., 2024) | —[(b)] | Suboptimal, for instance, even if $\tau_{i \to j} = 0\, \forall i, j \in [n]$ |
| Fragile SGD (Corollary 1) | $\frac{L\Delta}{\varepsilon} \min\limits_{j \in [n]} t^*(\sigma^2/\varepsilon, [h_i]_{i=1}^n, [\tau_{i \to j}]_{i=1}^n)^{(a)}$ | Optimal up to $\log n$ factor |
| Lower Bound (Theorem 1) | $\frac{1}{\log n + 1} \frac{L\Delta}{\varepsilon} \min\limits_{j \in [n]} t^*(\sigma^2/\varepsilon, [h_i]_{i=1}^n, [\tau_{i \to j}]_{i=1}^n)^{(a)}$ | — |

[(a)] The mapping $t^*$ is defined in Definition 2.
[(b)] It is not trivial to infer the time complexities for these methods. However, in Section 5.4, we discuss some cases where it is transparent that the obtained results are suboptimal.
[(c)] Meaning that the corresponding time complexity $\to \infty$ if $\max_{i \in [n]} h_i \to \infty$.

i.e.,

$$x^{k+1} = x^k - \frac{\gamma}{n} \sum_{i=1}^n \nabla f(x^k; \xi_i^k), \tag{4}$$

where $\{\xi_i^k\}$ are i.i.d. from $\mathcal{D}_\xi$ and the gradient $\nabla f(x^k; \xi_i^k)$ is calculated in worker $i$ in parallel. This method waits for stochastic gradients from all workers; thus, it is not robust to "stragglers" and in the worst case the time complexity of such an algorithm is

$$T_{\text{mini}}^{\tau=0} := \Theta\left(\max_{i \in [n]} h_i \times \left(\frac{L\Delta}{\varepsilon} + \frac{\sigma^2 L\Delta}{n\varepsilon^2}\right)\right),$$

which depends on the time $\max_{i \in [n]} h_i$ of the slowest worker. There are many other more advanced methods including Picky SGD (Cohen et al., 2021), Asynchronous SGD (e.g., (Recht et al., 2011; Nguyen et al., 2018; Mishchenko et al., 2022; Koloskova et al., 2022)), and Rennala SGD (Tyurin and Richtárik, 2023) that are designed to be robust to workers' chaotic computation times. Under the assumption that the computation times of the workers are heterogeneous and bounded by $\{h_i\}$, Tyurin and Richtárik (2023) showed that Rennala SGD is the first method that achieves the optimal time complexity

$$T_{\text{Rennala}}^{\tau=0} := \Theta\left(\min_{m \in [n]} \left[\left(\frac{1}{m} \sum_{i=1}^m \frac{1}{h_{\pi_i}}\right)^{-1} \left(\frac{L\Delta}{\varepsilon} + \frac{\sigma^2 L\Delta}{m\varepsilon^2}\right)\right]\right), \tag{5}$$

where $\pi$ is a permutation that sorts $h_i : h_{\pi_1} \leq \cdots \leq h_{\pi_n}$. For instance, one can see that $T_{\text{Rennala}}^{\tau=0} \leq T_{\text{mini}}^{\tau=0}$ for all parameters.

## 2.3 Parallel optimization with communication costs $\tau_{i \to j}$

We now consider the setup where workers' communication times can not be ignored. This problem leads to a research field called *decentralized optimization*. This setup is the primary case for us. All $n$ workers calculate stochastic gradients in parallel and communicate with each other. Numerous works consider this setup, and we refer to Yang et al. (2019); Koloskova (2024) for detailed surveys. Typically, methods in this setting use the gossip matrix framework (Duchi et al., 2011; Shi et al., 2015; Koloskova et al., 2021) and get an *iteration converge rate* that depends on the spectral gap of a mixing matrix. However, such rates do not give the physical time of algorithms (see also Section B).

Let us consider a straightforward baseline: Minibatch SGD. We can implement (4) in a way that all workers calculate one stochastic gradient (takes at most $\max_{i \in [n]} h_i$ seconds) and then aggregate them to one *pivot worker* $j^*$ (takes at most $\max_{i \in [n]} \tau_{i \to j^*}$ seconds). Then, pivot worker $j^*$ calculates

Table 2: **Heterogeneous Case** (17). Time complexities to get an $\varepsilon$-stationary point in the nonconvex setting. Abbr.: $\sigma^2$ is defined as $\mathbb{E}_\xi[\|\nabla f_i(x;\xi) - \nabla f_i(x)\|^2] \leq \sigma^2$ for all $x \in \mathbb{R}^d, i \in [n]$, $L$ is a smoothness constant of $f = \frac{1}{n}\sum_{i=1}^n f_i$, $\Delta := f(x^0) - f^*$.

| Method | The Worst-Case Time Complexity Guarantees | Comment |
|---|---|---|
| Minibatch SGD | $\frac{L\Delta}{\varepsilon}\max\left\{\left(1 + \frac{\sigma^2}{n\varepsilon}\right)\max\{\max\limits_{i,j\in[n]}\tau_{i\to j}, \max\limits_{i\in[n]} h_i\}\right\}$ | suboptimal if $\sigma^2/\varepsilon$ is large |
| RelaySGD, Gradient Tracking (Vogels et al., 2021) (Liu et al., 2024) | $\geq \frac{\max\limits_{i\in[n]} L_i\Delta}{\varepsilon}\frac{\sigma^2}{n\varepsilon}\max\limits_{i\in[n]} h_i$ | requires local $L_i$-smooth. of $f_i$, suboptimal if $\sigma^2/\varepsilon$ is large (even if $\max_{i\in[n]} L_i = L$) |
| Asynchronous SGD (Even et al., 2024) | — | requires similarity of the functions $\{f_i\}$, requires local $L_i$-smooth. of $f_i$ |
| Amelie SGD and Lower Bound (Thm. 7 and Cor. 2) | $\frac{L\Delta}{\varepsilon}\max\left\{\max\limits_{i,j\in[n]}\tau_{i\to j}, \max\limits_{i\in[n]} h_i, \frac{\sigma^2}{n\varepsilon}\left(\frac{1}{n}\sum\limits_{i=1}^n h_i\right)\right\}$ | Optimal up to a constant factor |

a new point $x^{k+1}$ and broadcasts it to all workers (takes $\max_{i\in[n]}\tau_{j^*\to i}$ seconds). One can easily see that the time complexity of such a procedure is[4]

$$T_{\text{mini}} := \Theta\left(\max\left\{\max_{i,j\in[n]}\tau_{i\to j}, \max_{i\in[n]} h_i\right\}\left(\frac{L\Delta}{\varepsilon} + \frac{\sigma^2 L\Delta}{n\varepsilon^2}\right)\right). \tag{6}$$

We can analyze any other asynchronous decentralized method, which will be done with more advanced methods in Section 5.4.

But what is the best possible (optimal) time complexity we can get in the setting from Section 1.1?

Unlike the setups from Sections 2.1 and 2.2 when the communication times are zero ($\tau_{i\to j} = 0$ for all $i, j \in [n]$), the optimal time complexity and an optimal method for the case $\tau_{i\to j} \geq 0$ for all $i, j \in [n]$ are not known. Our main goal in this paper is to solve this problem.

# 3   Contributions

We consider the class of functions that satisfy the setup and the assumptions from Section 1.1 and show that (informally) it is impossible to develop a method that will converge faster than (7) seconds. Next, we develop a new asynchronous stochastic method, Fragile SGD, that is *nearly* optimal (i.e., almost matches this lower bound; see Table 1 and Corollary 1). This is the first such method. It provably improves on Asynchronous SGD (Even et al., 2024) and all other synchronous and asynchronous methods (Bornstein et al., 2023). We also consider the heterogeneous setup (see Table 2 and Section C), where we discover the optimal time complexity by proving another lower bound and developing a new optimal method, Amelie SGD, with weak assumptions. The developed methods can guarantee the *iteration complexity* $\mathcal{O}(L\Delta/\varepsilon)$ with arbitrarily heterogeneous random computation and communication times (Theorems 4 and 8). Our findings are extended to the convex setup in Section D, where we developed new accelerated methods, Accelerated Fragile SGD and Accelerated Amelie SGD.

# 4   Lower Bound

In order to construct our lower bound, we consider any (zero-respecting) method that can be represented by Protocol 1. This protocol captures all virtually distributed synchronous and asynchronous methods, such as Minibatch SGD, SWIFT (Bornstein et al., 2023), Asynchronous SGD (Even et al., 2024), and Gradient Tracking (Koloskova et al., 2021).

For all such methods we prove the following theorem.

**Theorem 1** (Lower Bound; Simplified Presentation of Theorem 19). *Consider Protocol 1 with* $\nabla f(\cdot;\cdot)$. *We take any* $h_i \geq 0$ *and* $\tau_{i\to j} \geq 0$ *for all* $i, j \in [n]$ *such that* $\tau_{i\to j} \leq \tau_{i\to k} + \tau_{k\to j}$ *for all* $i, k, j \in [n]$. *We fix* $L, \Delta, \varepsilon, \sigma^2 > 0$ *that satisfy the inequality* $\varepsilon < cL\Delta$ *for some universal constant*

---

[4]because $\max_{i,j\in[n]}\tau_{i\to j} \leq \max_{i\in[n]}\tau_{i\to j^*} + \max_{i\in[n]}\tau_{j^*\to i} \leq 2\max_{i,j\in[n]}\tau_{i\to j}$ by the triangle inequality

*c. For any (zero-respecting) algorithm, there exists a function $f$, which satisfy Assumptions 1, 2 and $f(0) - f^* \leq \Delta$, and a stochastic gradient mapping $\nabla f(\cdot; \cdot)$, which satisfies Assumption 3, such that the required time to find $\varepsilon-$solution is*

$$\Omega\left(\frac{1}{\log n+1}\frac{L\Delta}{\varepsilon}\min_{j\in[n]} t^*(\sigma^2/\varepsilon, [h_i]_{i=1}^n, [\tau_{i\to j}]_{i=1}^n)\right) \tag{7}$$

$$\stackrel{\text{Def } 2}{\equiv} \Omega\left(\frac{1}{\log n+1}\frac{L\Delta}{\varepsilon}\min_{j\in[n]}\min_{k\in[n]}\max\left\{\max\{\tau_{\pi_{j,k}\to j}, h_{\pi_{j,k}}\}, \frac{\sigma^2}{\varepsilon}\left(\sum_{i=1}^k\frac{1}{h_{\pi_{j,i}}}\right)^{-1}\right\}\right), \tag{8}$$

*where, for $j \in [n]$, $\pi_{j,\cdot}$ is a permutation that sorts $\{\max\{\tau_{i\to j}, h_i\}\}_{i=1}^n$, i.e., $\max\{\tau_{\pi_{j,1}\to j}, h_{\pi_{j,1}}\} \leq \cdots \leq \max\{\tau_{\pi_{j,n}\to j}, h_{\pi_{j,n}}\}$.*

---

**Protocol 1** Simplified Presentation of Protocol 8

---

1: Init $S_i = \emptyset$ (all available information) on worker $i$ for all $i \in [n]$
2: Run the following two loops in each worker in parallel
3: **while** True **do**
4:      Calculate a new point $x_i^k$ based on $S_i$                             (takes 0 seconds)
5:      Calculate a stochastic gradient $\nabla f(x_i^k; \xi)$ (or $\nabla f_i(x_i^k; \xi)$)     $\xi \sim \mathcal{D}_\xi$      (takes $h_i$ seconds)
6:      Atomic add $\nabla f(x_i^k; \xi)$ (or $\nabla f_i(x_i^k; \xi)$) to $S_i$         (atomic operation, takes 0 seconds)
7: **end while**
8: **while** True **do**
9:      Send[(a)] any vector from $\mathbb{R}^d$ based on $S_i$ to any worker $j$ and go to the next step of this loop without waiting                (takes $\tau_{i\to j}$ seconds to send; worker $j$ adds this vector to $S_j$)
10: **end while**

(a): When we prove the lower bounds, we allow algorithms to send as many vectors as they want in parallel from worker $i$ to worker $j$ for all $i \neq j \in [n]$.

---

The intuition and meaning of the formula (8) is discussed in Section 5.2. Note that if we take $n = 1$ and $\tau_{1\to1} = 0$ our lower bound reduces to the lower bound (3) up to a log factor. Moreover, if we take $n > 1$ and $\tau_{i\to j} = 0$ for all $i, j \in [n]$, then (8) reduces to (5) up to a log factor. Thus, (8) is nearly consistent with the lower bounds from (Arjevani et al., 2022; Tyurin and Richtárik, 2023). We get an extra $\log n$ factor due to the generality of our setup. The reason is technical, and we explain it in Section E.5. In a nutshell, the lower problem reduces to the analysis of the concentration of the time series $y^T := \min_{j\in[n]} y_j^T$ and $y_j^T := \min_{i\in[n]}\left\{y_i^{T-1} + h_i\eta_i^T + \tau_{i\to j}\right\}$, where $y_i^0 = 0$ for all $i \in [n]$, and $\{\eta_i^k\}$ are i.i.d. geometric random variables. This analysis is not trivial due to the $\min_{i\in[n]}$ operations. Virtually all previous works that analyzed lower bounds did not have such a problem because they analyzed time series with a sum structure (e.g., $\bar{y}^T := \bar{y}^{T-1} + \varrho^T$, where $\{\varrho^k\}$ are some random variables, and $\bar{y}^0 = 0$).

Let us define an auxiliary function to simplify readability.

**Definition 2** (Equilibrium Time). *A mapping $t^* : \mathbb{R}_{\geq0} \times \mathbb{R}_{\geq0}^n \times \mathbb{R}_{\geq0}^n \to \mathbb{R}_{\geq0}$ with inputs $s$ (scalar), $[h_i]_{i=1}^n$ (vector), and $[\bar{\tau}_i]_{i=1}^n$ (vector) is called the equilibrium time if it is defined as follows. Find a permutation[5] $\pi$ that sorts $\max\{\bar{\tau}_i, h_i\}$ as $\max\{\bar{\tau}_{\pi_1}, h_{\pi_1}\} \leq \cdots \leq \max\{\bar{\tau}_{\pi_n}, h_{\pi_n}\}$. Then the mapping returns the value*

$$t^*(s, [h_i]_{i=1}^n, [\bar{\tau}_i]_{i=1}^n) \equiv \min_{k\in[n]}\max\left\{\max\{\bar{\tau}_{\pi_k}, h_{\pi_k}\}, s\left(\sum_{i=1}^k\frac{1}{h_{\pi_i}}\right)^{-1}\right\} \in [0, \infty]. \tag{9}$$

## 5 New Method: Fragile SGD

We introduce a novel optimization method characterized by time complexities that closely align with the lower bounds established in Section 4. Our algorithms leverage *spanning trees*. A spanning tree is a tree (undirected unweighted graph) encompassing all workers. The edges of spanning trees are *virtual* and not related to the edges defined in Section 1.1 (see Fig. 1).

---

[5]It is possible that a permutation is not unique, then the result of the mapping does not depend on the choice.

---

**Algorithm 2** Fragile SGD

---

1: **Input:** starting point $x^0$, stepsize $\gamma$, batch size $S$, pivot worker $j^*$, spanning trees $\overline{st}$ and $\overline{st}_{\text{bc}}$
2: Start Process 0 (Algorithm 3) in worker $j^*$
3: Start Process $i$ (Algorithm 4) in all workers for all $i \in [n]$ (including worker $j^*$)

---

**Algorithm 3** Process 0 (running in worker $j^*$)

---

1: **for** $k = 0, 1, \ldots, K-1$ **do**
2:     Send $x^k$ to Process $j^*$            $\triangleright$ takes $\tau_{j^* \to j^*} = 0$ seconds
3:     Init $(g^k, s^k) = (0, 0)$
4:     **while** $s < S$ **do**
5:         Wait for a message $(g^k_{j^*,\text{send}}, s^k_{j^*,\text{send}})$ from Process $j^*$
6:         $g^k = g^k + g^k_{j^*,\text{send}}; \quad s^k = s^k + s^k_{j^*,\text{send}}$
7:     **end while**
8:     $x^{k+1} = x^k - \frac{\gamma}{s^k} g^k$
9: **end for**

---

**Algorithm 4** Process $i$ (running in worker $i$)

---

1: Init $(g^k_{i,\text{next}}, s^k_{i,\text{next}}) = (0, 0)$ for all $k \in \{0, \ldots, K-1\}^{\text{(a)}}$, $k_{\max} = 0$
2: Run the following three functions in parallel
3: **function** BroadcastFurtherAndCalculateStochasticGradients
4:     **while** True
5:         Get a new point $x^{\bar{k}}$ sent by Process $\text{next}_{\overline{st}_{\text{bc}}, j^*}(i)$      $\triangleright$ $\bar{k}$ is not necessarily equals to the current $k$ from Alg. 3
6:         Atomic update $k_{\max} = \max\{\bar{k}, k_{\max}\}$
7:         **for** all $p$ such that $\text{next}_{\overline{st}_{\text{bc}}, j^*}(p) = i$ **do**      $\triangleright$ broadcasts $x^{\bar{k}}$ further
8:             Send$^{\text{(b)}}$ $x^{\bar{k}}$ to Process $p$ and go to the next step without waiting $\triangleright$ takes at most $\tau_{i \to p}$ seconds to send
9:         **end for**
10:        **while** not received a new point      $\triangleright$ immediately stops the loop when receives a new point
11:           Calculate $\nabla f(x^{\bar{k}}; \xi), \quad \xi \sim \mathcal{D}$      $\triangleright$ takes at most $h_i$ second
12:           Run atomic add $g^{\bar{k}}_{i,\text{next}} = g^{\bar{k}}_{i,\text{next}} + \nabla f(x^{\bar{k}}; \xi), s^{\bar{k}}_{i,\text{next}} = s^{\bar{k}}_{i,\text{next}} + 1$
13:        **end while**
14:     **end while**
15: **end function**
16: **function** ReceiveVectorsFromPreviousWorkers
17:     **while** True
18:         Wait for a message $(g^{\hat{k}}_{p,\text{send}}, s^{\hat{k}}_{p,\text{send}})$ from any Process $p$ such that $\text{next}_{\overline{st}, j^*}(p) = i$
19:         Run atomic add $g^{\hat{k}}_{i,\text{next}} = g^{\hat{k}}_{i,\text{next}} + g^{\hat{k}}_{p,\text{send}}, s^{\hat{k}}_{i,\text{next}} = s^{\hat{k}}_{i,\text{next}} + s^{\hat{k}}_{p,\text{send}}$
20:         Atomic update $k_{\max} = \max\{\hat{k}, k_{\max}\}$
21:     **end while**
22: **end function**
23: **function** SendVectorsToNextWorker
24:     **while** True
25:         Atomic init $g^{k_{\max}}_{i,\text{send}}, s^{k_{\max}}_{i,\text{send}} = g^{k_{\max}}_{i,\text{next}}, s^{k_{\max}}_{i,\text{next}}$ and reset $g^{k_{\max}}_{i,\text{next}} = 0, s^{k_{\max}}_{i,\text{next}} = 0$
26:         Send$^{\text{(b)}}$ $(g^{k_{\max}}_{i,\text{send}}, s^{k_{\max}}_{i,\text{send}})$ to Process $\text{next}_{\overline{st}, j^*}(i)$ and wait    $\triangleright$ takes at most $\tau_{i \to \text{next}_{\overline{st}, j^*}(i)}$ seconds to wait
27:     **end while**
28: **end function**

---

(a): To simplify the listing of the algorithm, we assume here that a worker can store $K$ auxiliary vectors in the memory. One can see that it's not necessary, and it is sufficient to maintain only one vector $g^{k_{\max}}_{i,\text{next}}$. In particular, it is sufficient to modify the logic of Lines 12 and 19, where we run the add operations only if $\bar{k} = k_{\max}$ and $\hat{k} = k_{\max}$. The efficient implementation has $\mathcal{O}(d)$ floats memory complexity per worker.
(b): BroadcastFurtherAndCalculateStochasticGradients and SendVectorsToNextWorker may try to send through the same edge. In this case, we can interleave their communications and decrease the speed of each line by at most two.

**Definition 3** (mapping $\text{next}_{T,j}(i)$)**.** Take a spanning tree $T$ and fix any worker $j \in [n]$. For $i = j$, we define $\text{next}_{T,j}(i) = 0$. For all $i \neq j \in [n]$, we define $\text{next}_{T,j}(i)$ as the index of the next worker on the path of the spanning tree $T$ from worker $i$ to worker $j$.

Our new method, Fragile SGD, is presented in Algorithms 2, 4, and 3. While Fragile SGD seems to be lengthy, the idea is pretty simple. All workers do three jobs in parallel: calculate stochastic gradients, receive vectors, and send vectors through spanning trees. A pivot worker aggregates all stochastic gradients in $g^k$ and, at some moment, does $x^{k+1} = x^k - \gamma g^k$. The algorithms formalize this idea.

Algorithm 2 requires a starting point $x^0$, a stepsize $\gamma$, a batch size $S$, the index $j^*$ of a pivot worker, and spanning trees $\overline{st}$ and $\overline{st}_{\text{bc}}$ for the input. We need two spanning $\overline{st}_{\text{bc}}$ and $\overline{st}$ trees because, in general, the fastest communication of a vector from $j^*$ to $i$ and from $i$ to $j^*$ should be arranged through two different paths. Algorithm 2 starts $n + 1$ processes running in parallel. Note that the pivot worker $j^*$ runs two parallel processes, called Process 0 and Process $j^*$, and any other worker $i$ runs one Process $i$. Process 0 broadcasts a new point $x^k$ through $\overline{st}_{\text{bc}}$ to all other processes and goes to the loop where it waits for messages from Process $j^*$. Process $i$ starts three functions that will be running in parallel: i) the first function's job is to receive a new point, broadcast it further, and start the calculation of stochastic gradients, ii) the second function receives stochastic gradients from all previous processes that are sending vectors to worker $j^*$, iii) the third function sends vectors the next worker on the path to $j^*$. By the definition of $\text{next}_{\overline{st},j^*}(\cdot)$, all calculated stochastic vectors are sent to worker $j^*$, where they are first aggregated in Process $j^*$, and then, since $\text{next}_{\overline{st},j^*}(j^*) = 0$, Process $j^*$ will send $g^k_{j^*,\text{send}}$ to Process 0. This process waits for the moment when the number of stochastic gradients $s^k$ aggregated in $g^k$ is greater or equal to $S$. When it happens, the loop stops, and Process 0 does a gradient-like step. The structure of the algorithm and the idea of spanning trees resemble the ideas from (Vogels et al., 2021; Tyurin and Richtárik, 2023). The main observation is that this algorithm is equivalent to $x^{k+1} = x^k - \frac{\gamma}{s^k} \sum_{i=1}^{s^k} \nabla f(x^k; \xi_i^k)$, where $s^k \geq S$ and $\{\xi_i^k\}$ are i.i.d. samples. Note that all stochastic gradients calculated at points $x^0, \ldots, x^{k-1}$ will be ignored in the $k^{\text{th}}$ iteration of Algorithm 3.

**Theorem 4.** *Let Assumptions 1, 2, and 3 hold. We take $\gamma = 1/2L$, batch size $S = \max\{\lceil \sigma^2/\varepsilon \rceil, 1\}$, any pivot worker $j^* \in [n]$, and any spanning trees $\overline{st}$ and $\overline{st}_{\text{bc}}$ in Algorithm 2. For all $K \geq 16L\Delta/\varepsilon$, we get $\frac{1}{K} \sum_{k=0}^{K-1} \mathbb{E}\left[\left\|\nabla f(x^k)\right\|^2\right] \leq \varepsilon$.*

The proof is simple and uses standard techniques from (Lan, 2020; Khaled and Richtárik, 2022). The result of the theorem holds even if $h_i = \infty$ and $\tau_{i \to j} = \infty$ for all $i, j \in [n]$ because $h_i$ and $\tau_{i \to j}$ are only upper bounds on the real computation and communications speeds. In Algorithm 3, each iteration $k$ can be arbitrarily slow, and still, the result of Theorem 4 holds and the method converges after $O(L\Delta/\varepsilon)$ iterations. The next result gives time complexity guarantees for our algorithm.

**Theorem 5.** *Consider the assumptions and the parameters from Theorem 4. For any pivot worker $j^* \in [n]$ and spanning trees $\overline{st}$ and $\overline{st}_{\text{bc}}$, Algorithm 2 converges after at most*

$$\Theta\left(\frac{L\Delta}{\varepsilon} t^*(\sigma^2/\varepsilon, [h_i]_{i=1}^n, [\mu_{i \to j^*} + \mu_{j^* \to i}]_{i=1}^n)\right) \tag{10}$$

*seconds, where $\mu_{i \to j^*}$ ($\mu_{j^* \to i}$) is an upper bound on the times required to send a vector from worker $i$ to worker $j^*$ (from worker $j^*$ to worker $i$) along the spanning tree $\overline{st}$ (spanning tree $\overline{st}_{\text{bc}}$).*

Note that our method does not need the knowledge of $\{h_i\}$ and $\{\mu_{i \to j}\}$ to guarantee the time complexity rate, and it *automatically* obtains it.

**Corollary 1.** *Consider the assumptions and the parameters from Theorem 5. Let us take a pivot worker $j^* = \arg\min_{j \in [n]} t^*(\sigma^2/\varepsilon, [h_i]_{i=1}^n, [\tau_{i \to j} + \tau_{j \to i}]_{i=1}^n)$, and a spanning tree $\overline{st}$ (spanning tree $\overline{st}_{\text{bc}}$) that connects every worker $i$ to worker $j^*$ (worker $j^*$ to every worker $i$) with the shortest distance $\tau_{i \to j^*}$ ($\tau_{j^* \to i}$). Then Algorithm 2 converges after at most*

$$T_* := \Theta\left(\frac{L\Delta}{\varepsilon} \min_{j \in [n]} t^*(\sigma^2/\varepsilon, [h_i]_{i=1}^n, [\tau_{i \to j} + \tau_{j \to i}]_{i=1}^n)\right) \tag{11}$$

$$\overset{\text{Def 2}}{\equiv} \Theta\left(\frac{L\Delta}{\varepsilon} \min_{j \in [n]} \min_{k \in [n]} \max\left\{\max\{\tau_{\pi_{j,k} \to j} + \tau_{j \to \pi_{j,k}}, h_{\pi_{j,k}}\}, \frac{\sigma^2}{\varepsilon}\left(\sum_{i=1}^k \frac{1}{h_{\pi_{j,i}}}\right)^{-1}\right\}\right) \tag{12}$$

*seconds, where, for all $j \in [n]$, $\pi_{j,\cdot}$ is a permutation that sorts $\{\max\{\tau_{i \to j} + \tau_{j \to i}, h_i\}\}_{i=1}^n$.*

This corollary has better time complexity guarantees than Theorem 5 because, by the definition of $\tau_{i \to j}$, $\tau_{i \to j} \leq \mu_{i \to j}$ for al $i, j \in [n]$. However, it requires the particular choice of a pivot worker and spanning trees.

## 5.1 Discussion

Comparing the lower bound (7) with the upper bound (11), one can see that Fragile SGD has a *nearly optimal time complexity*. If we ignore the $\log n$ factor in (7) and assume that $\tau_{i \to j} = \tau_{j \to i}$ for all $i, j \in [n]$, which is a weak assumption in many applications, then Fragile SGD is optimal.

Unlike most works (Even et al., 2024; Lian et al., 2018; Koloskova et al., 2021) in the decentralized setting, our time complexity guarantees *do not* depend on the spectral gap of the mixing matrix that defines the topology of the multigraph. The structure of the multigraph is coded in the times $\{\tau_{i \to j}\}$. We believe that this is an advantage of our guarantees since (11) defines a physical time instead of an iteration rate that depends on the spectral gap.

## 5.2 Interpretation of the upper and lower bounds (11) and (7)

One interesting property of our algorithm is that *some workers will potentially never contribute to optimization because either their computations are too slow or communication times to $j^*$ are too large.* Thus, only a subset of the workers should work to get the optimal time complexity!

Assume in this subsection that the computation and communication are fixed to $\{h_i\}$ and $\{\tau_{i \to j}\}$. One can see that (12) is the $\min_{j \in [n]} \min_{k \in [n]}$ over some formula. In view of our algorithm, an index $j^*$ that minimizes in (12) is the index of a pivot worker that is the most "central" in the multigraph. An index $k^*$ that minimizes $\min_{k \in [n]}$ defines a set of workers $\{\pi_{j^*,1}, \ldots \pi_{j^*,k^*}\}$ that can potentially contribute to optimization. The algorithm and and the time complexity *will not* depend on workers $\{\pi_{j^*,k^*+1}, \ldots \pi_{j^*,n}\}$ because they are too slow or they are too far from worker $j^*$. Thus, up to a constant factor, we have $T_* = {}^{L\Delta}/_\varepsilon \max \big\{ \max\{\tau_{\pi_{j^*,k^*} \to j^*} + \tau_{j^* \to \pi_{j^*,k^*}}, h_{\pi_{j^*,k^*}}\}, \sigma^2/_\varepsilon \big( \sum_{i=1}^{k^*} {}^1/_{h_{\pi_{j^*,i}}} \big)^{-1} \big\}$, where $\tau_{\pi_{j^*,k^*} \to j^*} + \tau_{j^* \to \pi_{j^*,k^*}}$ is the time required to communicate with the farthest worker that can contribute to optimization, $h_{\pi_{j^*,k^*}}$ is the computation time of the slowest worker that can contribute to optimization, and $\sigma^2/_\varepsilon \big( \sum_{i=1}^{k^*} {}^1/_{h_{\pi_{j^*,i}}} \big)^{-1}$ is the time required to "eliminate" enough noise before the algorithm does an update of $x^k$.

## 5.3 Limitations

To get the nearly optimal complexity, it is crucial to select the right pivot worker $j^*$ and spanning trees according to the rules of Corollary 1, which depend on the knowledge of the bounds of times. For now, we believe that is this a price for the optimality. Note that Theorem 5 does not require this knowledge and it works with any $j^*$ and any spanning tree; thus, we can use any heuristic to estimate an optimal $j^*$ and optimal spanning trees. One possible strategy is to estimate the performance of workers and the communication channels using load testings.

## 5.4 Comparison with previous methods

Let us discuss the time complexities of previous methods. Note that none of the previous methods can converge faster than (7) due to our lower bound. First, consider (6) of Minibatch SGD. This time complexity depends on the slowest computation time $\max_{i \in [n]} h_i$ and the slowest communication times $\max_{i,j \in [n]} \tau_{i \to j}$. In the asynchronous setup, it is possible that one the workers is a straggler, i.e., $\max_{i \in [n]} h_i \approx \infty$, and Minibatch SGD can be arbitrarily slow. Our time complexities (10) and (12) are robust to stragglers, and ignore them. Assume the last worker $n$ is a straggler and $h_n = \infty$, then one can take permutations with $\pi_{j,n} = n$ for all $j \in [n]$, and the minimum operator $\min_{k \in [n]}$ in (12) will not choose $k = n$ because $\max\{\tau_{\pi_{j,n} \to j} + \tau_{j \to \pi_{j,n}}, h_{\pi_{j,n}}\} = \infty$ for all $j \in [n]$.

We now consider a recent work by Even et al. (2024), where the authors analyzed Asynchronous SGD in the decentralized setting. In the homogeneous setting, their converge rate depends on the maximum compute delay and, thus, is not robust to stragglers. For the case $\tau_{i \to j} = 0$, our time complexity (11) reduces to (5). At the same time, it was shown (Tyurin and Richtárik, 2023) that the time complexity of Asynchronous SGD for $\tau_{i \to j} = 0$ is strongly worse than (5); thus, the result by Even et al. (2024) is

suboptimal in our setting even if $\tau_{i \to j} = 0$ for all $i, j \in [n]$. The papers by Bornstein et al. (2023); Lian et al. (2018) also consider the same setting, and they share a logic in that they sample a random worker and *wait* while it is calculating a stochastic gradient. If one of the workers is a straggler, they can wait arbitrarily long, while our method automatically ignores slow computations.

## 5.5 Time complexity with dynamic bounds

We can easily generalize Theorem 5 to the case when bounds on the times are not static.

**Theorem 6.** *Consider the assumptions and the parameters from Theorem 4. In each iteration $k$ of Algorithm 3, the computation times of worker $i$ are bounded by $h_i^k$. Let us fix any pivot worker $j^* \in [n]$ and any spanning trees $\overline{st}$ and $\overline{st}_{\mathrm{bc}}$. Then Algorithm 2 converges after at most*

$$\Theta \left( \sum_{k=0}^{\lceil 16L\Delta/\varepsilon \rceil} t^*(\sigma^2/\varepsilon, [h_i^k]_{i=1}^n, [\mu_{i \to j^*}^k + \mu_{j^* \to i}^k]_{i=1}^n) \right) \tag{13}$$

*seconds, where $\mu_{i \to j^*}^k$ ($\mu_{j^* \to i}^k$) is an upper bound on times required to send a vector from worker $i$ to worker $j^*$ (from worker $j^*$ to worker $i$) along the spanning tree $\overline{st}$ (spanning tree $\overline{st}_{\mathrm{bc}}$) in iteration $k$ of Algorithm 3.*

This result is more general than (10), and it shows that our method is robust to changing computation $\{h_i^k\}$ and communication $\{\mu_{i \to j}^k\}$ times bounds during optimization processes. For instance, worker $n$ can have either slow computation or communication to $j^*$ in the first iteration, i.e., $\max\{h_n^1, \mu_{n \to j^*}^1, \mu_{j^* \to n}^1\} \approx \infty$, then our method will ignore it, but if $\max\{h_n^2, \mu_{n \to j^*}^2, \mu_{j^* \to n}^2\}$ is small in the second iteration, then our method can potentially use the stochastic gradients from worker $n$.

# 6 Example: Line or Circle

Figure 2: Line with $\rho_{i+1 \to i} = \rho_{i \to i+1} = \rho$ for all $i \in [n-1]$, $\rho_{i \to j} = \infty$ otherwise. For all $i \neq j \in [n]$, edges $i \to j$ and $j \to i$ are merged and visualized with one undirected edge.

Let us consider Line graphs where we can get more explicit and interpretable formulas for (11). We analyze ND-Mesh, ND-Torus, and Star graphs in Section A. Surprisingly, even in some simple cases like Line or Star graphs, as far as we know, we provide new time complexity results and insights. In Section J, we show that our theoretical results are supported by computational experiments.

We take a Line graph with the computation speeds $h_i = h$ for all $i \in [n]$, and the communication speeds of the edges $\rho_{i \to i+1} = \rho_{i+1 \to i} = \rho$ for all $i \in [n-1]$ and $\rho_{i \to j} = \infty$ for all other $i, j \in [n]$. One can easily show the time required to send a vector between two workers $i, j \in [n]$ equals $\tau_{i \to j} = \tau_{j \to i} = \rho|i - j|$. See an example with $n = 7$ in Fig. 2. We can substitute these values to (11) and get

$$T_{\mathrm{line}} = \frac{L\Delta}{\varepsilon} \min_{j \in [n]} \min_{k \in [n]} \max \left\{ \max\{\rho|j - \pi_{j,k}|, h\}, \frac{\sigma^2 h}{\varepsilon k} \right\}, \tag{14}$$

where $\pi_{j,1} = j$, $\pi_{j,2}, \pi_{j,3} = j+1, j-1$ or $\pi_{j,2}, \pi_{j,3} = j-1, j+1$ (only for $n-1 \geq j \geq 2$) and so forth. For simplicity, assume that $n$ is odd, then, clearly, $j^* = \frac{n-1}{2} + 1$ minimizes $\min_{j \in [n]}$ and $T_{\mathrm{line}} =$

$$\frac{L\Delta}{\varepsilon} \min_{d \in \{0, \dots, \frac{n-1}{2}\}} \max \left\{ \rho d, h, \frac{\sigma^2 h}{\varepsilon(2d+1)} \right\} \simeq \frac{L\Delta}{\varepsilon} \left[ h + \begin{cases} \sigma^2 h/\varepsilon, & \text{if } \sqrt{\sigma^2 h/\varepsilon\rho} \leq 1, \\ \sqrt{\rho\sigma^2 h/\varepsilon}, & \text{if } n > \sqrt{\sigma^2 h/\varepsilon\rho} > 1, \\ \sigma^2 h/n\varepsilon, & \text{if } \sqrt{\sigma^2 h/\varepsilon\rho} \geq n \end{cases} \right]. \tag{15}$$

According to (15), there are three time complexity regimes: i) slow communication, i.e., $\sqrt{\sigma^2 h / \varepsilon \rho} \leq 1$, this inequality means that $\rho$ is so large, that communication between workers will not increase the convergence speed, and the best strategy is to work with only one worker!, ii) medium communication, i.e., $n > \sqrt{\sigma^2 h / \varepsilon \rho} > 1$, more than one worker will participate in the optimization process; however, *not all of them!*, some workers will not contribute since their distances $\tau_{j^* \to \cdot}$ to the pivot worker $j^*$ are large, iii) fast communication, i.e., $\sqrt{\sigma^2 h / \varepsilon \rho} \geq n$, all $n$ workers will participate in optimization because $\rho$ is small.

As far as we know, the result (15) is new even for such a simple structure as a line. Note that these regimes are fundamental and can not be improved due to our lower bound (up to logarithmic factors). For Circle graphs, the result is the same up to a constant factor.

## 7    Heterogeneous Setup

In Section C (in more details), we consider and analyze the problem

$$\min_{x \in \mathbb{R}^d} \left\{ f(x) := \frac{1}{n} \sum_{i=1}^{n} \mathbb{E}_{\xi_i \sim \mathcal{D}_i} \left[ f_i(x; \xi_i) \right] \right\},$$

where $f_i : \mathbb{R}^d \times \mathbb{S}_{\xi_i} \to \mathbb{R}^d$ and $\xi_i$ are random variables with some distributions $\mathcal{D}_i$ on $\mathbb{S}_{\xi_i}$. For all $i \in [n]$, worker $i$ can only access $f_i$. We show that the optimal time complexity is

$$\Theta \left( \frac{L\Delta}{\varepsilon} \max \left\{ \max_{i,j \in [n]} \mu_{i \to j}, \max_{i \in [n]} h_i, \frac{\sigma^2}{n\varepsilon} \left( \frac{1}{n} \sum_{i=1}^{n} h_i \right) \right\} \right) \tag{16}$$

in the heterogeneous setting achieved by a new method, Amelie SGD (Algorithm 5). Amelie SGD is closely related to Rennala SGD but with essential algorithmic changes to make it work with heterogeneous functions. The obtained complexity (16) is worse than (12), which is expected because the heterogeneous setting is more challenging than the homogeneous setting.

## 8    Highlights of Experiments

In Section J, we present experiments with quadratic optimization problems, logistic regression, and a neural network to substantiate our theoretical findings. Here, we focus on highlighting the results from the logistic regression experiments:

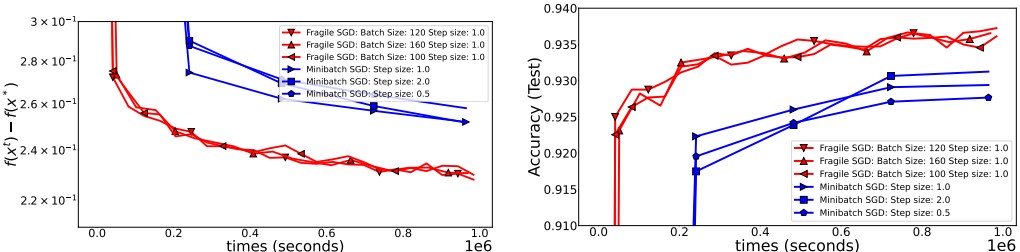

Figure 3: The communication time $\rho = 10$ seconds (Slow communication) in 2D-Mesh

On *MNIST* dataset (LeCun et al., 2010) with 100 workers, Fragile SGD is much faster and has better test accuracy than Minibatch SGD.

## Acknowledgments and Disclosure of Funding

The research reported in this publication was supported by funding from King Abdullah University of Science and Technology (KAUST): i) KAUST Baseline Research Scheme, ii) Center of Excellence for Generative AI, under award number 5940, iii) SDAIA-KAUST Center of Excellence in Artificial Intelligence and Data Science. The work of A.T. was partially supported by the Analytical center under the RF Government (subsidy agreement 000000D730321P5Q0002, Grant No. 70-2021-00145 02.11.2021).

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

# Contents

$$
\begin{array}{c}
1 \xrightarrow{\rho} 2 \xrightarrow{\rho} 3 \xrightarrow{\rho} 4 \xrightarrow{\rho} 5 \\
\rho \Big| \quad \rho \Big| \quad \rho \Big| \quad \rho \Big| \quad \rho \Big| \\
6 \xrightarrow{\rho} 7 \xrightarrow{\rho} 8 \xrightarrow{\rho} 9 \xrightarrow{\rho} 10 \\
\rho \Big| \quad \rho \Big| \quad \rho \Big| \quad \rho \Big| \quad \rho \Big| \\
11 \xrightarrow{\rho} 12 \xrightarrow{\rho} 13 \xrightarrow{\rho} 14 \xrightarrow{\rho} 15 \\
\rho \Big| \quad \rho \Big| \quad \rho \Big| \quad \rho \Big| \quad \rho \Big| \\
16 \xrightarrow{\rho} 17 \xrightarrow{\rho} 18 \xrightarrow{\rho} 19 \xrightarrow{\rho} 20 \\
\rho \Big| \quad \rho \Big| \quad \rho \Big| \quad \rho \Big| \quad \rho \Big| \\
21 \xrightarrow{\rho} 22 \xrightarrow{\rho} 23 \xrightarrow{\rho} 24 \xrightarrow{\rho} 25
\end{array}
$$

(a) 2D-Mesh

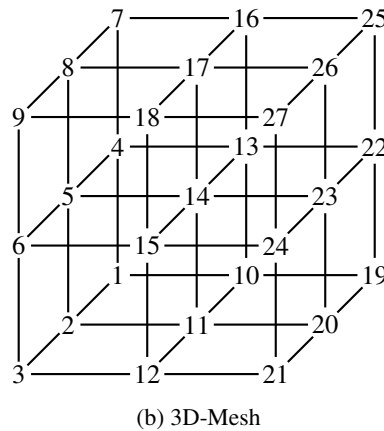

(b) 3D-Mesh

Figure 4: Examples of ND-Mesh graphs. For all $i \neq j \in [n]$, edges $i \to j$ and $j \to i$ are merged and visualized with one undirected edge.

# A   More Examples

## A.1   ND-Mesh or ND-Torus

We now consider a generalization of Line graphs: ND-Mesh graphs. In Figures 4a and 4b, we present examples of 2D-Mesh and 3D-Mesh. For simplicity, assume that $n = (2k+1)^N$ for some $k \in \mathbb{N}$. The computation speeds $h_i = h$ for all $i \in [n]$, and the communicate speeds of the edges $\rho_{i \to j} = \rho$ if workers $i$ and $j$ are connected in a mesh and $\rho_{i \to j} = \infty$ for all other $i, j \in [n]$. Using geometrical reasoning, it is clear an index $j^*$, that minimizes (11), corresponds to the worker in the middle of a graph (13 in Figures 4a and 14 in Figures 4b). Therefore,

$$
T_{\text{2D-Mesh}} = \Theta \left( \frac{L\Delta}{\varepsilon} \min_{k \in [n]} \max \left\{ \tau_{\pi_{j^*,k} \to j^*}, h, \frac{\sigma^2 h}{k \varepsilon} \right\} \right).
$$

For now, let us consider a 2D-Mesh graph. The number of workers, that have the length of the shortest path to $j^*$ equals to $0$, is $1$. The number of workers, that have the length of the shortest path to $j^*$ less or equal to $\rho$, is $5$. The number of workers, that have the length of the shortest path to $j^*$ less or equal to $2\rho$, is $13$. In general, the number of workers, that have the length of the shortest path to $j^*$ less or equal to $\sqrt{k}\rho$, is $\Theta(k)$ for all $k \in \{0, \ldots, \Theta(n)\}$. It means

$$
T_{\text{2D-Mesh}} = \Theta \left( \frac{L\Delta}{\varepsilon} \min_{k \in \{0, \ldots, \Theta(n)\}} \max \left\{ \sqrt{k}\rho, h, \frac{\sigma^2 h}{(k+1)\varepsilon} \right\} \right)
$$

$$
= \Theta \left( \frac{L\Delta}{\varepsilon} \left[ h + \begin{cases} \frac{\sigma^2 h}{\varepsilon}, & \left( \frac{\sigma^2 h}{\rho \varepsilon} \right)^{2/3} \leq 1, \\ \rho^{2/3} \left( \frac{\sigma^2 h}{\varepsilon} \right)^{1/3}, & n > \left( \frac{\sigma^2 h}{\rho \varepsilon} \right)^{2/3} > 1, \\ \frac{\sigma^2 h}{n \varepsilon}, & \left( \frac{\sigma^2 h}{\rho \varepsilon} \right)^{2/3} \geq n \end{cases} \right] \right).
$$

Using the same reasoning, for the general case with a ND-Mesh graph, we get

$$
T_{\text{ND-Mesh}} = \Theta \left( \frac{L\Delta}{\varepsilon} \min_{k \in \{0, \ldots, \Theta(n)\}} \max \left\{ k^{1/N}\rho, h, \frac{\sigma^2 h}{(k+1)\varepsilon} \right\} \right)
$$

$$
= \Theta \left( \frac{L\Delta}{\varepsilon} \left[ h + \begin{cases} \frac{\sigma^2 h}{\varepsilon}, & \left( \frac{\sigma^2 h}{\rho \varepsilon} \right)^{N/(N+1)} \leq 1, \\ \rho^{N/(N+1)} \left( \frac{\sigma^2 h}{\varepsilon} \right)^{1/(N+1)}, & n > \left( \frac{\sigma^2 h}{\rho \varepsilon} \right)^{N/(N+1)} > 1, \\ \frac{\sigma^2 h}{n \varepsilon}, & \left( \frac{\sigma^2 h}{\rho \varepsilon} \right)^{N/(N+1)} \geq n \end{cases} \right] \right).
$$

As in Line graphs, these complexities have three regimes depending on the problem's parameters. Up to a constant factor, the same conclusions apply to ND-Torus.

## A.2 Star graph

Let us consider Star graphs with different computation and communication speeds. Let us fix a graph with $n + 1$ workers, where one worker with the index $n + 1$ is in the center, and all other $n$ workers are only directly connected to worker $n + 1$. Then,

$$\tau_{i \to j} = \rho_{i \to n+1} + \rho_{n+1 \to j} \quad \forall i \neq j \in [n+1] \text{ and } \tau_{i \to i} = 0 \quad \forall i \in [n].$$

Using these constraints, we can simplify (11). There are two possible best strategies: i) a pivot worker $j^*$ works locally without communications, ii) a pivot worker $j^*$ works with communications but it would necessary require to communicate through the central worker; in this case, the central worker $n + 1$ is a pivot worker. Therefore, we get

$$T_{\text{star}} = \min \left[ \frac{L\Delta}{\varepsilon} \min_{j \in [n]} \max \left\{ h_j, \frac{\sigma^2 h_j}{\varepsilon} \right\} \right.$$

$$\left. \frac{L\Delta}{\varepsilon} \min_{k \in [n]} \max \left\{ \max\{\rho_{\pi_{n+1,k} \to n+1} + \rho_{n+1 \to \pi_{n+1,k}}, h_{\pi_{n+1,k}}\}, \frac{\sigma^2}{\varepsilon} \left( \sum_{i=1}^{k} \frac{1}{h_{\pi_{n+1,i}}} \right)^{-1} \right\} \right]$$

since $\tau_{n+1 \to \pi_{n+1,k}} = \rho_{n+1 \to \pi_{n+1,k}}$ for all $k \in [n+1]$. Let us slightly simplify the result and assume that broadcasting from the central worker is fast, i.e., $\rho_{n+1 \to i} \leq \rho_{i \to n+1}$ for all $i \in [n+1]$, then

$$T_{\text{star}} = \min \left[ \underbrace{\frac{L\Delta}{\varepsilon} \min_{j \in [n]} \max \left\{ h_j, \frac{\sigma^2 h_j}{\varepsilon} \right\}}_{T_{\text{slow comm.}}}, \right.$$

$$\left. \underbrace{\frac{L\Delta}{\varepsilon} \min_{k \in [n]} \max \left\{ \max\{\rho_{\pi_{n+1,k} \to n+1}, h_{\pi_{n+1,k}}\}, \frac{\sigma^2}{\varepsilon} \left( \sum_{i=1}^{k} \frac{1}{h_{\pi_{n+1,i}}} \right)^{-1} \right\}}_{T_{\text{fast comm.}}} \right].$$

Tyurin et al. (2024) also considered Star graphs and showed that $T_{\text{fast comm.}}$ is the optimal time complexity for methods that communicate through the central worker. Our result is more general since we also consider decentralized methods and capture the term $T_{\text{slow comm.}}$ that can be potentially smaller if communication is slow. Tyurin et al. (2024) conjectured that $T_{\text{star}}$ is the optimal bound, and we proved it (up to log factor) here.

## A.3 General case

We show how one can use our generic result (11) to get an explicit formula in some cases. For the general case with arbitrary communication and computation times, the minimizers $j$ and $k$ in (11) can be found in the following way. First, we have to find $\tau_{i \to j}$ using any algorithm that solves *the all-pairs shortest path problem* (e.g., Floyd–Warshall algorithm (Floyd, 1962)). Once we know $\tau_{i \to j}$, we should sort $\{\max\{\tau_{i \to j} + \tau_{j \to i}, h_i\}\}_{k=1}^{n}$ for all $j \in [n]$ to find the permutations. Finally, we have enough information to calculate $t^*$ from Definition 2.

## B  On the Connection to the Gossip Framework

Most of the previous methods were designed for a different setting, gossip-type communication. In fact, our setting is more general than the gossip communication. Indeed, recall that in the gossip communication, worker $i$ is allowed to get vectors from other workers through the operation $\sum_{j=1}^{n} w_{ij} x_j$, where $w_{ij} \in \{0, 1\}$. This is equivalent to our setting for the case when the communication time $\rho_{ij} = \infty$ when $w_{ij}$ is zero, and $\rho_{ij} = 1$ if $w_{ij}$ is not zero, and worker $i$ sums the received vectors. But our setting is richer since we allow different communication and computation times and allow workers to do whatever they like with vectors (not only to sum).

Moreover, the gossip framework codes the communication graph through the matrix $\{w_{ij}\}$. We propose to code graphs through the times $\{\rho_{ij}\}$ ($\{\tau_{ij}\}$). Our approach is closer to real scenarios because, as we explained previously, it includes the gossip framework.

## C    Heterogeneous Setup

We now consider the heterogeneous setting. The only difference is instead of (1), we consider the following problem.

$$\min_{x \in \mathbb{R}^d} \left\{ f(x) := \frac{1}{n} \sum_{i=1}^{n} \underbrace{\mathbb{E}_{\xi_i \sim \mathcal{D}_i} \left[ f_i(x; \xi_i) \right]}_{f_i(x):=} \right\}, \tag{17}$$

where $f_i : \mathbb{R}^d \times \mathbb{S}_{\xi_i} \to \mathbb{R}^d$ and $\xi_i$ are random variables with some distributions $\mathcal{D}_i$ on $\mathbb{S}_{\xi_i}$. We now present our upper and lower bounds and discuss them.

### C.1    Lower bound

In Section F, we prove the following lower bound.

**Theorem 7** (Lower Bound; Simplified Presentation of Theorem 20). *Consider Protocol 1 with* $\nabla f_i(\cdot; \cdot)$. *We take any* $h_i \geq 0$ *and* $\tau_{i \to j} \geq 0$ *for all* $i, j \in [n]$ *such that* $\tau_{i \to j} \leq \tau_{i \to k} + \tau_{k \to j}$ *for all* $i, k, j \in [n]$. *We fix* $L, \Delta, \varepsilon, \sigma^2 > 0$ *that satisfy the inequality* $\varepsilon < cL\Delta$ *for some universal constant* $c$.
*For any (zero-respecting) algorithm, there exists a function* $f = \frac{1}{n} \sum_{i=1}^{n} f_i$, *which satisfy Assumptions 1, 2 and* $f(0) - f^* \leq \Delta$, *and stochastic gradient mappings* $\nabla f_i(\cdot; \cdot)$, *which satisfy Assumption 3* $(\mathbb{E}_\xi[\nabla f_i(x; \xi)] = \nabla f_i(x)$ *and* $\mathbb{E}_\xi[\|\nabla f_i(x; \xi) - \nabla f_i(x)\|^2] \leq \sigma^2)$, *such that the required time to find* $\varepsilon$*–solution is*

$$\Omega \left( \frac{L\Delta}{\varepsilon} \max \left\{ \max_{i,j \in [n]} \tau_{i \to j}, \max_{i \in [n]} h_i, \frac{\sigma^2}{n\varepsilon} \left( \frac{1}{n} \sum_{i=1}^{n} h_i \right) \right\} \right)$$

*seconds.*

### C.2    Amelie SGD**: optimal method in the heterogeneous setting**

We now present a new method based on Malenia SGD from (Tyurin and Richtárik, 2023) and our Fragile SGD. As in Fragile SGD, we also have $n + 1$ processes running in all workers. The main idea is that all workers calculate stochastic gradients in parallel and accumulate them locally. Process 0 zero waits for the moment when $\frac{n}{b_{j^*}} \geq \frac{S}{n}$. Process $i$ calculates $b_i$ in Line 20 of Algorithm 7, and sends it to Process next$_{\overline{st},j^*}(i)$. Thus, $b_{j^*}$ accumulates the sum $\sum_{i=1}^{n} \frac{1}{s_i^k}$, which decreases with time since $s_i^k$ is the number of calculated stochastic gradients in worker $i$ in different moments of time. Therefore, $\frac{n}{b_{j^*}} \geq \frac{S}{n}$ will hold at some point of time if workers continue computing gradients. Finally, Algorithm 6 runs all reduce and does the update of $x^k$.

**Theorem 8.** *Let Assumptions 1 and 2 hold for the function* $f$ *and Assumption 3 holds for the functions* $f_i$ *for all* $i \in [n]$. *We take* $\gamma = 1/2L$, *the parameter* $S = \max\{\lceil \sigma^2/\varepsilon \rceil, n\}$, *any pivot worker* $j^* \in [n]$, *and any spanning trees* $\overline{st}$ *and* $\overline{st}_{bc}$, *in Algorithm 5. For all iterations number* $K \geq 16L\Delta/\varepsilon$, *we get* $\frac{1}{K} \sum_{k=0}^{K-1} \mathbb{E} \left[ \|\nabla f(x^k)\|^2 \right] \leq \varepsilon$.

Note that Theorem 8 states the convergence of Amelie SGD even if the computation and communication speeds are unbounded.

**Theorem 9.** *Consider the assumptions and the parameters from Theorem 8. For any pivot worker* $j^* \in [n]$ *and any spanning trees* $\overline{st}$ *and* $\overline{st}_{bc}$, *Algorithm 5 converges after at most*

$$\Theta \left( \frac{L\Delta}{\varepsilon} \max \left\{ \max_{i,j \in [n]} \mu_{i \to j}, \max_{i \in [n]} h_i, \frac{\sigma^2}{n\varepsilon} \left( \frac{1}{n} \sum_{i=1}^{n} h_i \right) \right\} \right)$$

*seconds, where* $\mu_{i \to j^*}^k$ $(\mu_{j^* \to i}^k)$ *is an upper bound on times required to send a vector from worker* $i$ *to worker* $j^*$ *(from worker* $j^*$ *to worker* $i$*) along the spanning tree* $\overline{st}$ *(spanning tree* $\overline{st}_{bc}$*) for all* $i \in [n]$.

---

**Algorithm 5** Amelie SGD

---

1: **Input:** starting point $x^0$, stepsize $\gamma$, parameter $S$, pivot worker $j^*$, spanning trees $\overline{st}$ and $\overline{st}_{\mathrm{bc}}$
2: Start Process 0 (Alg. 6) in worker $j^*$
3: Start Process $i$ (Alg. 7) in all workers for all $i \in [n]$ (including worker $j^*$)

---

**Algorithm 6** Process 0 (running in worker $j^*$)

---

1: **for** $k = 0, 1, \dots, K - 1$ **do**
2:     Broadcast $x^k$ to all workers using the spanning tree $\overline{st}_{\mathrm{bc}}$
3:     Init $s^k = 0$
4:     **while** $s^k < \frac{S}{n}$ **do**
5:         Wait for a message $b_{j^*}$ from Process $j^*$
6:         Calculate $s^k = \frac{n}{b_{j^*}}$
7:     **end while**
8:     Run all reduce with $\{\frac{1}{s_i^k} g_i^k\}_{i=1}^n$ to find $g^k = \frac{1}{n} \sum\limits_{i=1}^{n} \frac{1}{s_i^k} g_i^k$ using the spanning tree $\overline{st}$
9:     $x^{k+1} = x^k - \gamma g^k$
10: **end for**

---

**Algorithm 7** Process $i$ (running in worker $i$)

---

1: **while** True **do**
2:     Get a new point $x^k$ broadcasted by Process 0
3:     Init $(g_i^k, s_i^k) = (0, 0)$
4:     Init $b_{i,p} = \infty$ for all $p \in [n]$ s.t. $\mathrm{next}_{\overline{st},j^*}(p) = i$
5:     Run the following three functions in parallel and go to Line 24
6:     **function** CalculateStochasticGradients
7:         **while** True
8:             Calculate $\nabla f_i(x^k; \xi), \quad \xi \sim \mathcal{D}_i$
9:             Run atomic add $g_i^k = g_i^k + \nabla f_i(x^k; \xi), s_i^k = s_i^k + 1$
10:         **end while**
11:     **end function**
12:     **function** ReceiveCountersFromPreviousWorkers
13:         **while** True
14:             Wait for a message $b_p$ from any Process $p$ s.t. $\mathrm{next}_{\overline{st},j^*}(p) = i$
15:             Run atomic update $b_{i,p} = b_p$
16:         **end while**
17:     **end function**
18:     **function** SendCounterToNextWorker
19:         **while** True
20:             Run atomic sum $b_i = \sum\limits_{p \in [n]: \mathrm{next}_{\overline{st},j^*}(p) = i} b_{i,p} + \frac{1}{s_i^k}$
21:             Send $b_i \in \mathbb{R} \cup \{\infty\}$ (one float) to Process $\mathrm{next}_{\overline{st},j^*}(i)$ and wait while it sending
                (Process $j^*$ sends to Process 0 by the definition of $\mathrm{next}_{\overline{st},j^*}(\cdot)$)
22:         **end while**
23:     **end function**
24:     Wait for new point. If receives new point, stop all computations in functions, and continue
25:     Ignore all non-received messages
26: **end while**

---

**Corollary 2.** *Consider the assumptions and the parameters from Theorem 9. Let us take any pivot worker $j^* \in [n]$ and a spanning tree $\overline{st}$ (spanning tree $\overline{st}_{\mathrm{bc}}$) that connects every worker $i$ to worker $j^*$ (worker $j^*$ to worker $i$) with the shortest distance $\tau_{i \to j^*}$ ($\tau_{j^* \to i}$), then Algorithm 5 converges after at most*

$$\Theta\left(\frac{L\Delta}{\varepsilon}\max\left\{\max_{i,j\in[n]}\tau_{i\to j},\max_{i\in[n]}h_i,\frac{\sigma^2}{n\varepsilon}\left(\frac{1}{n}\sum_{i=1}^{n}h_i\right)\right\}\right) \tag{18}$$

*seconds.*

## C.3  Discussion

Corollary 2 together with Theorem 7 states that the time complexity (18) is optimal. The result is pessimistic since the time complexity depends on the "diameter" $\max_{i,j\in[n]}\tau_{i\to j}$ and the slowest performance $\max_{i\in[n]}h_i$. A similar dependence was observed in (Lu and De Sa, 2021; Tyurin and Richtárik, 2023). As in (Tyurin and Richtárik, 2023), the stochastic term $\sigma^2/n\varepsilon\left(1/n\sum_{i=1}^{n}h_i\right)$ depends on the average of $\{h_i\}$ if $\sigma^2/\varepsilon$ is large.

## C.4  Comparison with previous methods

Let us consider Minibatch SGD described in Section 2.3. This method converges after

$$\Theta\left(\frac{L\Delta}{\varepsilon}\max\left\{\max_{i,j\in[n]}\tau_{i\to j},\max_{i\in[n]}h_i,\frac{\sigma^2}{n\varepsilon}\max\{\max_{i,j\in[n]}\tau_{i\to j},\max_{i\in[n]}h_i\}\right\}\right)$$

seconds in the heterogeneous setting. If $\sigma^2/\varepsilon$ is large, then Amelie SGD can be at least

$$\max\{\max_{i,j\in[n]}\tau_{i\to j},\max_{i\in[n]}h_i\}/\left(\frac{1}{n}\sum_{i=1}^{n}h_i\right)$$

times faster than Minibatch SGD. There are many more advanced methods (Lu and De Sa, 2021; Vogels et al., 2021; Even et al., 2024) that work in decentralized stochastic heterogeneous setting. Lu and De Sa (2021) developed similar lower and upper bounds, but there are at least three main differences: i) they derived an *iteration complexity* instead of a time complexity and assumed the performances of all workers are the same ii) the obtained lower bound holds only for one particular multigraph while our complexity holds for any multigraph iii) they assumed the smoothness of $f_i$, while we consider the smoothness of $f$. The RelaySGD and Gradient Tracking methods by Vogels et al. (2021); Liu et al. (2024) wait for the slowest worker; thus, they depend on $\max_{i\in[n]}h_i$ in all regimes of $\sigma^2/\varepsilon$, unlike our method. Even et al. (2024) consider the heterogeneous asynchronous setting, but their method assumes the similarity of the functions $f_i$, which is not required in our method, and Amelie SGD converges even if there is no similarity of functions.

# D  Convex Functions in the Homogeneous and Heterogeneous Setups

We will be slightly more brief in the convex setting since the idea, the structure of time complexities, and the general approach do not change significantly. For instance, instead of the time complexity (11) that we get for the nonconvex case, in the nonsmooth convex case, we get (20). Thus, we will get $\Theta\left(\frac{M^2R^2}{\varepsilon^2}\min_{j\in[n]}t^*(\sigma^2/M^2,\dots)\right)$ instead of $\Theta\left(\frac{L\Delta}{\varepsilon}\min_{j\in[n]}t^*(\sigma^2/\varepsilon,\dots)\right)$. The same idea applies to the smooth convex case. In the convex setting, we need the following assumptions.

## D.1  Assumptions in convex world

**Assumption 4.** *The function $f$ is convex and attains a minimum at some point $x^* \in \mathbb{R}^d$.*

**Assumption 5.** *The function $f$ is $M$–Lipschitz, i.e.,*

$$|f(x) - f(y)| \leq M\|x - y\|, \quad \forall x, y \in \mathbb{R}^d$$

*for some $M \in (0, \infty]$.*

**Assumption 6.** *For all $x \in \mathbb{R}^d$, stochastic (sub)gradients $\nabla f(x; \xi)$ are unbiased and are $\sigma^2$-variance-bounded, i.e., $\mathbb{E}_{\xi \sim \mathcal{D}}[\nabla f(x; \xi)] \in \partial f(x)$ and $\mathbb{E}_{\xi \sim \mathcal{D}}\left[\|\nabla f(x; \xi) - \mathbb{E}_{\xi \sim \mathcal{D}}[\nabla f(x; \xi)]\|^2\right] \leq \sigma^2$, where $\sigma^2 \geq 0$.*

### D.2 Homogeneous setup and nonsmooth case

**Theorem 10.** *Let Assumptions 4, 5 and 6 hold. Choose any $\varepsilon > 0$. Let us take the batch size $S = \max\left\{\lceil \sigma^2/M^2 \rceil, 1\right\}$, stepsize $\gamma = \frac{\varepsilon}{M^2 + \sigma^2/S} \in \left[\frac{\varepsilon}{2M^2}, \frac{\varepsilon}{M^2}\right]$, any pivot worker $j^* \in [n]$, and any spanning trees $\overline{st}$ and $\overline{st}_{bc}$ in Algorithm 2. Then after $K \geq 2M^2 R^2/\varepsilon^2$ iterations the method guarantees $\mathbb{E}\left[f(\widehat{x}^K)\right] - f(x^*) \leq \varepsilon$, where $\widehat{x}^K = \frac{1}{K}\sum_{k=0}^{K-1} x^k$ and $R = \|x^* - x^0\|$.*

*Proof.* The proof of Theorem 10 is almost the same as in Theorem 4 (see Section G). The proof of Theorem 4 states that the steps of Fragile SGD are equivalent to the classical SGD method. Thus, we can use the classical result from the literature (Lan, 2020). Using Theorem 22, we get

$$\mathbb{E}\left[f(\widehat{x}^K)\right] - f(x^*) \leq \varepsilon$$

if

$$K \geq \frac{2M^2 \left\|x^* - x^0\right\|^2}{\varepsilon^2} \geq \frac{(M^2 + \frac{\sigma^2}{S}) \left\|x^* - x^0\right\|^2}{\varepsilon^2}$$

for the stepsize

$$\gamma = \frac{\varepsilon}{M^2 + \frac{\sigma^2}{S}} \in \left[\frac{\varepsilon}{2M^2}, \frac{\varepsilon}{M^2}\right],$$

where we use the fact that $S \geq \sigma^2/M^2$. $\qquad\square$

**Theorem 11.** *Consider the assumptions and the parameters from Theorem 10. For any pivot worker $j^* \in [n]$ and spanning trees $\overline{st}$ and $\overline{st}_{bc}$, Algorithm 2 converges after at most*

$$\Theta\left(\frac{M^2 R^2}{\varepsilon^2} t^*(\sigma^2/M^2, [h_i]_{i=1}^n, [\mu_{i \to j^*} + \mu_{j^* \to i}]_{i=1}^n)\right) \tag{19}$$

*seconds, where $\mu_{i \to j^*}$ $(\mu_{j^* \to i})$ is an upper bound on times required to send a vector from worker $i$ to worker $j^*$ (from worker $j^*$ to worker $i$) along the spanning tree $\overline{st}$ (spanning tree $\overline{st}_{bc}$).*

*Proof.* The proof is identical to the proof of Theorem 5. $\qquad\square$

**Corollary 3.** *Consider the assumptions and the parameters from Theorem 11. Let us take a pivot worker $j^* = \arg\min_{j \in [n]} t^*(\sigma^2/M^2, [h_i]_{i=1}^n, [\tau_{i \to j} + \tau_{j \to i}]_{i=1}^n)$, and a spanning tree $\overline{st}$ (spanning tree $\overline{st}_{bc}$) that connects every worker $i$ to worker $j^*$ (worker $j^*$ to every worker $i$) with the shortest distance $\tau_{i \to j^*}$ $(\tau_{j^* \to i})$. Then Algorithm 2 converges after at most*

$$\Theta\left(\frac{M^2 R^2}{\varepsilon^2} \min_{j \in [n]} t^*(\sigma^2/M^2, [h_i]_{i=1}^n, [\tau_{i \to j} + \tau_{j \to i}]_{i=1}^n)\right) \tag{20}$$

*seconds.*

### D.3 Homogeneous setup and smooth case

In the homogeneous and smooth case, we will slightly modify Fragile SGD. Instead of Line 8 of Algorithm 3, we use the following steps:

$$\begin{aligned}
\gamma_{k+1} &= \gamma \cdot (k+1), \quad \alpha_{k+1} = 2/(k+2) \\
y^{k+1} &= (1 - \alpha_{k+1})x^k + \alpha_{k+1} u^k, \qquad (u^0 = x^0) \\
u^{k+1} &= u^k - \frac{\gamma_{k+1}}{s^k} g^k, \\
x^{k+1} &= (1 - \alpha_{k+1})x^k + \alpha_{k+1} u^{k+1}.
\end{aligned} \tag{21}$$

We will call such a method the Accelerated Fragile SGD method. The idea is to use the acceleration technique from (Lan, 2020) (which is based on (Nesterov, 1983)).

**Theorem 12.** *Let Assumptions 4, 1 and 3 hold. Choose any $\varepsilon > 0$. Let us take the batch size*
$S = \max\left\{\left\lceil (\sigma^2 R)/(\varepsilon^{3/2}\sqrt{L})\right\rceil, 1\right\}, \gamma = \min\left\{\frac{1}{4L}, \left[\frac{3R^2 S}{4\sigma^2(K+1)(K+2)^2}\right]^{1/2}\right\}$, *any pivot worker $j^*$,*
*and any spanning trees $\overline{st}$ and $\overline{st}_{bc}$ in Accelerated Method 2 (*Accelerated Fragile SGD*), then after*
$K \geq \frac{8\sqrt{L}R}{\sqrt{\varepsilon}}$ *iterations the method guarantees that $\mathbb{E}\left[f(x^K)\right] - f(x^*) \leq \varepsilon$, where $R = \left\|x^* - x^0\right\|$.*

*Proof.* Using the same reasoning as in Theorem 4, Accelerated Fragile SGD is just the classical accelerated stochastic gradient method with a batch size greater or equal to $S$. We can use Proposition 4.4 from Lan (2020). For the stepsize

$$\gamma = \min\left\{\frac{1}{4L}, \left[\frac{3R^2 S}{4\sigma^2(K+1)(K+2)^2}\right]^{1/2}\right\},$$

we have

$$\mathbb{E}\left[f(x^K)\right] - f(x^*) \leq \frac{4LR^2}{K^2} + \frac{4\sqrt{\sigma^2 R^2}}{\sqrt{SK}}.$$

Therefore,

$$\mathbb{E}\left[f(x^K)\right] - f(x^*) \leq \varepsilon$$

if

$$K \geq \frac{8\sqrt{L}R}{\sqrt{\varepsilon}} \geq 8\max\left\{\frac{\sqrt{L}R}{\sqrt{\varepsilon}}, \frac{\sigma^2 R^2}{\varepsilon^2 S}\right\},$$

where we use the choice of $S$. $\qquad\square$

**Theorem 13.** *Consider the assumptions and the parameters from Theorem 12. For any pivot worker $j^* \in [n]$ and spanning trees $\overline{st}$ and $\overline{st}_{bc}$, Accelerated Algorithm 2 (*Accelerated Fragile SGD*) converges after at most*

$$\Theta\left(\frac{\sqrt{L}R}{\sqrt{\varepsilon}}t^*\left(\frac{\sigma^2 R}{\varepsilon^{3/2}\sqrt{L}}, [h_i]_{i=1}^n, [\mu_{i\to j^*} + \mu_{j^*\to i}]_{i=1}^n\right)\right) \qquad (22)$$

*seconds, where $\mu_{i\to j^*}$ ($\mu_{j^*\to i}$) is an upper bound on times required to send a vector from worker $i$ to worker $j^*$ (from worker $j^*$ to worker $i$) along the spanning tree $\overline{st}$ (spanning tree $\overline{st}_{bc}$).*

*Proof.* The proof is identical to the proof of Theorem 5. $\qquad\square$

**Corollary 4.** *Consider the assumptions and the parameters from Theorem 13. Let us take a pivot worker $j^* = \arg\min_{j\in[n]} t^*\left(\frac{\sigma^2 R}{\varepsilon^{3/2}\sqrt{L}}, [h_i]_{i=1}^n, [\tau_{i\to j} + \tau_{j\to i}]_{i=1}^n\right)$, and a spanning tree $\overline{st}$ (spanning tree $\overline{st}_{bc}$) that connects every worker $i$ to worker $j^*$ (worker $j^*$ to every worker $i$) with the shortest distance $\tau_{i\to j^*}$ ($\tau_{j^*\to i}$). Then Accelerated Algorithm 2 (*Accelerated Fragile SGD*) converges after at most*

$$\Theta\left(\frac{\sqrt{L}R}{\sqrt{\varepsilon}}\min_{j\in[n]} t^*\left(\frac{\sigma^2 R}{\varepsilon^{3/2}\sqrt{L}}, [h_i]_{i=1}^n, [\tau_{i\to j} + \tau_{j\to i}]_{i=1}^n\right)\right) \qquad (23)$$

*seconds.*

### D.4 Heterogeneous setup and nonsmooth case

Consider the optimization problem (17).

**Theorem 14.** *Let Assumptions 4, 5 hold for the function $f$ and Assumption 6 holds for the functions $f_i$ for all $i \in [n]$. Choose any $\varepsilon > 0$. Let us take the batch size $S = \max\left\{\left\lceil\sigma^2/M^2\right\rceil, n\right\}$, $\gamma = \frac{\varepsilon}{M^2 + \sigma^2/S} \in \left[\frac{\varepsilon}{2M^2}, \frac{\varepsilon}{M^2}\right]$, any pivot worker $j^* \in [n]$, and any spanning trees $\overline{st}$ and $\overline{st}_{bc}$, in Algorithm 5, then after $K \geq 2M^2 R^2/\varepsilon^2$ iterations the method guarantees that $\mathbb{E}\left[f(\widehat{x}^K)\right] - f(x^*) \leq \varepsilon$, where $\widehat{x}^K = \frac{1}{K}\sum_{k=0}^{K-1} x^k$ and $R = \left\|x^* - x^0\right\|$.*

*Proof.* The proof of Theorem 14 is almost the same as in Theorems 8 (see Section H). The proof of Theorem 8 states that the steps of Amelie SGD are equivalent to the classical SGD method. Thus, we can use the classical result from the literature (Lan, 2020). Using Theorem 22, we get

$$\mathbb{E}\left[f(\widehat{x}^K)\right] - f(x^*) \leq \varepsilon$$

if

$$K \geq \frac{2M^2 \left\| x^* - x^0 \right\|^2}{\varepsilon^2} \geq \frac{(M^2 + \frac{\sigma^2}{S}) \left\| x^* - x^0 \right\|^2}{\varepsilon^2}$$

for the stepsize

$$\gamma = \frac{\varepsilon}{M^2 + \frac{\sigma^2}{S}} \in \left[ \frac{\varepsilon}{2M^2}, \frac{\varepsilon}{M^2} \right],$$

where we use the fact that $S \geq \sigma^2/M^2$. □

**Theorem 15.** *Consider the assumptions and the parameters from Theorem 14. For any pivot worker $j^* \in [n]$ and any spanning trees $\overline{st}$ and $\overline{st}_{\text{bc}}$, Algorithm 5 converges after at most*

$$\Theta\left( \frac{M^2 R^2}{\varepsilon^2} \max\left\{ \max_{i,j \in [n]} \mu_{i \to j}, \max_{i \in [n]} h_i, \frac{\sigma^2}{nM^2}\left( \frac{1}{n}\sum_{i=1}^n h_i \right) \right\} \right) \tag{24}$$

*seconds, where $\mu_{i \to j^*}^k$ ($\mu_{j^* \to i}^k$) is an upper bound on times required to send a vector from worker $i$ to worker $j^*$ (worker $j^*$ to worker $i$) along the spanning tree $\overline{st}$ (spanning tree $\overline{st}_{\text{bc}}$) for all $i \in [n]$.*

*Proof.* The proof is identical to the proof of Theorem 9. □

**Corollary 5.** *Consider the assumptions and the parameters from Theorem 15. Let us take any pivot worker $j^* \in [n]$ and a spanning tree $\overline{st}$ (spanning tree $\overline{st}_{\text{bc}}$) that connects every worker $i$ to worker $j^*$ (worker $j^*$ to worker $i$) with the shortest distance $\tau_{i \to j^*}$ ($\tau_{j^* \to i}$), then Algorithm 5 converges after at most*

$$\Theta\left( \frac{M^2 R^2}{\varepsilon^2} \max\left\{ \max_{i,j \in [n]} \tau_{i \to j}, \max_{i \in [n]} h_i, \frac{\sigma^2}{nM^2}\left( \frac{1}{n}\sum_{i=1}^n h_i \right) \right\} \right)$$

*seconds.*

### D.5 Heterogeneous setup and smooth case

Consider the optimization problem (17). In this section, we use the same idea as in Section D.3. We will modify Amelie SGD and, instead of Line 9 from Algorithm 6, we use the lines (21). We call such a method the Accelerated Amelie SGD method.

**Theorem 16.** *Let Assumptions 4 and 1 hold for the function $f$ and Assumption 3 holds for the functions $f_i$. Choose any $\varepsilon > 0$. Let us take the batch size $S = \max\left\{ \left\lceil (\sigma^2 R)/(\varepsilon^{3/2}\sqrt{L}) \right\rceil, n \right\}$, $\gamma = \min\left\{ \frac{1}{4L}, \left[ \frac{3R^2 S}{4\sigma^2(K+1)(K+2)^2} \right]^{1/2} \right\}$, any pivot worker $j^*$, and any spanning trees $\overline{st}$ and $\overline{st}_{\text{bc}}$ in Accelerated Method 5 (Accelerated Amelie SGD), then after $K \geq \frac{8\sqrt{L}R}{\sqrt{\varepsilon}}$ iterations the method guarantees that $\mathbb{E}\left[ f(x^K) \right] - f(x^*) \leq \varepsilon$, where $R = \left\| x^* - x^0 \right\|$.*

*Proof.* Accelerated Amelie SGD is equivalent to the accelerated stochastic gradient method with a mini-batch from Lan (2020). The proof repeats the proofs of Theorem 12 and Theorem 8. □

**Theorem 17.** *Consider the assumptions and the parameters from Theorem 16. For any pivot worker $j^* \in [n]$ and any spanning trees $\overline{st}$ and $\overline{st}_{\text{bc}}$, Accelerated Algorithm 5 (Accelerated Amelie SGD) converges after at most*

$$\Theta\left( \frac{\sqrt{L}R}{\sqrt{\varepsilon}} \max\left\{ \max_{i,j \in [n]} \mu_{i \to j}, \max_{i \in [n]} h_i, \frac{\sigma^2 R}{n\varepsilon^{3/2}\sqrt{L}}\left( \frac{1}{n}\sum_{i=1}^n h_i \right) \right\} \right) \tag{25}$$

*seconds, where $\mu_{i \to j^*}^k$ ($\mu_{j^* \to i}^k$) is an upper bound on times required to send a vector from worker $i$ to worker $j^*$ (worker $j^*$ to worker $i$) along the spanning tree $\overline{st}$ (spanning tree $\overline{st}_{\text{bc}}$) for all $i \in [n]$.*

*Proof.* The proof is identical to the proof of Theorem 9. □

**Corollary 6.** *Consider the assumptions and the parameters from Theorem 17. Let us take any pivot worker $j^* \in [n]$ and a spanning tree $\overline{st}$ (spanning tree $\overline{st}_{\mathrm{bc}}$) that connects every worker $i$ to worker $j^*$ (worker $j^*$ to worker $i$) with the shortest distance $\tau_{i \to j^*}$ ($\tau_{j^* \to i}$), then Accelerated Algorithm 5 (*Accelerated Amelie SGD*) converges after at most*

$$\Theta\left(\frac{\sqrt{L}R}{\sqrt{\varepsilon}} \max\left\{\max_{i,j\in[n]} \tau_{i \to j}, \max_{i\in[n]} h_i, \frac{\sigma^2 R}{n\varepsilon^{3/2}\sqrt{L}}\left(\frac{1}{n}\sum_{i=1}^{n} h_i\right)\right\}\right) \tag{26}$$

*seconds.*

### D.6 On lower bounds

In previous subsections, we provide upper bounds on the time complexities for different classes of convex functions and optimization problems. Using the same reasoning as in Section 4 and (Tyurin and Richtárik, 2023)[Section B], we conjecture that the obtained upper bounds are tight and optimal (up to log factors in the homogeneous case).

### D.7 Previous works

Let us consider the time complexities (20) and (23) (accelerated rate) in the homogeneous and convex cases. When $\tau_{i \to j} = 0$ for all $i, j \in [n]$, Even et al. (2024) recovers the time complexity (nonaccelerated in the smooth case) of Asynchronous SGD (Mishchenko et al., 2022; Koloskova et al., 2022) which is suboptimal (Tyurin and Richtárik, 2023). When the communication is free, we recover the time complexity (accelerated in the smooth case) of Accelerated Rennala SGD from (Tyurin and Richtárik, 2023), which is optimal if $\tau_{i \to j} = 0$ for all $i, j \in [n]$.

In the heterogeneous and convex setting, (26) improves the result from (Even et al., 2024) since (26) is an accelerated rate and does not depend on a quantity that measures the similarity of the functions $f_i$. When $\sigma = 0$, the result (26) is consistent with the lower bound from (Scaman et al., 2017). However, when $\sigma > 0$, as far as we know, the time complexity (26) is new in the convex smooth setting.

# E Lower Bound: Diving Deeper into the Construction

---

**Protocol 8** Time Multiple Oracles Protocol with Communication

---

1: **Input:** function $f$ (or functions $f_i$) computation oracles $\{O_i\}_{i=1}^n \in \mathcal{O}(f)$ communication oracles $\{C_{i \to j}^p\}_{i \in [n], j \in [n], p \geq 1}$, algorithm $\{(M^k, L^k, D^k, P_1^k, \ldots, P_n^k, V_1^k, \ldots, V_n^k)\}_{k=0}^\infty \in \mathcal{A}$
2: $s_i^{h,0} = s_{i,j,p}^{\tau,0} = 0$ for all $i, j \in [n], p \in \mathbb{N}$
3: **for** $k = 0, \ldots, \infty$ **do**
4:     $(t^{k+1}, c^{k+1}) = M^k(g_1^1, \ldots g_n^1, g_1^2, \ldots, g_n^2, \ldots, g_1^k, \ldots, g_n^k)$         $\triangleright t^{k+1} \geq t^k$
5:     **if** $c^{k+1} = 0$ **then**
6:         $i^{k+1} = L^k(g_1^1, \ldots g_n^1, g_1^2, \ldots, g_n^2, \ldots, g_1^k, \ldots, g_n^k)$
7:         $x_{i^{k+1}}^k = P_{i^{k+1}}^k(g_{i^{k+1}}^1, \ldots, g_{i^{k+1}}^k)$
8:         $(s_{i^{k+1}}^{h,k+1}, g_{i^{k+1}}^{k+1}) = O_{i^{k+1}}(t^{k+1}, x_{i^{k+1}}^k, s_{i^{k+1}}^{h,k})$
        $\triangleright \forall j \neq i^{k+1}: s_j^{h,k+1} = s_j^{h,k}, \quad g_j^{k+1} = 0$
9:     **else**
10:         $i^{k+1}, j^{k+1}, p^{k+1} = D^k(g_1^1, \ldots g_n^1, g_1^2, \ldots, g_n^2, \ldots, g_1^k, \ldots, g_n^k)$
11:         $v_{i^{k+1}}^k = V_{i^{k+1}}^k(g_{i^{k+1}}^1, \ldots, g_{i^{k+1}}^k)$
12:         $(s_{i^{k+1}, j^{k+1}, p^{k+1}}^{\tau, k+1}, g_{j^{k+1}}^{k+1}) = C_{i^{k+1} \to j^{k+1}}^{p^{k+1}}(t^{k+1}, v_{i^{k+1}}^k, s_{i^{k+1}, j^{k+1}, p^{k+1}}^{\tau, k})$
        $\triangleright \forall j \neq j^{k+1}: g_j^{k+1} = 0, \forall i \neq i^{k+1}, j \neq j^{k+1}, p \neq p^{k+1}: s_{i,j,p}^{\tau, k+1} = s_{i,j,p}^{\tau,k},$
13:     **end if**
14: **end for**

---

In Section 4, we present a brief overview and simplified theorem for the lower bound. We now provide a formal and strict mathematical construction.

## E.1 Description of Protocols 8 and 1

One way how we can formalize Protocol 1 is to use Protocol 8. Let us explain it. Using Protocol 8, we consider any possible method that works in our distributed asynchronous setting. The mapping $M^k$ of the algorithm returns the time $t^{k+1}$ (ignore for now) and $c^{k+1}$. If $c^{k+1} = 0$, then the algorithm decides to start the calculation of a stochastic gradient: it determines the index $i^{k+1}$ of a worker that will start the calculation, then, using all locally available information $g_{i^{k+1}}^1, \ldots, g_{i^{k+1}}^k$, it calculates a new point $x_{i^{k+1}}^k$, and passes this point to the computation oracle $O_{i^{k+1}}$ that will return a new stochastic gradient after $h_{i^{k+1}}$ seconds. If $c^{k+1} = 1$, then the algorithm decides to communicate a vector: it returns the indices $i^{k+1}$ and $j^{k+1}$ of two workers that will communicate, and the index $p^{k+1} \in \mathbb{N}$ of a communication oracle, calculates $v_{i^{k+1}}^k$ in worker $i^{k+1}$ using only all locally available information $g_{i^{k+1}}^1, \ldots, g_{i^{k+1}}^k$, and passes it to the communication oracle $C_{i^{k+1} \to j^{k+1}}^{p^{k+1}}$ that will send the vector after $\tau_{i^{k+1} \to j^{k+1}}$ seconds.

The computation oracles we define as

$$O_i : \underbrace{\mathbb{R}_{\geq 0}}_{\text{time}} \times \underbrace{\mathbb{R}^d}_{\text{point}} \times \underbrace{(\mathbb{R}_{\geq 0} \times \mathbb{R}^d \times \{0, 1\})}_{\text{input state}} \to \underbrace{(\mathbb{R}_{\geq 0} \times \mathbb{R}^d \times \{0, 1\})}_{\text{output state}} \times \mathbb{R}^d$$

such that
$$O_i(t, x, (s_t, s_x, s_q)) = \begin{cases} ((t, x, 1), & 0), & s_q = 0, \\ ((s_t, s_x, 1), & 0), & s_q = 1, t < s_t + h_i, \\ ((0, 0, 0), & \nabla f(s_x; \xi)), & s_q = 1, t \geq s_t + h_i, \end{cases} \quad (27)$$

where $\xi \sim \mathcal{D}$.

The communication oracles we define as

$$C_{i \to j}^p : \underbrace{\mathbb{R}_{\geq 0}}_{\text{time}} \times \underbrace{\mathbb{R}^d}_{\text{point}} \times \underbrace{(\mathbb{R}_{\geq 0} \times \mathbb{R}^d \times \{0, 1\})}_{\text{input state}} \to \underbrace{(\mathbb{R}_{\geq 0} \times \mathbb{R}^d \times \{0, 1\})}_{\text{output state}} \times \mathbb{R}^d$$

such that
$$C_{i \to j}^p (t, x, (s_t, s_x, s_q)) = \begin{cases} ((t, x, 1), & 0), & s_q = 0, \\ ((s_t, s_x, 1), & 0), & s_q = 1, t < s_t + \tau_{i \to j}, \\ ((0, 0, 0), & s_x), & s_q = 1, t \geq s_t + \tau_{i \to j}. \end{cases} \tag{28}$$

The idea is that the computation oracle (27) emulates the behavior of a real worker $i$ that requires $h_i$ seconds to calculate a stochastic gradient. The communication oracle emulates the behavior of a real communication channel that requires $\tau_{i \to j}$ seconds to send a vector from worker $i$ to worker $j$.

One can see that, for all $i \neq j \in [n]$, an algorithm can access an infinite number of communication oracles $C_{i \to j}^1, C_{i \to j}^2, \dots$. We allow an algorithm to send as many vectors from worker $i$ to worker $j$ in parallel as it wants.

We now discuss the role of $t^{k+1}$. One can see that the time $t^{k+1}$ is returned by the algorithm, and $t^{k+1}$ is passed to the oracles $O_{i^{k+1}}$ and $C_{i^{k+1} \to j^{k+1}}^{p^{k+1}}$. Consider that $O_{i^{k+1}}$ was called with $t^{k+1}$ for the first time, then it will return zero vector because $(s_{i^{k+1}}^{h,k})_q = 0$ (see (27)) at the beginning. Then, by the construction of (27), the oracle will return a non-zero vector in the second output if only the algorithm returns a time that is greater or equal to $t^{k+1} + h_{i^{k+1}}$. The same idea applies to $C_{i^{k+1} \to j^{k+1}}^{p^{k+1}}$ : if it was called $t^{k+1}$ for the first time, then worker $j^{k+1}$ will get a non-zero vector only if the algorithm passes a time that is greater or equal to $t^{k+1} + \tau_{i^{k+1} \to j^{k+1}}$.

The oracles force the algorithm to increase the time $t^{k+1}$; otherwise, it will not get new information ($\nabla f(s_x; \xi)$ in (27)) about the function. One crucial constraint is $t^{k+1} \geq t^k$, meaning that the algorithm can not "cheat" and travel into the past. The idea of such a protocol was proposed in Tyurin and Richtárik (2023), and we refer to Sections 3-5 for a detailed description. A more readable and less formal version of Protocol 8 is presented in the Protocol 1.

### E.2 Lower bound

Before we state the main theorem, let us define the class of zero-respecting algorithms.

**Definition 18** (Algorithm Class $\mathcal{A}_{zr}$). Let us consider Protocol 8. We say that the sequence of tuples of mappings $\{(M^k, L^k, D^k, P_1^k, \dots, P_n^k, V_1^k, \dots, V_n^k)\}_{k=0}^\infty$ is a zero-respecting algorithm, if

1. $M^k : \underbrace{\mathbb{R}^d \times \cdots \times \mathbb{R}^d}_{n \times k \text{ times}} \to \mathbb{R}_{\geq 0} \times \{0, 1\}$ for all $k \geq 1$, and $M^0 \in \mathbb{R}_{\geq 0} \times \{0, 1\}$.

2. $L^k : \underbrace{\mathbb{R}^d \times \cdots \times \mathbb{R}^d}_{n \times k \text{ times}} \to [n]$ for all $k \geq 1$, and $L^0 \in [n]$.

3. $D^k : \underbrace{\mathbb{R}^d \times \cdots \times \mathbb{R}^d}_{n \times k \text{ times}} \to [n] \times [n] \times \mathbb{N}$ for all $k \geq 1$, and $D^0 \in [n] \times [n] \times \mathbb{N}$.

4. For all $i \in [n]$, $P_i^k : \underbrace{\mathbb{R}^d \times \cdots \times \mathbb{R}^d}_{k \text{ times}} \to \mathbb{R}^d$ for all $k \geq 1$, and $P_i^0 \in \mathbb{R}^d$.

5. For all $i \in [n]$, $V_i^k : \underbrace{\mathbb{R}^d \times \cdots \times \mathbb{R}^d}_{k \text{ times}} \to \mathbb{R}^d$ for all $k \geq 1$, and $V_i^0 \in \mathbb{R}^d$.

6. For all $k \geq 1$ and $g_1^1, \dots g_n^1, g_1^2, \dots, g_n^2, \dots, g_1^k, \dots, g_n^k \in \mathbb{R}^d$,
$$t^{k+1} \geq t^k,$$
where $t^{k+1}$ and $t^k$ are defined as $(t^{k+1}, \dots) = M^k(g_1^1, \dots g_n^1, g_1^2, \dots, g_n^2, \dots, g_1^k, \dots, g_n^k)$ and $(t^k, \dots) = M^{k-1}(g_1^1, \dots g_n^1, g_1^2, \dots, g_n^2, \dots, g_1^{k-1}, \dots, g_n^{k-1})$.

7. For all $i \in [n]$, $\text{supp}\left(x_i^k\right) \subseteq \bigcup_{j=1}^k \text{supp}\left(g_i^j\right)$, and $\text{supp}\left(v_i^k\right) \subseteq \bigcup_{j=1}^k \text{supp}\left(g_i^j\right)$, for all $k \in \mathbb{N}_0$, where $\text{supp}(x) := \{i \in [d] \mid x_i \neq 0\}$.

The set of all algorithms with this properties we define as $\mathcal{A}_{zr}$.

Constraints 1–5 define domains of the mappings. Constraint 6 is required to ensure that the time sequence $t^k$ does not decrease. Constraint 7 is a standard assumption that an algorithm is zero-respecting (Arjevani et al., 2022).

**Theorem 19.** *Consider Protocol 8. We take any $h_i \geq 0$ and $\tau_{i\to j} \geq 0$ for all $i, j \in [n]$ such that $\tau_{i\to j} \leq \tau_{i\to k} + \tau_{k\to j}$ for all $i, k, j \in [n]$. We fix $L, \Delta, \varepsilon, \sigma^2 > 0$ that satisfy the inequality $\varepsilon < c'L\Delta$. For any algorithm $A \in \mathcal{A}_{zr}$, there exists a function $f$, which satisfy Assumptions 1, 2 and $f(0) - f^* \leq \Delta$, and a stochastic gradient mapping $\nabla f(\cdot; \cdot)$, which satisfy Assumption 3, such that*

$$\mathbb{E}\left[\inf_{k \in S_t, i \in [n]} \left\|\nabla f(x_i^k)\right\|^2\right] > \varepsilon, \text{ where } S_t := \left\{k \in \mathbb{N}_0 \,|\, t^k \leq t\right\},$$

$$t = c \times \frac{1}{\log n + 1} \frac{L\Delta}{\varepsilon} \min_{j \in [n]} \min_{k \in [n]} \max\left\{\max\{\tau_{\pi_{j,k}\to j}, h_{\pi_{j,k}}\}, \frac{\sigma^2}{\varepsilon}\left(\sum_{i=1}^{k} \frac{1}{h_{\pi_{j,i}}}\right)^{-1}\right\},$$

*where $\pi_{j,\cdot}$ is a permutation that sorts $\max\{h_i, \tau_{i\to j}\}$, i.e.,*

$$\max\{h_{\pi_{j,1}}, \tau_{\pi_{j,1}\to j}\} \leq \cdots \leq \max\{h_{\pi_{j,n}}, \tau_{\pi_{j,n}\to j}\}$$

*for all $j \in [n]$. The quantities $c'$ and $c$ are universal constants. The sequences $x^k$ and $t^k$ are defined in Protocol 8.*

### E.3 Proof sketch of Theorem 19

Let us provide a proof sketch that will give intuition behind the theorem. The full proof starts in Section E.4.

*Proof Sketch.*

#### Part 1: Construct a function and stochastic gradient

The first part of the proof is standard (Carmon et al., 2020; Arjevani et al., 2022; Tyurin and Richtárik, 2023; Huang et al., 2022; Lu and De Sa, 2021), and we delegate it to Section E.4. For any algorithm, we construct oracles and a "worst-case" function such that

$$\inf_{k \in S_t, i \in [n]} \left\|\nabla f(x_i^k)\right\|^2 > 2\varepsilon \inf_{k \in S_t, i \in [n]} \mathbb{1}\left[\text{prog}(x_i^k) < T\right], \tag{29}$$

where $\text{prog}(x) := \max\{i \geq 0 \,|\, x_i \neq 0\}$ $(x_0 \equiv 1)$ and $T \approx {}^{L\Delta}\!/\varepsilon$. This inequality says that while all points in Protocol 8 have the last coordinate equals 0 by the time $t$, an algorithm can not find an $\varepsilon$–stationary point.

Since we assume that $A \in \mathcal{A}_{zr}$ is zero-respecting, the only way to discover the next non-zero coordinate is through stochastic gradients. They are constructed in the following way (Arjevani et al., 2022):

$$[\nabla f(x; \xi)]_j := \nabla_j f(x)\left(1 + \mathbb{1}[j > \text{prog}(x)]\left(\frac{\xi}{p} - 1\right)\right) \quad \forall x \in \mathbb{R}^T,$$

and $\xi \sim \text{Bernoulli}(p)$ for all $i \in [n]$, where $p \approx \varepsilon/\sigma^2$. We denote $[x]_j$ as the $j^{\text{th}}$ index of a vector $x \in \mathbb{R}^T$. The stochastic gradient equals to the exact gradient except for the last non-zero coordinate: it zeros out it with the high probability $1 - p$.

#### Part 2: The Level Game

In essence, the protocol is equivalent to the following collaborative game. Each worker $i$ starts with level $\ell_i = 0$. The goal is to reach level $T$ with at least one worker as fast as possible. There are two ways how a worker can increase its level: i) worker $i$ flips one coin per time from $\xi \sim \text{Bernoulli}(p)$, it takes $h_i$ seconds to flip one coin, and if the worker is lucky, i.e., $\xi = 1$, then it moves to the next level $\ell_i = \ell_i + 1$; ii) worker $i$ can share its level with another worker $j$, and it takes $\tau_{i\to j}$ seconds (we are allowed to run this operation again even if the previous is not finished). Both options can be executed in parallel. What is the minimum possible time to reach the game's goal?

Since all workers have the levels equal to $0$ at the beginning, they should flip coins in parallel and wait for the moment when at least one worker moves to level 1. We define $\eta_i^1$ as the number of flips in worker $i$ to get $\xi = 1$. Clearly, $\eta_i^1 \sim \text{Geometric}(p)$ are i.i.d. geometric random variables with the probability $p$. With any strategy, it is *necessary* to wait at least

$$y_j^1 := \min_{i \in [n]} \left\{ h_i \eta_i^1 + \tau_{i \to j} \right\}$$

seconds to reach level 1 in worker $j$ because once worker $i$ flips $\xi = 1$, it will take at least $\tau_{i \to j}$ seconds to share level 1 to worker $j$ due to the triangle inequality $\tau_{i \to j} \leq \tau_{i \to k} + \tau_{k \to j}$ for all $i, j, k \in [n]$, and we should minimize $h_i \eta_i^1 + \tau_{i \to j}$ over all workers. We now use mathematical induction to prove that it is necessary to wait at least

$$y_j^T := \min_{i \in [n]} \left\{ y_i^{T-1} + h_i \eta_i^T + \tau_{i \to j} \right\}$$

seconds to reach level $T$, where we define $y_i^0 := 0$ for all $i \in [n]$ and $\{\eta_i^T\}$ are i.i.d. random variables from Geometric$(p)$. The base case is proven above. Assume that it is true for $1, \ldots, T-1$, then worker $i$ requires at least $y_i^{T-1} + h_i \eta_i^T$ seconds to flip a coin that moves to level $T$, it takes at least $\tau_{i \to j}$ seconds to share the level $T$, so it is necessary to wait at least $\min_{i \in [n]} \left\{ y_i^{T-1} + h_i \eta_i^T + \tau_{i \to j} \right\}$ seconds in worker $i$ to get level $T$. Ultimately, the minimum possible time to reach the game's goal is

$$y^T := \min_{j \in [n]} y_j^T.$$

From the previous result, we can conclude that if $t \leq \frac{1}{2} y^T$, then

$$\inf_{k \in S_t, i \in [n]} \left\| \nabla f(x_i^k) \right\|^2 > 2\varepsilon \tag{30}$$

for any algorithm $A \in \mathcal{A}_{\text{zr}}$.

## Part 3: The high probability bound of $y^T$

Let us fix any determenistic value $\bar{y} \in \mathbb{R}$ and take

$$t = \frac{1}{2} \bar{y}. \tag{31}$$

Using (30), we have

$$\mathbb{E} \left[ \inf_{k \in S_t, i \in [n]} \left\| \nabla f(x_i^k) \right\|^2 \right] \geq \mathbb{E} \left[ \inf_{k \in S_t, i \in [n]} \left\| \nabla f(x_i^k) \right\|^2 \middle| y^T > \bar{y} \right] \mathbb{P} \left( y^T > \bar{y} \right) > \mathbb{P} \left( y^T > \bar{y} \right) 2\varepsilon.$$

Thus, it is sufficient to find any $\bar{y} \in \mathbb{R}$ such that

$$\mathbb{P} \left( y^T \leq \bar{y} \right) \leq \frac{1}{2}. \tag{32}$$

The sequence $y^T$ is a well-define time series. In Lemma 2, we show that (32) holds with

$$\bar{y} = \Theta \left( \frac{1}{\log n + 1} \frac{L\Delta}{\varepsilon} \min_{j \in [n]} \min_{k \in [n]} \max \left\{ \max\{\tau_{\pi_{j,k} \to j}, h_{\pi_{j,k}}\}, \frac{\sigma^2}{\varepsilon} \left( \sum_{i=1}^{k} \frac{1}{h_{\pi_{j,i}}} \right)^{-1} \right\} \right),$$

$\pi_{j,\cdot}$ is a permutation that sorts $\max\{h_i, \tau_{i \to j}\}$, i.e.,

$$\max\{h_{\pi_{j,1}}, \tau_{\pi_{j,1} \to j}\} \leq \cdots \leq \max\{h_{\pi_{j,n}}, \tau_{\pi_{j,n} \to j}\}$$

for all $j \in [n]$. We substitute this $\bar{y}$ to (31) and get the result of theorem. $\qquad \square$

### E.4 Full proof of Theorem 19

This section considers the standard "worst case" function that helps to provide lower bounds in the nonconvex world. Let us define

$$\text{prog}(x) := \max\{i \geq 0 \,|\, x_i \neq 0\} \quad (x_0 \equiv 1).$$

For any $T \in \mathbb{N}$, Carmon et al. (2020); Arjevani et al. (2022) define

$$F_T(x) := -\Psi(1)\Phi(x_1) + \sum_{i=2}^{T} \left[ \Psi(-x_{i-1})\Phi(-x_i) - \Psi(x_{i-1})\Phi(x_i) \right], \tag{33}$$

where

$$\Psi(x) = \begin{cases} 0, & x \leq 1/2, \\ \exp\left(1 - \frac{1}{(2x-1)^2}\right), & x \geq 1/2, \end{cases} \quad \text{and} \quad \Phi(x) = \sqrt{e} \int_{-\infty}^{x} e^{-\frac{1}{2}t^2} dt.$$

We will only rely on the following facts.

**Lemma 1** (Carmon et al. (2020); Arjevani et al. (2022)). *The function $F_T$ satisfies:*

1. $F_T(0) - \inf_{x \in \mathbb{R}^T} F_T(x) \leq \Delta^0 T$, *where* $\Delta^0 = 12$.

2. *The function $F_T$ is $l_1$–smooth, where* $l_1 = 152$.

3. *For all $x \in \mathbb{R}^T$, $\|\nabla F_T(x)\|_\infty \leq \gamma_\infty$, where* $\gamma_\infty = 23$.

4. *For all $x \in \mathbb{R}^T$, $\text{prog}(\nabla F_T(x)) \leq \text{prog}(x) + 1$.*

5. *For all $x \in \mathbb{R}^T$, if $\text{prog}(x) < T$, then $\|\nabla F_T(x)\| > 1$.*

**Theorem 19.** *Consider Protocol 8. We take any $h_i \geq 0$ and $\tau_{i \to j} \geq 0$ for all $i, j \in [n]$ such that $\tau_{i \to j} \leq \tau_{i \to k} + \tau_{k \to j}$ for all $i, k, j \in [n]$. We fix $L, \Delta, \varepsilon, \sigma^2 > 0$ that satisfy the inequality $\varepsilon < c'L\Delta$. For any algorithm $A \in \mathcal{A}_{zr}$, there exists a function $f$, which satisfy Assumptions 1, 2 and $f(0) - f^* \leq \Delta$, and a stochastic gradient mapping $\nabla f(\cdot; \cdot)$, which satisfy Assumption 3, such that*

$$\mathbb{E}\left[ \inf_{k \in S_t, i \in [n]} \left\| \nabla f(x_i^k) \right\|^2 \right] > \varepsilon, \text{ where } S_t := \left\{ k \in \mathbb{N}_0 \,|\, t^k \leq t \right\},$$

$$t = c \times \frac{1}{\log n + 1} \frac{L\Delta}{\varepsilon} \min_{j \in [n]} \min_{k \in [n]} \max \left\{ \max\{\tau_{\pi_{j,k} \to j}, h_{\pi_{j,k}}\}, \frac{\sigma^2}{\varepsilon} \left( \sum_{i=1}^{k} \frac{1}{h_{\pi_{j,i}}} \right)^{-1} \right\},$$

*where $\pi_{j,\cdot}$ is a permutation that sorts $\max\{h_i, \tau_{i \to j}\}$, i.e.,*

$$\max\{h_{\pi_{j,1}}, \tau_{\pi_{j,1} \to j}\} \leq \cdots \leq \max\{h_{\pi_{j,n}}, \tau_{\pi_{j,n} \to j}\}$$

*for all $j \in [n]$. The quantities $c'$ and $c$ are universal constants. The sequences $x^k$ and $t^k$ are defined in Protocol 8.*

*Proof.*
**Part 1: The Worst Case Function**
This part of the proof mirrors the proofs from Carmon et al. (2020); Arjevani et al. (2022); Tyurin and Richtárik (2023); Huang et al. (2022); Lu and De Sa (2021). We provide it for completeness. The goal of this part to construct a hard instance.

We fix $\lambda > 0$, $T \in \mathbb{N}$ and take the function $f(x) := \frac{L\lambda^2}{l_1} F_T\left(\frac{x}{\lambda}\right)$, where the function $F_T$ is defined in Section E.4. Note that the function $f$ is $L$–smooth:

$$\|\nabla f(x) - \nabla f(y)\| = \frac{L\lambda}{l_1} \left\| \nabla F_T\left(\frac{x}{\lambda}\right) - \nabla F_T\left(\frac{y}{\lambda}\right) \right\| \leq L\lambda \left\| \frac{x}{\lambda} - \frac{y}{\lambda} \right\| = L \|x - y\| \quad \forall x, y \in \mathbb{R}^d,$$

where $l_1$–smoothness of $F_T$ (Lemma 1). Let us take

$$T = \left\lfloor \frac{\Delta l_1}{L\lambda^2 \Delta^0} \right\rfloor,$$

then

$$f(0) - \inf_{x \in \mathbb{R}^T} f(x) = \frac{L\lambda^2}{l_1}(F_T(0) - \inf_{x \in \mathbb{R}^T} F_T(x)) \le \frac{L\lambda^2 \Delta^0 T}{l_1} \le \Delta.$$

We showed that the function $f$ satisfy Assumptions 1, 2 and $f(0) - f^* \le \Delta$.

For each worker $i$ we take an oracle $O_i$, from (27) with the mapping $g$ such that

$$[\nabla f(x; \xi)]_j := \nabla_j f(x) \left(1 + \mathbb{1}\left[j > \text{prog}(x)\right] \left(\frac{\xi}{p} - 1\right)\right) \quad \forall x \in \mathbb{R}^T,$$

and $\mathcal{D}_i = \text{Bernouilli}(p)$ for all $i \in [n]$, where $p \in (0, 1]$. We denote $[x]_j$ as the $j^{\text{th}}$ index of a vector $x \in \mathbb{R}^T$. This stochastic gradient is unbiased and $\sigma^2$-variance-bounded. We have

$$\mathbb{E}\left[[\nabla f(x, \xi)]_i\right] = \nabla_i f(x) \left(1 + \mathbb{1}\left[i > \text{prog}(x)\right] \left(\frac{\mathbb{E}[\xi]}{p} - 1\right)\right) = \nabla_i f(x)$$

for all $i \in [T]$, and

$$\mathbb{E}\left[\|\nabla f(x; \xi) - \nabla f(x)\|^2\right] \le \max_{j \in [T]} |\nabla_j f(x)|^2 \, \mathbb{E}\left[\left(\frac{\xi}{p} - 1\right)^2\right]$$

because the difference is non-zero only in one coordinate. Thus

$$\mathbb{E}\left[\|\nabla f(x, \xi) - \nabla f(x)\|^2\right] \le \frac{\|\nabla f(x)\|_\infty^2 (1-p)}{p} = \frac{L^2\lambda^2 \left\|\nabla F_T\left(\frac{x}{\lambda}\right)\right\|_\infty^2 (1-p)}{l_1^2 p}$$

$$\le \frac{L^2 \lambda^2 \gamma_\infty^2 (1-p)}{l_1^2 p} \le \sigma^2,$$

where we use Lemma 1 and take

$$p = \min\left\{\frac{L^2 \lambda^2 \gamma_\infty^2}{\sigma^2 l_1^2}, 1\right\}.$$

Let us take

$$\lambda = \frac{\sqrt{2\varepsilon} l_1}{L}$$

to ensure that

$$\|\nabla f(x)\|^2 = \frac{L^2 \lambda^2}{l_1^2} \left\|\nabla F_T\left(\frac{x}{\lambda}\right)\right\|^2 = 2\varepsilon \left\|\nabla F_T\left(\frac{x}{\lambda}\right)\right\|^2$$

for all $x \in \mathbb{R}^T$. From Lemma 1, we know that if $\text{prog}(x) < T$, then $\|\nabla F_T(x)\| > 1$. Thus, we get

$$\|\nabla f(x)\|^2 > 2\varepsilon \mathbb{1}\left[\text{prog}(x) < T\right] \tag{34}$$

Therefore,

$$T = \left\lfloor \frac{\Delta L}{2\varepsilon l_1 \Delta^0} \right\rfloor \tag{35}$$

and

$$p = \min\left\{\frac{2\varepsilon \gamma_\infty^2}{\sigma^2}, 1\right\}. \tag{36}$$

The inequality (34) implies

$$\inf_{k \in S_t, i \in [n]} \left\|\nabla f(x_i^k)\right\|^2 > 2\varepsilon \inf_{k \in S_t, i \in [n]} \mathbb{1}\left[\text{prog}(x_i^k) < T\right], \tag{37}$$

where $\{x_i^k\}_{k=0}^\infty$ are sequences from Protocol 8.

**Part 2: Reduction to Lower Bound Time Series**

We now focus on (27). In (27), when the oracle in worker $i$ calculates a new stochastic gradient, it

samples a random variable $\xi \sim \mathcal{D}$. This is equivalent to the procedure if we had $T$ infinite sequences of i.i.d. Bernoulli random variables

$$
\begin{aligned}
&\xi_i^{1,1}, \xi_i^{1,2}, \ldots && (\text{prog}(s_x) = 0) \\
&\xi_i^{2,1}, \xi_i^{2,2}, \ldots && (\text{prog}(s_x) = 1) \\
&\ldots \\
&\xi_i^{T,1}, \xi_i^{T,2}, \ldots && (\text{prog}(s_x) = T - 1)
\end{aligned}
$$

for all $i \in [n]$ and the oracle would look at the progress of $s_x$ and take the next non-taken Bernoulli random variable in the sequence that corresponds to that progress. For instance, if $\text{prog}(s_x) = j$ for the first time in worker $i$, then the oracle will apply $\xi_i^{j,1}$ in (27). The next time when it gets $\text{prog}(s_x) = j$ it will apply $\xi_i^{j,2}$ and so on. One by one, we get i.i.d. Bernoulli random variables.

Let us define

$$
\eta_i^k = \inf\{j \in \mathbb{N} \,|\, \xi_i^{k,j} = 1\}
$$

for all $i \in [n]$ and $k \in [T]$. This is the first Bernoulli random variable from the sequence $\xi_i^{k,1}, \xi_i^{k,2}, \ldots$ that equals 1. The random variables $\{\eta_i^k\}$ are i.i.d. geometrically distributed random variables with the probability $p$.

The next steps mirror *Proof Sketch* from Theorem 19. The oracles constructed in such a way that it takes $h_i$ seconds to calculate a stochastic gradient, and at least $\tau_{i \to j}$ seconds to send a vector from one worker to another. Using the same reasoning as in the Level Game in *Proof Sketch* of Theorem 19, the first time moment when worker $j$ can get a vector with the first non-zero coordinate greater or equal

$$
y_j^1 := \min_{i \in [n]} \left\{ h_i \eta_i^1 + \tau_{i \to j} \right\}
$$

because, at the beginning, each worker $i$ calculates stochastic gradients with $\text{prog}(s_x) = 0$ and should wait at least $h_i \eta_i^1$ seconds to get a stochastic gradient with the progress equals 1. Then, it can share this vector with any other worker $j$, but it takes at least $\tau_{i \to j}$ seconds.

As in *Proof Sketch* of Theorem 19, we can use mathematical induction to prove that it is necessary to wait at least

$$
y_j^T := \min_{i \in [n]} \left\{ y_i^{T-1} + h_i \eta_i^T + \tau_{i \to j} \right\}
$$

seconds to get a point such that $\text{prog}(s_x) = T$. The base case for $y_j^1$ has been proven. Worker $i$ requires at least $y_i^{T-1} + h_i \eta_i^T$ seconds to wait for the moment when the corresponding oracle will return a stochastic gradient with progress $T$ because $y_i^{T-1}$ is the first time possible time to get a vector with progress $T - 1$ by the induction, and it will take at least additional $h_i \eta_i^T$ seconds to calculate $\eta_i^T$ vectors with $\text{prog}(s_x) = T - 1$ in (27). Also, it takes at least $\tau_{i \to j}$ seconds to share a vector, so it is necessary to wait at least $\min_{i \in [n]} \left\{ y_i^{T-1} + h_i \eta_i^T + \tau_{i \to j} \right\}$ seconds in worker $i$ to get the first vector with progress $T$.

In the end, the fastest possible time to get a vector with progress $T$ is at least

$$
y^T := \min_{j \in [n]} y_j^T.
$$

**Part 3: Reduction to the concentration of $y^T$**
The last statement means that $\text{prog}(x_i^k) < T$ for all $i \in [n]$ and $k$ such that $t^k \leq \frac{1}{2} y^T$. Therefore, we obtain

$$
\inf_{k \in S_t, i \in [n]} \left\| \nabla f(x_i^k) \right\|^2 > 2\varepsilon \tag{38}
$$

for

$$
t \leq \frac{1}{2} y^T,
$$

where

$$
T = \left\lfloor \frac{\Delta L}{2 \varepsilon l_1 \Delta^0} \right\rfloor = \left\lfloor c_T \cdot \frac{\Delta L}{\varepsilon} \right\rfloor
$$

and $\eta_i^j \sim \text{Geometric}(p)$ with

$$p = \min\left\{\frac{2\varepsilon\gamma_\infty^2}{\sigma^2}, 1\right\} = \min\left\{c_p \cdot \frac{\varepsilon}{\sigma^2}, 1\right\},$$

where $c_T = 3648^{-1}$ and $c_p = 1058$ are universal constants. Let us fix any determenistic value $\bar{y} \in \mathbb{R}$ and take

$$t = \frac{1}{2}\bar{y}. \tag{39}$$

Using (38), we have

$$\mathbb{E}\left[\inf_{k \in S_t, i \in [n]} \left\|\nabla f(x_i^k)\right\|^2\right] \geq \mathbb{E}\left[\inf_{k \in S_t, i \in [n]} \left\|\nabla f(x_i^k)\right\|^2 \middle| y^T > \bar{y}\right] \mathbb{P}\left(y^T > \bar{y}\right) > \mathbb{P}\left(y^T > \bar{y}\right) 2\varepsilon.$$

Thus, it is sufficient to find any $\bar{y} \in \mathbb{R}$ such that

$$\mathbb{P}\left(y^T \leq \bar{y}\right) \leq \frac{1}{2}. \tag{40}$$

In the following lemma we use the notation of this theorem. We prove it in Section E.5.

**Lemma 2.** *With*

$$\bar{y} = c \times \frac{1}{\log n + 1}\frac{L\Delta}{\varepsilon}\min_{j \in [n]}\min_{k \in [n]}\max\left\{\max\{\tau_{\pi_{j,k} \to j}, h_{\pi_{j,k}}\}, \frac{\sigma^2}{\varepsilon}\left(\sum_{i=1}^k \frac{1}{h_{\pi_{j,i}}}\right)^{-1}\right\}, \tag{41}$$

*we have* $\mathbb{P}\left(y^T \leq \bar{y}\right) \leq \frac{1}{2}$, *where $\pi_{j,\cdot}$ is a permutation that sorts $\max\{h_i, \tau_{i \to j}\}$, i.e.,*

$$\max\{h_{\pi_{j,1}}, \tau_{\pi_{j,1} \to j}\} \leq \cdots \leq \max\{h_{\pi_{j,n}}, \tau_{\pi_{j,n} \to j}\}$$

*for all $j \in [n]$. The quantity $c$ is a universal constant.*

Using Lemma 2, we can conclude that (40) holds and

$$\mathbb{E}\left[\inf_{k \in S_t, i \in [n]} \left\|\nabla f(x_i^k)\right\|^2\right] > \varepsilon$$

for

$$t = \frac{c}{2} \times \frac{1}{\log n + 1}\frac{L\Delta}{\varepsilon}\min_{j \in [n]}\min_{k \in [n]}\max\left\{\max\{\tau_{\pi_{j,k} \to j}, h_{\pi_{j,k}}\}, \frac{\sigma^2}{\varepsilon}\left(\sum_{i=1}^k \frac{1}{h_{\pi_{j,i}}}\right)^{-1}\right\}.$$

$$\square$$

### E.5   Proof of Lemma 2

**Lemma 2.** *With*

$$\bar{y} = c \times \frac{1}{\log n + 1}\frac{L\Delta}{\varepsilon}\min_{j \in [n]}\min_{k \in [n]}\max\left\{\max\{\tau_{\pi_{j,k} \to j}, h_{\pi_{j,k}}\}, \frac{\sigma^2}{\varepsilon}\left(\sum_{i=1}^k \frac{1}{h_{\pi_{j,i}}}\right)^{-1}\right\}, \tag{41}$$

*we have* $\mathbb{P}\left(y^T \leq \bar{y}\right) \leq \frac{1}{2}$, *where $\pi_{j,\cdot}$ is a permutation that sorts $\max\{h_i, \tau_{i \to j}\}$, i.e.,*

$$\max\{h_{\pi_{j,1}}, \tau_{\pi_{j,1} \to j}\} \leq \cdots \leq \max\{h_{\pi_{j,n}}, \tau_{\pi_{j,n} \to j}\}$$

*for all $j \in [n]$. The quantity $c$ is a universal constant.*

*Proof.* Using the Chernoff method for any $s > 0$, we get

$$\mathbb{P}\left(y^k \leq t\right) = \mathbb{P}\left(-sy^k \geq -st\right) = \mathbb{P}\left(e^{-sy^k} \geq e^{-st}\right) \leq e^{st}\mathbb{E}\left[e^{-sy^k}\right]$$

$$= e^{st}\mathbb{E}\left[\exp\left(-s\min_{j \in [n]} y_j^k\right)\right] = e^{st}\mathbb{E}\left[\max_{j \in [n]}\exp\left(-sy_j^k\right)\right].$$

We have a maximum operation that complicates the analysis. In response to this problem, we use a well-known trick that bounds a maximum by a sum.

$$\mathbb{P}\left(y^k \leq t\right) \leq e^{st} \sum_{j=1}^{n} \mathbb{E}\left[\exp\left(-sy_j^k\right)\right] \leq n e^{st} \max_{j \in [n]} \mathbb{E}\left[\exp\left(-sy_j^k\right)\right]. \tag{42}$$

We refer to (Van Handel, 2014)[Part II] for the explanation why it can be (almost) tight. This is the main reason why we get an extra $\log n$ factor in (41). Let us consider the last exponent separately and use the same trick again:

$$\mathbb{E}\left[\exp\left(-sy_j^k\right)\right] = \mathbb{E}\left[\exp\left(-s \min_{i \in [n]}\left\{y_i^{k-1} + h_i \eta_i^k + \tau_{i \to j}\right\}\right)\right]$$

$$= \mathbb{E}\left[\max_{i \in [n]} \exp\left(-s\left\{y_i^{k-1} + h_i \eta_i^k + \tau_{i \to j}\right\}\right)\right].$$

Next, we get

$$\mathbb{E}\left[\exp\left(-sy_j^k\right)\right] \leq \sum_{i=1}^{n} \mathbb{E}\left[\exp\left(-s\left\{y_i^{k-1} + h_i \eta_i^k + \tau_{i \to j}\right\}\right)\right]$$

$$= \sum_{i=1}^{n} \mathbb{E}\left[e^{-s\left(h_i \eta_i^k + \tau_{i \to j}\right)}\right] \mathbb{E}\left[\exp\left(-sy_i^{k-1}\right)\right].$$

In the last equality we use the independence. We now bound $\mathbb{E}\left[\exp\left(-sy_i^{k-1}\right)\right]$ by $\max_{i \in [n]} \mathbb{E}\left[\exp\left(-sy_i^{k-1}\right)\right]$ and get

$$\mathbb{E}\left[\exp\left(-sy_j^k\right)\right] \leq \sum_{i=1}^{n} \mathbb{E}\left[e^{-s\left(h_i \eta_i^k + \tau_{i \to j}\right)}\right] \mathbb{E}\left[\exp\left(-sy_i^{k-1}\right)\right]$$

$$\leq \left(\sum_{i=1}^{n} \mathbb{E}\left[e^{-s\left(h_i \eta_i^k + \tau_{i \to j}\right)}\right]\right) \max_{i \in [n]} \mathbb{E}\left[\exp\left(-sy_i^{k-1}\right)\right].$$

Let us fix any $s_j > 0$ for all $j \in [n]$ and take $s = \max_{j \in [n]} s_j$. Then

$$\mathbb{E}\left[\exp\left(-sy_j^k\right)\right] \leq \left(\sum_{i=1}^{n} \mathbb{E}\left[e^{-s_j\left(h_i \eta_i^k + \tau_{i \to j}\right)}\right]\right) \max_{i \in [n]} \mathbb{E}\left[\exp\left(-sy_i^{k-1}\right)\right]. \tag{43}$$

Let us fix $\bar{t}_j > 0$ and consider

$$a_j := \sum_{i=1}^{n} \mathbb{E}\left[e^{-s_j\left(h_i \eta_i^k + \tau_{i \to j}\right)}\right]. \tag{44}$$

Then, we can use the following inequalities:

$$a_j \leq \sum_{i=1}^{n} \mathbb{E}\left[e^{-s_j\left(h_i \eta_i^k + \tau_{i \to j}\right)} \mathbb{1}\left[h_i \eta_i^k + \tau_{i \to j} \leq \bar{t}_j\right] + e^{-s_j\left(h_i \eta_i^k + \tau_{i \to j}\right)}\left(1 - \mathbb{1}\left[h_i \eta_i^k + \tau_{i \to j} \leq \bar{t}_j\right]\right)\right]$$

$$\leq \sum_{i=1}^{n} \mathbb{E}\left[\mathbb{1}\left[h_i \eta_i^k + \tau_{i \to j} \leq \bar{t}_j\right] + e^{-s_j \bar{t}_j}\left(1 - \mathbb{1}\left[h_i \eta_i^k + \tau_{i \to j} \leq \bar{t}_j\right]\right)\right]$$

$$= n e^{-s_j \bar{t}_j} + (1 - e^{-s_j \bar{t}_j}) \sum_{i=1}^{n} \mathbb{E}\left[\mathbb{1}\left[h_i \eta_i^k + \tau_{i \to j} \leq \bar{t}_j\right]\right]$$

$$\leq n e^{-s_j \bar{t}_j} + \sum_{i=1}^{n} \mathbb{E}\left[\mathbb{1}\left[h_i \eta_i^k + \tau_{i \to j} \leq \bar{t}_j\right]\right]$$

$$\leq n e^{-s_j \bar{t}_j} + \sum_{i=1}^{n} \mathbb{P}\left(h_i \eta_i^k + \tau_{i \to j} \leq \bar{t}_j\right)$$

$$\leq n e^{-s_j \bar{t}_j} + \sum_{i=1}^{n} \mathbb{P}\left(h_i \eta_i^k \leq \bar{t}_j\right) \mathbb{1}\left[\tau_{i \to j} \leq \bar{t}_j\right].$$

Since $\eta_i^k \sim \text{Geometric}(p)$, we get

$$\mathbb{P}\left(h_i \eta_i^k \le \bar{t}_j\right) = 1 - (1-p)^{\left\lfloor \frac{\bar{t}_j}{h_i} \right\rfloor} \le p \left\lfloor \frac{\bar{t}_j}{h_i} \right\rfloor.$$

Then

$$a_j \le ne^{-s_j \bar{t}_j} + p \sum_{i=1}^{n} \left\lfloor \frac{\bar{t}_j}{h_i} \right\rfloor \mathbb{1}\left[\tau_{i \to j} \le \bar{t}_j\right].$$

For all $j \in [n]$, we take any permutation $\pi_{j,\cdot}$ that sorts $\max\{h_i, \tau_{i \to j}\}$, i.e.,

$$\max\{h_{\pi_{j,1}}, \tau_{\pi_{j,1} \to j}\} \le \cdots \le \max\{h_{\pi_{j,n}}, \tau_{\pi_{j,n} \to j}\}.$$

We have

$$a_j \le ne^{-s_j \bar{t}_j} + p \sum_{i=1}^{n} \left\lfloor \frac{\bar{t}_j}{h_{\pi_{j,i}}} \right\rfloor \mathbb{1}\left[\tau_{\pi_{j,i} \to j} \le \bar{t}_j\right]. \tag{45}$$

Recall that $\bar{t}_j > 0$ is a parameter. In the following technical lemma, we choose $\bar{t}_j$ and show that the second term in (45) is "small." We prove it in Section E.6.

**Lemma 3.** *For any $n \ge 1$, $h_i \ge 0, \tau_{i \to j} \ge 0$ for all $i, j \in [n]$, and $p \in (0, 1]$, we have*

$$p \sum_{i=1}^{n} \left\lfloor \frac{\bar{t}_j}{h_{\pi_{j,i}}} \right\rfloor \mathbb{1}\left[\tau_{\pi_{j,i} \to j} \le \bar{t}_j\right] \le \frac{1}{8}$$

*for all $j \in [n]$, where*

$$\bar{t}_j := \frac{1}{8} \min_{k \in [n]} \max \left\{ \max\{\tau_{\pi_{j,k} \to j}, h_{\pi_{j,k}}\}, \left(\sum_{i=1}^{k} \frac{p}{h_{\pi_{j,i}}}\right)^{-1} \right\}. \tag{46}$$

*and $\pi_{j,\cdot}$ is a permutation that sorts $\max\{h_i, \tau_{i \to j}\}$, i.e.,*

$$\max\{h_{\pi_{j,1}}, \tau_{\pi_{j,1} \to j}\} \le \cdots \le \max\{h_{\pi_{j,n}}, \tau_{\pi_{j,n} \to j}\}$$

*for all $j \in [n]$.*

Using Lemma 3 and (45), we get

$$a_j \le ne^{-s_j \bar{t}_j} + \frac{1}{8}$$

for all $j \in [n]$. Let us take

$$s_j = \frac{\log 8n}{\bar{t}_j} \tag{47}$$

to get

$$a_j \le \frac{1}{8} + \frac{1}{8} \le e^{-1}.$$

for all $j \in [n]$. Using (44) and (43), we obtain

$$\mathbb{E}\left[\exp\left(-sy_j^k\right)\right] \le e^{-1} \max_{i \in [n]} \mathbb{E}\left[\exp\left(-sy_i^{k-1}\right)\right], \tag{48}$$

for all $j \in [n]$. We can conclude that

$$\max_{i \in [n]} \mathbb{E}\left[\exp\left(-sy_i^k\right)\right] \le e^{-1} \max_{i \in [n]} \mathbb{E}\left[\exp\left(-sy_i^{k-1}\right)\right] \le e^{-k},$$

where use the same reasoning in the recursion and $y_j^0 = 0$ for all $j \in [n]$. We substitute the inequality to (42):

$$\mathbb{P}\left(y^k \le t\right) \le ne^{st-k} = e^{st-k+\log n}$$

It is sufficient to take $k = T$ and any

$$t \leq \frac{1}{s}\left(T - \log n + \log \frac{1}{2}\right)$$

$$= \frac{1}{8 \log 8n} \min_{i \in [n]} \min_{k \in [n]} \max\left\{\max\{\tau_{\pi_{j,k} \to j}, h_{\pi_{j,k}}\}, \left(\sum_{i=1}^{k} \frac{p}{h_{\pi_{j,i}}}\right)^{-1}\right\}\left(T - \log n + \log \frac{1}{2}\right)$$

to get

$$\mathbb{P}\left(y^T \leq t\right) \leq \frac{1}{2},$$

where we use the definitions $s = \max_{i \in [n]} s_i$, (46) and (47). Recall the choice of $T$ and $p$ in (35) and (36): $T = \left\lfloor c_T \cdot \frac{\Delta L}{\varepsilon} \right\rfloor$ and $p = \min\left\{c_p \cdot \frac{\varepsilon}{\sigma^2}, 1\right\}$ for some universal constants $c_T$ and $c_p$. Since we have the condition $\varepsilon < c'L\Delta$ for some universal constant $c'$ in the conditions of Theorem 19, we can conclude that we can take

$$t = c \times \frac{1}{\log n + 1}\frac{L\Delta}{\varepsilon} \min_{i \in [n]} \min_{k \in [n]} \max\left\{\max\{\tau_{\pi_{j,k} \to j}, h_{\pi_{j,k}}\}, \frac{\sigma^2}{\varepsilon}\left(\sum_{i=1}^{k} \frac{1}{h_{\pi_{j,i}}}\right)^{-1}\right\},$$

where $c$ is a universal constant. $\qquad\square$

### E.6   Proof of Lemma 3

**Lemma 3.** *For any $n \geq 1$, $h_i \geq 0, \tau_{i \to j} \geq 0$ for all $i, j \in [n]$, and $p \in (0, 1]$, we have*

$$p\sum_{i=1}^{n}\left\lfloor\frac{\bar{t}_j}{h_{\pi_{j,i}}}\right\rfloor \mathbb{1}\left[\tau_{\pi_{j,i} \to j} \leq \bar{t}_j\right] \leq \frac{1}{8}$$

*for all $j \in [n]$, where*

$$\bar{t}_j := \frac{1}{8}\min_{k \in [n]} \max\left\{\max\{\tau_{\pi_{j,k} \to j}, h_{\pi_{j,k}}\}, \left(\sum_{i=1}^{k} \frac{p}{h_{\pi_{j,i}}}\right)^{-1}\right\}. \tag{46}$$

*and $\pi_{j,\cdot}$ is a permutation that sorts $\max\{h_i, \tau_{i \to j}\}$, i.e.,*

$$\max\{h_{\pi_{j,1}}, \tau_{\pi_{j,1} \to j}\} \leq \cdots \leq \max\{h_{\pi_{j,n}}, \tau_{\pi_{j,n} \to j}\}$$

*for all $j \in [n]$.*

Let us define $k^* \in [n]$ as the largest index that minimizes (46).
**(Part 1): bound $\bar{t}_j$ by $\max\{\tau_{\pi_{j,k^*+1} \to j}, h_{\pi_{j,k^*+1}}\}$**
Let us consider the case $k^* < n$. We have two options:

1. Let $\max\{\tau_{\pi_{j,k^*} \to j}, h_{\pi_{j,k^*}}\} \geq \left(\sum_{i=1}^{k^*} \frac{p}{h_{\pi_{j,i}}}\right)^{-1}$, then

$$\max\{\tau_{\pi_{j,k^*+1} \to j}, h_{\pi_{j,k^*+1}}\} \geq \left(\sum_{i=1}^{k^*+1} \frac{p}{h_{\pi_{j,i}}}\right)^{-1}, \tag{49}$$

since $\max\{\tau_{\pi_{j,i} \to j}, h_{\pi_{j,i}}\}$ are sorted. Then, we get

$$\bar{t}_j < \frac{1}{8}\max\left\{\max\{\tau_{\pi_{j,k^*+1} \to j}, h_{\pi_{j,k^*+1}}\}, \left(\sum_{i=1}^{k^*+1} \frac{p}{h_{\pi_{j,i}}}\right)^{-1}\right\}$$

$$\overset{(49)}{=} \frac{1}{8}\max\{\tau_{\pi_{j,k^*+1} \to j}, h_{\pi_{j,k^*+1}}\} \leq \max\{\tau_{\pi_{j,k^*+1} \to j}, h_{\pi_{j,k^*+1}}\}$$

The first inequality follows from the fact that $k^*$ is the largest minimizer.

2. Let $\max\{\tau_{\pi_{j,k^*}\to j}, h_{\pi_{j,k^*}}\} < \left(\sum_{i=1}^{k^*}\frac{p}{h_{\pi_{j,i}}}\right)^{-1}$, then it is not possible that

$\max\{\tau_{\pi_{j,k^*+1}\to j}, h_{\pi_{j,k^*+1}}\} < \left(\sum_{i=1}^{k^*+1}\frac{p}{h_{\pi_{j,i}}}\right)^{-1}$ because it would yield the inequality

$$\bar{t}_j = \frac{1}{8}\left(\sum_{i=1}^{k^*}\frac{p}{h_{\pi_{j,i}}}\right)^{-1} \geq \frac{1}{8}\left(\sum_{i=1}^{k^*+1}\frac{p}{h_{\pi_{j,i}}}\right)^{-1} = \frac{1}{8}\max\left\{\max\{\tau_{\pi_{j,k^*+1}\to j}, h_{\pi_{j,k^*+1}}\}, \left(\sum_{i=1}^{k^*+1}\frac{p}{h_{\pi_{j,i}}}\right)^{-1}\right\}.$$

The last inequality contradicts the fact that $k^*$ is the largest minimizer. Thus, if $k^* < n$ and $\max\{\tau_{\pi_{j,k^*}\to j}, h_{\pi_{j,k^*}}\} < \left(\sum_{i=1}^{k^*}\frac{p}{h_{\pi_{j,i}}}\right)^{-1}$, then

$$\bar{t}_j < \frac{1}{8}\max\left\{\max\{\tau_{\pi_{j,k^*+1}\to j}, h_{\pi_{j,k^*+1}}\}, \left(\sum_{i=1}^{k^*+1}\frac{p}{h_{\pi_{j,i}}}\right)^{-1}\right\} = \frac{1}{8}\max\{\tau_{\pi_{j,k^*+1}\to j}, h_{\pi_{j,k^*+1}}\}.$$

In total, we have

$$\bar{t}_j < \max\{\tau_{\pi_{j,k^*+1}\to j}, h_{\pi_{j,k^*+1}}\}$$

if $k^* < n$. Using this inequality, we get

$$p\sum_{i=1}^{n}\left\lfloor\frac{\bar{t}_j}{h_{\pi_{j,i}}}\right\rfloor \mathbb{1}\left[\tau_{\pi_{j,i}\to j} \leq \bar{t}_j\right] = p\sum_{i=1}^{k^*}\left\lfloor\frac{\bar{t}_j}{h_{\pi_{j,i}}}\right\rfloor \mathbb{1}\left[\tau_{\pi_{j,i}\to j} \leq \bar{t}_j\right] \tag{50}$$

for any $k^*$.

**(Part 2)**

We have three options:

1. If $\left(\sum_{i=1}^{k^*}\frac{p}{h_{\pi_{j,i}}}\right)^{-1} \geq \max\{\tau_{\pi_{j,k^*}\to j}, h_{\pi_{j,k^*}}\}$, then, using (50) and $\lfloor x\rfloor \leq x$ for all $x \geq 0$, we get

$$p\sum_{i=1}^{n}\left\lfloor\frac{\bar{t}_j}{h_{\pi_{j,i}}}\right\rfloor \mathbb{1}\left[\tau_{\pi_{j,i}\to j} \leq \bar{t}_j\right] \leq p\sum_{i=1}^{k^*}\frac{\bar{t}_j}{h_{\pi_{j,i}}} = \frac{1}{8}\left(\sum_{i=1}^{k^*}\frac{p}{h_{\pi_{j,i}}}\right)^{-1} p\sum_{i=1}^{k^*}\frac{1}{h_{\pi_{j,i}}} = \frac{1}{8}.$$

2. If $\left(\sum_{i=1}^{k^*}\frac{p}{h_{\pi_{j,i}}}\right)^{-1} < \max\{\tau_{\pi_{j,k^*}\to j}, h_{\pi_{j,k^*}}\}$ and $k^* = 1$, then

$$p\sum_{i=1}^{n}\left\lfloor\frac{\bar{t}_j}{h_{\pi_{j,i}}}\right\rfloor \mathbb{1}\left[\tau_{\pi_{j,i}\to j} \leq \bar{t}_j\right] = p\left\lfloor\frac{\bar{t}_j}{h_{\pi_{j,k^*}}}\right\rfloor \mathbb{1}\left[\tau_{\pi_{j,k^*}\to j} \leq \bar{t}_j\right] = 0$$

because

$$\bar{t}_j = \frac{1}{8}\max\left\{\max\{\tau_{\pi_{j,k^*}\to j}, h_{\pi_{j,k^*}}\}, \left(\sum_{i=1}^{k^*}\frac{p}{h_{\pi_{j,i}}}\right)^{-1}\right\} = \frac{1}{8}\max\{\tau_{\pi_{j,k^*}\to j}, h_{\pi_{j,k^*}}\} < \max\{\tau_{\pi_{j,k^*}\to j}, h_{\pi_{j,k^*}}\}.$$

3. If $\left(\sum_{i=1}^{k^*}\frac{p}{h_{\pi_{j,i}}}\right)^{-1} < \max\{\tau_{\pi_{j,k^*}\to j}, h_{\pi_{j,k^*}}\}$ and $k^* > 1$, then $\bar{t}_j = \frac{1}{8}\max\{\tau_{\pi_{j,k^*}\to j}, h_{\pi_{j,k^*}}\}$ and

$$\left\lfloor\frac{\bar{t}_j}{h_{\pi_{j,i}}}\right\rfloor \mathbb{1}\left[\tau_{\pi_{j,i}\to j} \leq \bar{t}_j\right] = 0$$

for all $i \leq k^*$ such that $\max\{\tau_{\pi_{j,i}\to j}, h_{\pi_{j,i}}\} = \max\{\tau_{\pi_{j,k^*}\to j}, h_{\pi_{j,k^*}}\}$. If this equality holds for all $i \leq k^*$, then

$$p\sum_{i=1}^{n}\left\lfloor\frac{\bar{t}_j}{h_{\pi_{j,i}}}\right\rfloor \mathbb{1}\left[\tau_{\pi_{j,i}\to j} \leq \bar{t}_j\right] = 0.$$

Otherwise, there exists $\ell < k^*$ such that $\max\{\tau_{\pi_{j,\ell}\to j}, h_{\pi_{j,\ell}}\} < \max\{\tau_{\pi_{j,k^*}\to j}, h_{\pi_{j,k^*}}\}$ and

$$p\sum_{i=1}^{n}\left\lfloor\frac{\bar{t}_j}{h_{\pi_{j,i}}}\right\rfloor \mathbb{1}\left[\tau_{\pi_{j,i}\to j}\leq \bar{t}_j\right] = p\sum_{i=1}^{\ell}\left\lfloor\frac{\bar{t}_j}{h_{\pi_{j,i}}}\right\rfloor \mathbb{1}\left[\tau_{\pi_{j,i}\to j}\leq \bar{t}_j\right].$$

It is not possible that $\max\{\tau_{\pi_{j,\ell}\to j}, h_{\pi_{j,\ell}}\} \geq \left(\sum_{i=1}^{\ell}\frac{p}{h_{\pi_{j,i}}}\right)^{-1}$ because it would yield the inequality

$$\bar{t}_j = \frac{1}{8}\max\{\tau_{\pi_{j,k^*}\to j}, h_{\pi_{j,k^*}}\} > \frac{1}{8}\max\{\tau_{\pi_{j,\ell}\to j}, h_{\pi_{j,\ell}}\}$$

$$= \frac{1}{8}\max\left\{\max\{\tau_{\pi_{j,\ell}\to j}, h_{\pi_{j,\ell}}\}, \left(\sum_{i=1}^{\ell}\frac{p}{h_{\pi_{j,i}}}\right)^{-1}\right\}$$

that contradicts the fact that $\bar{t}_j$ is the minimum (see (46)). Thus, we have

$$\bar{t}_j \leq \frac{1}{8}\max\left\{\max\{\tau_{\pi_{j,\ell}\to j}, h_{\pi_{j,\ell}}\}, \left(\sum_{i=1}^{\ell}\frac{p}{h_{\pi_{j,i}}}\right)^{-1}\right\} = \frac{1}{8}\left(\sum_{i=1}^{\ell}\frac{p}{h_{\pi_{j,i}}}\right)^{-1}$$

and

$$p\sum_{i=1}^{n}\left\lfloor\frac{\bar{t}_j}{h_{\pi_{j,i}}}\right\rfloor \mathbb{1}\left[\tau_{\pi_{j,i}\to j}\leq \bar{t}_j\right] = p\sum_{i=1}^{\ell}\left\lfloor\frac{\bar{t}_j}{h_{\pi_{j,i}}}\right\rfloor \mathbb{1}\left[\tau_{\pi_{j,i}\to j}\leq \bar{t}_j\right] \leq p\sum_{i=1}^{\ell}\frac{\bar{t}_j}{h_{\pi_{j,i}}}$$

$$\leq \frac{1}{8}\left(\sum_{i=1}^{\ell}\frac{p}{h_{\pi_{j,i}}}\right)^{-1} p\sum_{i=1}^{\ell}\frac{1}{h_{\pi_{j,i}}} \leq \frac{1}{8}.$$

In total, we have

$$p\sum_{i=1}^{n}\left\lfloor\frac{\bar{t}_j}{h_{\pi_{j,i}}}\right\rfloor \mathbb{1}\left[\tau_{\pi_{j,i}\to j}\leq \bar{t}_j\right] \leq \frac{1}{8}$$

for $\bar{t}_j$ from (46).

# F  Lower Bound in the Heterogeneous Setup

In this case, we consider the following oracle mappings. For all $i \in [n]$, we define

$$O_i : \underbrace{\mathbb{R}_{\geq 0}}_{\text{time}} \times \underbrace{\mathbb{R}^d}_{\text{point}} \times \underbrace{(\mathbb{R}_{\geq 0}\times\mathbb{R}^d\times\{0,1\})}_{\text{input state}} \to \underbrace{(\mathbb{R}_{\geq 0}\times\mathbb{R}^d\times\{0,1\})}_{\text{output state}}\times\mathbb{R}^d$$

such that

$$O_i(t, x, (s_t, s_x, s_q)) = \begin{cases} ((t, x, 1), & 0), & s_q = 0, \\ ((s_t, s_x, 1), & 0), & s_q = 1, t < s_t + h_i, \quad (51) \\ ((0, 0, 0), & \nabla f_i(s_x; \xi)), & s_q = 1, t \geq s_t + h_i, \end{cases}$$

where $\xi \sim \mathcal{D}$. Unlike (27), the mapping (51) returns $\nabla f_i(s_x; \xi)$.

**Theorem 20.** *Consider Protocol 8 with the mappings (51). We take any $h_i \geq 0$ and $\tau_{i\to j} \geq 0$ for all $i, j \in [n]$ such that $\tau_{i\to j} \leq \tau_{i\to k} + \tau_{k\to j}$ for all $i, k, j \in [n]$. We fix $L, \Delta, \varepsilon, \sigma^2 > 0$ that satisfy the inequality $\varepsilon < c_1 L\Delta$. For any algorithm $A \in \mathcal{A}_{\mathrm{zr}}$, there exists a function $f = \frac{1}{n}\sum_{i=1}^{n}f_i$, which satisfy Assumptions 1, 2 and $f(0) - f^* \leq \Delta$, and stochastic gradient mappings $\nabla f_i(\cdot;\cdot)$, which satisfy Assumption 3 ($\mathbb{E}_\xi[\nabla f_i(x;\xi)] = \nabla f_i(x)$ and $\mathbb{E}_\xi[\|\nabla f_i(x;\xi) - \nabla f_i(x)\|^2] \leq \sigma^2$), such that*

$$\mathbb{E}\left[\inf_{k\in S_t, i\in[n]}\left\|\nabla f(x_i^k)\right\|^2\right] > \varepsilon, \text{ where } S_t := \left\{k \in \mathbb{N}_0 \,|\, t^k \leq t\right\},$$

$$t = c_2 \times \frac{L\Delta}{\varepsilon}\max\left\{\max_{i,j\in[n]}\tau_{i\to j}, \max_{i\in[n]}h_i, \frac{\sigma^2}{n\varepsilon}\left(\frac{1}{n}\sum_{i=1}^{n}h_i\right)\right\},$$

*The quantities $c_1$ and $c_2$ are universal constants. The sequences $x^k$ and $t^k$ are defined in Protocol 8.*

*Proof.* The last two terms in the max follow from Theorem A.2 by Tyurin and Richtárik (2023), who considered the same setup but with $\tau_{i \to j} = 0$ for all $i, j \in [n]$. It is left to prove the first term. Let us fix $\lambda > 0$. Let us take any pair $(\bar{i}, \bar{j})$ of workers such that $\max_{i,j \in [n]} \tau_{i \to j} = \tau_{\bar{i} \to \bar{j}}$. Next, we split the blocks of the function $F_T(x)$ from (33) and define two new functions:

$$F_{T,1}(x) := -\Psi(1)\Phi(x_1) + \sum_{i \in \{2,\dots,T\}, i \,|\, 2=1} [\Psi(-x_{i-1})\Phi(-x_i) - \Psi(x_{i-1})\Phi(x_i)], \quad (52)$$

and

$$F_{T,2}(x) := \sum_{i \in \{2,\dots,T\}, i \,|\, 2=0} [\Psi(-x_{i-1})\Phi(-x_i) - \Psi(x_{i-1})\Phi(x_i)].$$

We consider the following functions $f_i$ :

$$f_i(x) := \begin{cases} \frac{nL\lambda^2}{l_1} F_{T,1}\left(\frac{x}{\lambda}\right), & i = \bar{i}, \\ \frac{nL\lambda^2}{l_1} F_{T,2}\left(\frac{x}{\lambda}\right), & i = \bar{j}, \\ 0, & i \neq \bar{i} \text{ and } i \neq \bar{j}. \end{cases}$$

Then, we get

$$f(x) = \frac{1}{n} \sum_{i=1}^{n} f_i(x) = \frac{1}{n} \left( \frac{nL\lambda^2}{l_1} F_{T,1}\left(\frac{x}{\lambda}\right) + \frac{nL\lambda^2}{l_1} F_{T,2}\left(\frac{x}{\lambda}\right) \right) = \frac{L\lambda^2}{l_1} F_T\left(\frac{x}{\lambda}\right).$$

Let us show that the function $f$ is $L$-smooth:

$$\|\nabla f(x) - \nabla f(y)\| = \frac{L\lambda}{l_1} \left\| \nabla F_T\left(\frac{x}{\lambda}\right) - \nabla F_T\left(\frac{y}{\lambda}\right) \right\| \leq L \|x - y\|.$$

Let us take

$$T = \left\lfloor \frac{\Delta l_1}{L\lambda^2 \Delta^0} \right\rfloor,$$

then

$$f(0) - \inf_{x \in \mathbb{R}^T} f(x) = \frac{L\lambda^2}{l_1} (F_T(0) - \inf_{x \in \mathbb{R}^T} F_T(x)) \leq \frac{L\lambda^2 \Delta^0 T}{l_1} \leq \Delta.$$

We showed that the function $f$ satisfy Assumptions 1, 2 and $f(0) - f^* \leq \Delta$.

In the oracles $O_i$, we simply take the non-stochastic mappings $\nabla f_i(x; \xi) := \nabla f_i(x)$ that are unbiased and $0$-variance-bounded.

We take

$$\lambda = \frac{l_1 \sqrt{\varepsilon}}{L}$$

to ensure that

$$\|\nabla f(x)\|^2 = \frac{L^2 \lambda^2}{l_1^2} \left\| \nabla F_T\left(\frac{x}{\lambda}\right) \right\|^2 > \frac{L^2 \lambda^2}{l_1^2} = \varepsilon$$

for all $x \in \mathbb{R}^T$ such that $\mathrm{prog}(x) < T$. In the last inequality, we use Lemma 1. Thus

$$T = \left\lfloor \frac{\Delta L}{l_1 \varepsilon \Delta^0} \right\rfloor.$$

Only workers $\bar{i}$ and $\bar{j}$ contain the information about the function $f$. The function $f$ is a zero-chain function, and we split it between workers $\bar{i}$ and $\bar{j}$. Due to this splitting, workers $\bar{i}$ and $\bar{j}$ have to communicate to find the next non-zero coordinate. Only worker $\bar{i}$ can get a non-zero value in the first coordinate through the gradient of $F_{T,1}$. Next, this worker can not get a non-zero value in the second coordinate due to the construction of (52). Thus, it has to pass a vector with a non-zero value in the first coordinate to worker $\bar{j}$ because only this worker can get a non-zero value in the second coordinate. This communication takes at least $\tau_{\bar{i} \to \bar{j}}$ seconds. Using the same reasoning, worker $\bar{j}$ has to send a vector to worker $\bar{i}$ once worker $\bar{j}$ has discovered a non-zero value in the second coordinate.

An algorithm has to repeat such communications at least $\frac{T-1}{2}$ times to find a vector $x \in \mathbb{R}^T$ such that $\text{prog}(x) = T$.

Thus, we get

$$\inf_{k \in S_t, i \in [n]} \left\| \nabla f(x_i^k) \right\|^2 > \varepsilon$$

for

$$t = \tau_{i \to \bar{j}} \left( \frac{T-1}{2} \right) = \frac{\max_{i,j \in [n]} \tau_{i \to j}}{2} \left( \left\lfloor \frac{\Delta L}{l_1 \varepsilon \Delta^0} \right\rfloor - 1 \right).$$

$\square$

# G   Proof of the Time Complexity for Homogeneous Case

**Theorem 4.** *Let Assumptions 1, 2, and 3 hold. We take $\gamma = 1/2L$, batch size $S = \max\{\lceil \sigma^2/\varepsilon \rceil, 1\}$, any pivot worker $j^* \in [n]$, and any spanning trees $\overline{st}$ and $\overline{st}_{\text{bc}}$ in Algorithm 2. For all $K \geq 16L\Delta/\varepsilon$, we get $\frac{1}{K} \sum_{k=0}^{K-1} \mathbb{E}\left[ \left\| \nabla f(x^k) \right\|^2 \right] \leq \varepsilon.$*

*Proof.* Algorithm 2 produces the sequence $x^k$ such as $x^{k+1} = x^k - \frac{\gamma}{s^k} g^k = x^k - \gamma \bar{g}^k$, where $\bar{g}^k := \frac{1}{s^k} g^k$. By the design of the algorithm,

$$\bar{g}^k = \frac{1}{s^k} \sum_{i=1}^{s^k} \nabla f(x^k; \xi_i),$$

where the $\xi_i$ are independent random samples and $s^k \geq S$. We do not dismiss the possibility that the computation and communication times are random, so $s^k$ can be random. Assume $\mathcal{V}_k$ is a $\sigma$-algebra generated by all computation and communication times and $g^0, \ldots, g^{k-1}$, then $s^k$ is $\mathcal{V}_k$–measurable. Using the independence and Assumption 3, we have

$$\mathbb{E}\left[ \bar{g}^k \big| \mathcal{G}_k \right] = \mathbb{E}\left[ \mathbb{E}\left[ \frac{1}{s^k} \sum_{i=1}^{s^k} \nabla f(x^k; \xi_i) \bigg| \mathcal{V}_k \right] \bigg| \mathcal{G}_k \right] = \mathbb{E}\left[ \frac{1}{s^k} \sum_{i=1}^{s^k} \mathbb{E}\left[ \nabla f(x^k; \xi_i) \big| \mathcal{V}_k \right] \bigg| \mathcal{G}_k \right] = \nabla f(x^k)$$

and

$$\mathbb{E}\left[ \left\| \bar{g}^k - \nabla f(x^k) \right\|^2 \big| \mathcal{G}_k \right]$$

$$= \mathbb{E}\left[ \mathbb{E}\left[ \left\| \frac{1}{s^k} \sum_{i=1}^{s^k} \nabla f(x^k; \xi_i) - \nabla f(x^k) \right\|^2 \bigg| \mathcal{V}_k \right] \bigg| \mathcal{G}_k \right]$$

$$= \mathbb{E}\left[ \frac{1}{(s^k)^2} \sum_{i=1}^{s^k} \mathbb{E}\left[ \left\| \nabla f(x^k; \xi_i) - \nabla f(x^k) \right\|^2 \big| \mathcal{V}_k \right] \bigg| \mathcal{G}_k \right]$$

$$\leq \mathbb{E}\left[ \frac{\sigma^2}{s^k} \bigg| \mathcal{G}_k \right] \leq \frac{\sigma^2}{S}.$$

where $\mathcal{G}_k$ is a $\sigma$-algebra generated by $\bar{g}^0, \ldots, \bar{g}^{k-1}$. We can use a well-known SGD result (Ghadimi and Lan, 2013; Khaled and Richtárik, 2022). Using Theorem 21, for the stepsize

$$\gamma = \frac{1}{2L} \min\left\{ 1, \frac{\varepsilon S}{\sigma^2} \right\},$$

we have

$$\frac{1}{K} \sum_{k=0}^{K-1} \mathbb{E}\left[ \left\| \nabla f(x^k) \right\|^2 \right] \leq \varepsilon,$$

if
$$K \geq \frac{8\Delta L}{\varepsilon} + \frac{8\Delta L\sigma^2}{\varepsilon^2 S}.$$

Using the choice of $S$, we get that Algorithm 2 converges after

$$K \geq \frac{16\Delta L}{\varepsilon}$$

steps with

$$\gamma = \frac{1}{2L}\min\left\{1, \frac{\varepsilon S}{\sigma^2}\right\} = \frac{1}{2L}.$$

$\square$

**Theorem 5.** *Consider the assumptions and the parameters from Theorem 4. For any pivot worker $j^* \in [n]$ and spanning trees $\overline{st}$ and $\overline{st}_{\text{bc}}$, Algorithm 2 converges after at most*

$$\Theta\left(\frac{L\Delta}{\varepsilon}t^*(\sigma^2/\varepsilon, [h_i]_{i=1}^n, [\mu_{i\to j^*} + \mu_{j^*\to i}]_{i=1}^n)\right) \tag{10}$$

*seconds, where $\mu_{i\to j^*}$ ($\mu_{j^*\to i}$) is an upper bound on the times required to send a vector from worker $i$ to worker $j^*$ (from worker $j^*$ to worker $i$) along the spanning tree $\overline{st}$ (spanning tree $\overline{st}_{\text{bc}}$).*

*Proof.* Due to Theorem 4, we know that Algorithm 3 finds an $\varepsilon$–stationary point after at most $K = \Theta\left(\frac{L\Delta}{\varepsilon}\right)$ iterations. It is left to bound the time of one iteration to prove the theorem.

For all $j \in [n]$, we define $\pi_{j,\cdot}$ as a permutation that sorts $\{\max\{\mu_{i\to j} + \mu_{j\to i}, h_i\}\}_{i=1}^n$ as

$$\max\{\mu_{\pi_{j,1}\to j} + \mu_{j\to\pi_{j,1}}, h_{\pi_{j,1}}\} \leq \cdots \leq \max\{\mu_{\pi_{j,n}\to j} + \mu_{j\to\pi_{j,n}}, h_{\pi_{j,n}}\}.$$

Let us define the index

$$k^* = \arg\min_{k\in[n]}\max\left\{\max\{\mu_{\pi_{j^*,k}\to j^*} + \mu_{j^*\to\pi_{j^*,k}}, h_{\pi_{j^*,k}}\}, S\left(\sum_{i=1}^k \frac{1}{h_{\pi_{j^*,i}}}\right)^{-1}\right\}.$$

and the set

$$A^* := \{\pi_{j^*,i} \in [n] \,|\, i \leq k^*\}$$

that represents a set of the "fastest" workers that can potentially contribute to an optimization process. We take

$$\bar{t} := 2\max\left\{\max\{\mu_{\pi_{j^*,k^*}\to j^*} + \mu_{j^*\to\pi_{j^*,k^*}}, h_{\pi_{j^*,k^*}}\}, S\left(\sum_{i=1}^{k^*} \frac{1}{h_{\pi_{j^*,i}}}\right)^{-1}\right\}$$
$$= 2\min_{k\in[n]}\max\left\{\max\{\mu_{\pi_{j^*,k}\to j^*} + \mu_{j^*\to\pi_{j^*,k}}, h_{\pi_{j^*,k}}\}, S\left(\sum_{i=1}^k \frac{1}{h_{\pi_{j^*,i}}}\right)^{-1}\right\}. \tag{53}$$

In the following steps of the proof we show that every iteration takes at most

$$\underbrace{\bar{t}}_{\text{(Step 1): Calculate enough stochastic gradients}} + \underbrace{\bar{t}}_{\text{(Step 2): Send stochastic gradients to Process } j^*} + \underbrace{\bar{t}}_{\text{(Step 3): Broadcast a new point}} = 3\bar{t}$$

seconds.

**(Step 1):** Since it takes at most $h_i$ seconds to calculate a stochastic gradient in worker $i$, all workers from the set $A^*$ will calculate at least

$$\sum_{i\in A^*}\left\lfloor\frac{\bar{t}}{h_i}\right\rfloor = \sum_{i=1}^{k^*}\left\lfloor\frac{\bar{t}}{h_{\pi_{j^*,i}}}\right\rfloor \tag{54}$$

stochastic gradients after $\bar{t}$ seconds at the point $x^k$. We have

$$\bar{t} \geq 2\max\{\mu_{\pi_{j^*,k^*}\to j^*} + \mu_{j^*\to\pi_{j^*,k^*}}, h_{\pi_{j^*,k^*}}\} \geq 2\max\{\mu_{\pi_{j^*,i}\to j^*} + \mu_{j^*\to\pi_{j^*,i}}, h_{\pi_{j^*,i}}\} \tag{55}$$

for all $i \leq k^*$ by the definition of the permutations $\pi_{\cdot,\cdot}$. Therefore,

$$\bar{t} \geq 2h_{\pi_{j^*,i}}$$

for all $i \leq k^*$. Thus, using (54) and $\lfloor x \rfloor \geq \frac{x}{2}$ for all $x \geq 1$, we get

$$\sum_{i \in A^*} \left\lfloor \frac{\bar{t}}{h_i} \right\rfloor \geq \sum_{i=1}^{k^*} \frac{\bar{t}}{2h_{\pi_{j^*,i}}} \overset{(53)}{\geq} S.$$

Therefore, after $\bar{t}$ seconds the algorithm will calculate at least $S$ stochastic gradients at the point $x^k$ using the workers $A^*$.

**(Step 2):** By the design of Algorithm 4, once a stochastic gradient $\nabla f(x^k; \bar{\xi})$ is calculated, it is added to $g^k_{i,\text{next}}$. Then, $g^k_{i,\text{next}}$ is assigned to $g^k_{i,\text{send}}$ which is sent to Process $\text{next}_{\overline{st},j^*}(i)$. Finally, Process $\text{next}_{\overline{st},j^*}(i)$ receives $g^k_{i,\text{send}}$ and adds it to $g^k_{(\text{next}_{\overline{st},j^*}(i)),\text{next}}$. Thus, the stochastic gradient $\nabla f(x^k; \bar{\xi})$ is presented in the sum of $g^k_{(\text{next}_{\overline{st},j^*}(i)),\text{next}}$ of Process $\text{next}_{\overline{st},j^*}(i)$. At some point, Process 0 in worker $j^*$ will receive a vector $g^k_{\cdot,\text{send}}$ where the stochastic gradient $\nabla f(x^k; \bar{\xi})$ is presented.

Let us bound the time required to transmit a stochastic gradient to Process 0 of worker $j^*$. Once a stochastic gradient $\nabla f(x^k; \bar{\xi})$ is calculated in Process $i$ from $A^*$, it is added to a vector $g^k_{i,\text{next}}$ in Process $i$. It will take at most $2\rho_{i \to \text{next}_{\overline{st},j^*}(i)}$ seconds to transmit it to Process $\text{next}_{\overline{st},j^*}(i)$ because it takes at most $\rho_{i \to \text{next}_{\overline{st},j^*}(i)}$ seconds to wait for the transmission of a message $g^k_{i,\text{send}}$ where the stochastic gradient $\nabla f(x^k; \bar{\xi})$ is not presented, and an additional $\rho_{i \to \text{next}_{\overline{st},j^*}(i)}$ seconds to send the next $g^k_{i,\text{send}}$ where it will present. After that, Process $\text{next}_{\overline{st},j^*}(i)$ will receive $g^k_{i,\text{send}}$, where the stochastic gradient $\nabla f(x^k; \bar{\xi})$ presents, and add the vector $g^k_{i,\text{send}}$ to $g^k_{(\text{next}_{\overline{st},j^*}(i)),\text{next}}$. Then, it will take at most $2\rho_{\text{next}_{\overline{st},j^*}(i) \to \text{next}(\text{next}_{\overline{st},j^*}(i))}$ seconds to send a vector, where the stochastic gradient $\nabla f(x^k; \bar{\xi})$ presents, to Process $\text{next}(\text{next}_{\overline{st},j^*}(i))$ and so forth. In total, after a finite number of such steps a stochastic gradient calculated in Process $i$ will be transmitted to Process 0 of worker $j^*$. From Definition 3 of $\text{next}_{\overline{st},j^*}$, we can conclude that the vector $\nabla f(x^k; \bar{\xi})$ will be transmitted through the path between workers $i$ and $j^*$ in the spanning tree $\overline{st}$. Thus, it will take at most $2\mu_{i \to j^*}$ seconds by the definition of $\mu_{i \to j^*}$.

Using (55) and the definition of $A^*$, we have

$$\bar{t} \geq 2\max\{\mu_{\pi_{j^*,k^*} \to j^*} + \mu_{j^* \to \pi_{j^*,k^*}}, h_{\pi_{j^*,k^*}}\} \geq 2\max\{\mu_{i \to j^*} + \mu_{j^* \to i}, h_i\} \geq 2\mu_{i \to j^*} \quad (56)$$

for all $i \in A^*$. Therefore, it will take at most $\bar{t}$ seconds to calculate at least $S$ stochastic gradients, and at most $\bar{t}$ seconds to send all these stochastic gradients to Process 0.

**(Step 3):** It is left to estimate the time of the broadcast steps (Lines 7–9 in Algorithm 4) through the spanning tree $\overline{st}_{\text{bc}}$. By the definition of $\mu_{j^* \to i}$, the time required to broadcast $x^k$ to Process $i$ through the spanning tree $\overline{st}_{\text{bc}}$ is less or equal to $2\mu_{j^* \to i}$ since, in all edges from $j^*$ to $i$, workers wait at most $\rho_{\cdot \to \cdot}$ seconds while edges are blocked by previous communications, and additional $\rho_{\cdot \to \cdot}$ seconds to send $x^k$ to next workers. Using (55) and the definition of $A^*$, we have

$$\bar{t} \geq 2\max\{\mu_{\pi_{j^*,k^*} \to j^*} + \mu_{j^* \to \pi_{j^*,k^*}}, h_{\pi_{j^*,k^*}}\} \geq 2\max\{\mu_{i \to j^*} + \mu_{j^* \to i}, h_i\} \geq 2\mu_{j^* \to i}.$$

Thus, every worker from $A^*$ will get $x^k$ after $\bar{t}$ seconds. By combining all times, we can conclude that every iteration in Algorithm 3 will take at most $\bar{t} + \bar{t} + \bar{t} = 3\bar{t}$ seconds.

It left to show that

$$\bar{t} = O\left(\max\left\{\max\{\mu_{\pi_{j^*,k^*} \to j^*} + \mu_{j^* \to \pi_{j^*,k^*}}, h_{\pi_{j^*,k^*}}\}, \frac{\sigma^2}{\varepsilon}\left(\sum_{i=1}^{k^*} \frac{1}{h_{\pi_{j^*,i}}}\right)^{-1}\right\}\right). \quad (57)$$

If $S > 1$, then $S = \max\{\lceil \sigma^2/\varepsilon \rceil, 1\} = \lceil \sigma^2/\varepsilon \rceil \leq 2\sigma^2/\varepsilon$, so it is true. Otherwise, if $S \leq 1$, then

$$\bar{t} \leq 2\max\left\{\max\{\mu_{\pi_{j^*,k^*} \to j^*} + \mu_{j^* \to \pi_{j^*,k^*}}, h_{\pi_{j^*,k^*}}\}, \left(\sum_{i=1}^{k^*} \frac{1}{h_{\pi_{j^*,i}}}\right)^{-1}\right\}$$

$$\leq 2\max\left\{\max\{\mu_{\pi_{j^*,k^*} \to j^*} + \mu_{j^* \to \pi_{j^*,k^*}}, h_{\pi_{j^*,k^*}}\}, h_{\pi_{j^*,k^*}}\right\}$$

$$= 2\max\{\mu_{\pi_{j^*,k^*} \to j^*} + \mu_{j^* \to \pi_{j^*,k^*}}, h_{\pi_{j^*,k^*}}\}$$

and (57) holds. Notice that the r.h.s. of (57) equals to $\mathrm{O}\left(t^*(\sigma^2/\varepsilon, [h_i]_{i=1}^n, [\mu_{i \to j^*} + \mu_{j^* \to i}]_{i=1}^n)\right)$, where $t^*$ is the equilibrium time defined in Definition 2. $\square$

**Theorem 6.** *Consider the assumptions and the parameters from Theorem 4. In each iteration $k$ of Algorithm 3, the computation times of worker $i$ are bounded by $h_i^k$. Let us fix any pivot worker $j^* \in [n]$ and any spanning trees $\overline{st}$ and $\overline{st}_{\mathrm{bc}}$. Then Algorithm 2 converges after at most*

$$\Theta\left(\sum_{k=0}^{\lceil 16L\Delta/\varepsilon \rceil} t^*(\sigma^2/\varepsilon, [h_i^k]_{i=1}^n, [\mu_{i \to j^*}^k + \mu_{j^* \to i}^k]_{i=1}^n)\right) \tag{13}$$

*seconds, where $\mu_{i \to j^*}^k$ ($\mu_{j^* \to i}^k$) is an upper bound on times required to send a vector from worker $i$ to worker $j^*$ (from worker $j^*$ to worker $i$) along the spanning tree $\overline{st}$ (spanning tree $\overline{st}_{\mathrm{bc}}$) in iteration $k$ of Algorithm 3.*

*Proof.* The proof is almost the same as in Theorem 5. If we fix a pivot worker $j^*$, then the $k^{\mathrm{th}}$ iteration will finish after at most

$$c \times t^*(\sigma^2/\varepsilon, [h_i^k]_{i=1}^n, [\mu_{i \to j^*}^k + \mu_{j^* \to i}^k]_{i=1}^n)$$

seconds, where $c$ is a universal constant. According to Theorem 4, the number of iterations is at most $\left\lceil \frac{16L\Delta}{\varepsilon} \right\rceil$. Therefore, the total required time is at most

$$c \times \sum_{k=0}^{\left\lceil \frac{16L\Delta}{\varepsilon} \right\rceil} t^*(\sigma^2/\varepsilon, [h_i^k]_{i=1}^n, [\mu_{i \to j^*}^k + \mu_{j^* \to i}^k]_{i=1}^n).$$

$\square$

# H Proof of the Time Complexity for Heterogeneous Case

**Theorem 8.** *Let Assumptions 1 and 2 hold for the function $f$ and Assumption 3 holds for the functions $f_i$ for all $i \in [n]$. We take $\gamma = 1/2L$, the parameter $S = \max\{\lceil \sigma^2/\varepsilon \rceil, n\}$, any pivot worker $j^* \in [n]$, and any spanning trees $\overline{st}$ and $\overline{st}_{\mathrm{bc}}$, in Algorithm 5. For all iterations number $K \geq 16L\Delta/\varepsilon$, we get $\frac{1}{K} \sum_{k=0}^{K-1} \mathbb{E}\left[\|\nabla f(x^k)\|^2\right] \leq \varepsilon$.*

*Proof.* Algorithm 5 produces the sequence $x^k$ such that

$$x^{k+1} = x^k - \gamma g^k = x^k - \gamma\left(\frac{1}{n}\sum_{i=1}^n \frac{1}{s_i^k} g_i^k\right) = x^k - \gamma\left(\frac{1}{n}\sum_{i=1}^n \frac{1}{s_i^k}\sum_{j=1}^{s_i^k} \nabla f_i(x^k; \xi_{ij})\right),$$

where the $\xi_{ij}$ are independent random samples. Using the independence and Assumption 3, we have $\mathbb{E}\left[g^k \mid \mathcal{G}_k\right] = \nabla f(x^k)$ and

$$\mathbb{E}\left[\|g^k - \nabla f(x^k)\|^2 \,\middle|\, \mathcal{G}_k\right]$$

$$= \mathbb{E}\left[\left\|\frac{1}{n}\sum_{i=1}^n \frac{1}{s_i^k}\sum_{j=1}^{s_i^k} \nabla f_i(x^k; \xi_{ij}) - \frac{1}{n}\sum_{i=1}^n \nabla f_i(x^k)\right\|^2 \,\middle|\, \mathcal{G}_k\right]. \tag{58}$$

where $\mathcal{G}_k$ is a $\sigma$-algebra generated by $g^0, \ldots, g^{k-1}$. We do not dismiss the possibility that the computation and communication times are random, so $s_i^k$ can be random. Assume $\mathcal{V}_k$ is a $\sigma$-algebra generated by all computation and communication times and $g^0, \ldots, g^{k-1}$, then $s_i^k$ is $\mathcal{V}_k$–measurable for all $i \in [n]$. Using the independence of stochastic gradients and the times and the tower property, we get

$$\mathbb{E}\left[\left\|g^k - \nabla f(x^k)\right\|^2 \Big| \mathcal{G}_k\right]$$

$$= \mathbb{E}\left[\mathbb{E}\left[\left\|\frac{1}{n}\sum_{i=1}^{n}\frac{1}{s_i^k}\sum_{j=1}^{s_i^k}\nabla f_i(x^k;\xi_{ij}) - \frac{1}{n}\sum_{i=1}^{n}\nabla f_i(x^k)\right\|^2 \Big| \mathcal{V}_k\right] \Big| \mathcal{G}_k\right] \tag{59}$$

$$= \frac{1}{n^2}\sum_{i=1}^{n}\mathbb{E}\left[\mathbb{E}\left[\left\|\frac{1}{s_i^k}\sum_{j=1}^{s_i^k}\nabla f_i(x^k;\xi_{ij}) - \nabla f_i(x^k)\right\|^2 \Big| \mathcal{V}_k\right] \Big| \mathcal{G}_k\right]$$

$$= \frac{1}{n^2}\sum_{i=1}^{n}\mathbb{E}\left[\frac{1}{(s_i^k)^2}\sum_{j=1}^{s_i^k}\mathbb{E}\left[\left\|\nabla f_i(x^k;\xi_{ij}) - \nabla f_i(x^k)\right\|^2 \Big| \mathcal{V}_k\right] \Big| \mathcal{G}_k\right] \leq \frac{\sigma^2}{n^2}\mathbb{E}\left[\sum_{i=1}^{n}\frac{1}{s_i^k} \Big| \mathcal{G}_k\right]. \tag{60}$$

Algorithm 6 waits for the moment when $s^k \geq \frac{S}{n}$, which is equivalent to

$$b_{j^*} \leq \frac{n^2}{S}. \tag{61}$$

The value $b_{j^*}$ is calculated in Line 20 of Algorithm 7. Due to the asynchronous nature of the algorithm, we can only conclude that

$$b_{j^*} \geq \sum_{p\in[n]:\text{next}_{\overline{st},j^*}(p)=j^*} b_{i,p} + \frac{1}{s_{j^*}^k} \tag{62}$$

because $s_{j^*}^k$ can be increased by the time when Process 0 will receive $b_{j^*}$. Using Line 15 from Algorithm 7, we can unroll the recursion in (62) and get

$$b_{j^*} \geq \sum_{i=1}^{n}\frac{1}{s_i^k}. \tag{63}$$

Let us substitute this inequality to (61) and (60) and get

$$\mathbb{E}\left[\left\|g^k - \nabla f(x^k)\right\|^2 \Big| \mathcal{G}_k\right] \leq \frac{\sigma^2}{S}.$$

As in Theorem 4, we can use the classical SGD result. Using Theorem 21, for the stepsize

$$\gamma = \frac{1}{2L}\min\left\{1, \frac{\varepsilon S}{\sigma^2}\right\},$$

we have

$$\frac{1}{K}\sum_{k=0}^{K-1}\mathbb{E}\left[\left\|\nabla f(x^k)\right\|^2\right] \leq \varepsilon,$$

if

$$K \geq \frac{8\Delta L}{\varepsilon} + \frac{8\Delta L\sigma^2}{\varepsilon^2 S}.$$

Using the choice of $S$, we get that Algorithm 5 converges after

$$K \geq \frac{16\Delta L}{\varepsilon}$$

steps with

$$\gamma = \frac{1}{2L}\min\left\{1, \frac{\varepsilon S}{\sigma^2}\right\} = \frac{1}{2L}.$$

$\square$

**Theorem 9.** *Consider the assumptions and the parameters from Theorem 8. For any pivot worker $j^* \in [n]$ and any spanning trees $\overline{st}$ and $\overline{st}_{bc}$, Algorithm 5 converges after at most*

$$\Theta\left(\frac{L\Delta}{\varepsilon}\max\left\{\max_{i,j\in[n]}\mu_{i\to j}, \max_{i\in[n]}h_i, \frac{\sigma^2}{n\varepsilon}\left(\frac{1}{n}\sum_{i=1}^n h_i\right)\right\}\right)$$

*seconds, where $\mu_{i\to j^*}^k$ ($\mu_{j^*\to i}^k$) is an upper bound on times required to send a vector from worker $i$ to worker $j^*$ (from worker $j^*$ to worker $i$) along the spanning tree $\overline{st}$ (spanning tree $\overline{st}_{bc}$) for all $i \in [n]$.*

*Proof.* Due to Theorem 8, the algorithm converges after $K = \Theta\left(L\Delta/\varepsilon\right)$ iterations. Thus, it left to bound the time of one iteration. At the beginning of every iteration Process 0 broadcasts $x^k$, which takes at most $\max_{i,j\in[n]}\mu_{i\to j}$ seconds. Then, Algorithm 6 waits for the moment when $s^k \geq \frac{S}{n}$, which is equivalent to $\frac{n^2}{S} \geq b_{j^*}$. Thus, Algorithm 6 waits for the moment when $\frac{n^2}{S} \geq b_{j^*}$. We will return to this fact later.

Let us consider the term

$$\frac{S}{n^2}\sum_{i=1}^n \frac{1}{s_i^k},$$

where $s_i^k$ is the number of stochastic gradients calculated in worker $i$. Let us fix any time $\bar{t} > 0$. Then, worker $i$ will calculate at least $\left\lfloor \frac{\bar{t}}{h_i}\right\rfloor$ stochastic gradients by the time $\bar{t}$. Using this, we get

$$\frac{S}{n^2}\sum_{i=1}^n \frac{1}{s_i^k} \leq \frac{S}{n^2}\sum_{i=1}^n \frac{1}{\left\lfloor \frac{\bar{t}}{h_i}\right\rfloor}.$$

Let us take

$$\bar{t} = 2\left(\max_{i\in[n]}h_i + \frac{S}{n}\left(\frac{1}{n}\sum_{i=1}^n h_i\right)\right).$$

Then, since $\lfloor x\rfloor \geq \frac{x}{2}$ for all $x \geq 1$, we get $\left\lfloor \frac{\bar{t}}{h_i}\right\rfloor \geq \frac{\bar{t}}{2h_i}$ and

$$\frac{S}{n^2}\sum_{i=1}^n \frac{1}{s_i^k} \leq \frac{2S}{n\bar{t}}\left(\frac{1}{n}\sum_{i=1}^n h_i\right).$$

Using $\bar{t} \geq \frac{2S}{n}\left(\frac{1}{n}\sum_{i=1}^n h_i\right)$, we get

$$\frac{S}{n^2}\sum_{i=1}^n \frac{1}{s_i^k} \leq 1$$

and

$$\sum_{i=1}^n \frac{1}{s_i^k} \leq \frac{n^2}{S} \tag{64}$$

after at most $\bar{t}$ seconds.

Recall that Algorithm 6 waits for the moment when $\frac{n^2}{S} \geq b_{j^*}$. Note that by the time when Process 0 receives $b_{j^*}$, the counter $s_i^k$ can be the same or increased; thus, $b_{j^*}$ captures potentially outdated information about $s_i^k$. We know that (64) holds after at most $\bar{t}$ seconds. In Line 20 of Algorithm 7, Processes recursively collect $\frac{1}{s_i^k}$ to $b_{j^*}$. Such a procedure will take at most $2\max_{i\in[n]}\mu_{i\to j^*} \leq 2\max_{i,j\in[n]}\mu_{i\to j}$ seconds. Thus, the value $b_{j^*}$ will be less or equal $\frac{n^2}{S}$ after at most $\bar{t} + 2\max_{i,j\in[n]}\mu_{i\to j}$ seconds.

The all reduce operation in (8) will take at most $\max_{i,j \in [n]} \mu_{i \to j}$ seconds. Thus, the total time of one iteration can be bounded by

$$\underbrace{\max_{i,j \in [n]} \mu_{i \to j}}_{\text{broadcast}} + (\bar{t} + 2 \max_{i,j \in [n]} \mu_{i \to j}) + \underbrace{\max_{i,j \in [n]} \mu_{i \to j}}_{\text{all reduce}}$$

$$= O\left( \max_{i,j \in [n]} \mu_{i \to j} + \max_{i \in [n]} h_i + \frac{S}{n}\left( \frac{1}{n} \sum_{i=1}^{n} h_i \right) \right)$$

$$= O\left( \max_{i,j \in [n]} \mu_{i \to j} + \max_{i \in [n]} h_i + \frac{\sigma^2}{n\varepsilon}\left( \frac{1}{n} \sum_{i=1}^{n} h_i \right) \right).$$

seconds.  □

# I  Classical SGD Theory

We reprove the classical SGD result (Ghadimi and Lan, 2013; Khaled and Richtárik, 2022), for completeness.

**Theorem 21.** *Let Assumptions 1 and 2 hold. We consider the SGD method:*

$$x^{k+1} = x^k - \gamma g(x^k),$$

*where*

$$\gamma = \frac{1}{2L} \min\left\{ 1, \frac{\varepsilon}{\sigma^2} \right\}$$

*For all $k \geq 0$, the vector $g(x)$ is a random vector such that $\mathbb{E}\left[ g(x^k) \middle| \mathcal{G}_k \right] = \nabla f(x^k)$,*

$$\mathbb{E}\left[ \left\| g(x^k) - \nabla f(x^k) \right\|^2 \middle| \mathcal{G}_k \right] \leq \sigma^2, \tag{65}$$

*where $\mathcal{G}_k$ is a $\sigma$-algebra generated by $g(x^0), \ldots, g(x^{k-1})$. Then*

$$\frac{1}{K} \sum_{k=0}^{K-1} \mathbb{E}\left[ \left\| \nabla f(x^k) \right\|^2 \right] \leq \varepsilon$$

*for*

$$K \geq \frac{8\Delta L}{\varepsilon} + \frac{8\Delta L \sigma^2}{\varepsilon^2}.$$

*Proof.* From Assumption 1, we have

$$f(x^{k+1}) \leq f(x^k) + \left\langle \nabla f(x^k), x^{k+1} - x^k \right\rangle + \frac{L}{2} \left\| x^{k+1} - x^k \right\|^2$$

$$= f(x^k) - \gamma \left\langle \nabla f(x^k), g(x^k) \right\rangle + \frac{L\gamma^2}{2} \left\| g(x^k) \right\|^2.$$

We denote $\mathcal{G}^k$ as a sigma-algebra generated by $g(x^0), \ldots, g(x^{k-1})$. Using unbiasedness and (65), we obtain

$$\mathbb{E}\left[ f(x^{k+1}) \middle| \mathcal{G}^k \right] \leq f(x^k) - \gamma\left( 1 - \frac{L\gamma}{2} \right) \left\| \nabla f(x^k) \right\|^2 + \frac{L\gamma^2}{2} \mathbb{E}\left[ \left\| g(x^k) - \nabla f(x^k) \right\|^2 \middle| \mathcal{G}^k \right]$$

$$\leq f(x^k) - \gamma\left( 1 - \frac{L\gamma}{2} \right) \left\| \nabla f(x^k) \right\|^2 + \frac{L\gamma^2 \sigma^2}{2}.$$

Since $\gamma \leq 1/L$, we get

$$\mathbb{E}\left[ f(x^{k+1}) \middle| \mathcal{G}^k \right] \leq f(x^k) - \frac{\gamma}{2} \left\| \nabla f(x^k) \right\|^2 + \frac{L\gamma^2 \sigma^2}{2}.$$

We subtract $f^*$ and take the full expectation to obtain

$$\mathbb{E}\left[f(x^{k+1}) - f^*\right] \le \mathbb{E}\left[f(x^k) - f^*\right] - \frac{\gamma}{2}\mathbb{E}\left[\left\|\nabla f(x^k)\right\|^2\right] + \frac{L\gamma^2\sigma^2}{2}.$$

Next, we sum the inequality for $k \in \{0, \dots, K-1\}$:

$$\mathbb{E}\left[f(x^K) - f^*\right] \le f(x^0) - f^* - \sum_{k=0}^{K-1}\frac{\gamma}{2}\mathbb{E}\left[\left\|\nabla f(x^k)\right\|^2\right] + \frac{KL\gamma^2\sigma^2}{2}$$

$$= \Delta - \sum_{k=0}^{K-1}\frac{\gamma}{2}\mathbb{E}\left[\left\|\nabla f(x^k)\right\|^2\right] + \frac{KL\gamma^2\sigma^2}{2}.$$

Finally, we rearrange the terms and use that $\mathbb{E}\left[f(x^K) - f^*\right] \ge 0$:

$$\frac{1}{K}\sum_{k=0}^{K-1}\mathbb{E}\left[\left\|\nabla f(x^k)\right\|^2\right] \le \frac{2\Delta}{\gamma K} + L\gamma\sigma^2.$$

The choice of $\gamma$ and $K$ ensures that

$$\frac{1}{K}\sum_{k=0}^{K-1}\mathbb{E}\left[\left\|\nabla f(x^k)\right\|^2\right] \le \varepsilon.$$

$\square$

**Theorem 22.** *Let Assumptions 4 and 5 hold. We consider the SGD method:*

$$x^{k+1} = x^k - \gamma g(x^k),$$

*where*

$$\gamma = \frac{\varepsilon}{M^2 + \sigma^2}$$

*For all $k \ge 0$, the vector $g(x)$ is a random vector such that $\mathbb{E}\left[g(x^k)\big| \mathcal{G}_k\right] \in \partial f(x^k)$*

$$\mathbb{E}\left[\left\|g(x^k) - \mathbb{E}\left[g(x^k)\big|\mathcal{G}_k\right]\right\|^2\bigg|\mathcal{G}_k\right] \le \sigma^2,$$

*where $\mathcal{G}_k$ is a $\sigma$-algebra generated by $g(x^0), \dots, g(x^{k-1})$. Then*

$$\mathbb{E}\left[f\left(\frac{1}{K}\sum_{k=0}^{K-1}x^k\right)\right] - f(x^*) \le \varepsilon \tag{66}$$

*for*

$$K \ge \frac{(M^2 + \sigma^2)\left\|x^* - x^0\right\|^2}{\varepsilon^2}.$$

*Proof.* We denote $\mathcal{G}^k$ as a sigma-algebra generated by $g(x^0), \dots, g(x^{k-1})$. Using the convexity, for all $x \in \mathbb{R}^d$, we have

$$f(x) \ge f(x^k) + \left\langle \mathbb{E}\left[g(x^k)\big|\mathcal{G}^k\right], x - x^k\right\rangle = f(x^k) + \mathbb{E}\left[\left\langle g(x^k), x - x^k\right\rangle\big|\mathcal{G}^k\right].$$

Note that

$$\left\langle g(x^k), x - x^k\right\rangle = \left\langle g(x^k), x^{k+1} - x^k\right\rangle + \left\langle g(x^k), x - x^{k+1}\right\rangle$$

$$= -\gamma\left\|g(x^k)\right\|^2 + \frac{1}{\gamma}\left\langle x^k - x^{k+1}, x - x^{k+1}\right\rangle$$

$$= -\gamma\left\|g(x^k)\right\|^2 + \frac{1}{2\gamma}\left\|x^k - x^{k+1}\right\|^2 + \frac{1}{2\gamma}\left\|x - x^{k+1}\right\|^2 - \frac{1}{2\gamma}\left\|x - x^k\right\|^2$$

$$= -\frac{\gamma}{2}\left\|g(x^k)\right\|^2 + \frac{1}{2\gamma}\left\|x - x^{k+1}\right\|^2 - \frac{1}{2\gamma}\left\|x - x^k\right\|^2$$

and

$$\mathbb{E}\left[\left.\left\|g(x^k)\right\|^2\right|\mathcal{G}^k\right] = \mathbb{E}\left[\left.\left\|g(x^k) - \mathbb{E}\left[\left.g(x^k)\right|\mathcal{G}^k\right]\right\|^2\right|\mathcal{G}^k\right] + \left\|\mathbb{E}\left[\left.g(x^k)\right|\mathcal{G}^k\right]\right\|^2 \leq \sigma^2 + M^2.$$

Therefore, we get

$$
\begin{aligned}
f(x^k) &\leq f(x) + \mathbb{E}\left[\left.\langle g(x^k), x^k - x\rangle\right|\mathcal{G}^k\right] \\
&= f(x) + \frac{\gamma}{2}\mathbb{E}\left[\left.\left\|g(x^k)\right\|^2\right|\mathcal{G}^k\right] + \frac{1}{2\gamma}\left\|x - x^k\right\|^2 - \frac{1}{2\gamma}\mathbb{E}\left[\left.\left\|x - x^{k+1}\right\|^2\right|\mathcal{G}^k\right] \\
&\leq f(x) + \frac{\gamma}{2}\left(M^2 + \sigma^2\right) + \frac{1}{2\gamma}\left\|x - x^k\right\|^2 - \frac{1}{2\gamma}\mathbb{E}\left[\left.\left\|x - x^{k+1}\right\|^2\right|\mathcal{G}^k\right].
\end{aligned}
$$

By taking the full expectation and summing the last inequality for $t$ from 0 to $K-1$, we obtain

$$
\begin{aligned}
\mathbb{E}\left[\sum_{k=0}^{K-1} f(x^k)\right] &\leq Kf(x) + \frac{K\gamma}{2}\left(M^2 + \sigma^2\right) + \frac{1}{2\gamma}\left\|x - x^0\right\|^2 - \frac{1}{2\gamma}\mathbb{E}\left[\left\|x - x^K\right\|^2\right] \\
&\leq Kf(x) + \frac{K\gamma}{2}\left(M^2 + \sigma^2\right) + \frac{1}{2\gamma}\left\|x - x^0\right\|^2.
\end{aligned}
$$

Let divide the last inequality by $K$, take $x = x^*$, and use the convexity:

$$\mathbb{E}\left[f\left(\frac{1}{K}\sum_{k=0}^{K-1} x^k\right)\right] - f(x^*) \leq \frac{\gamma}{2}\left(M^2 + \sigma^2\right) + \frac{1}{2\gamma K}\left\|x^* - x^0\right\|^2.$$

The choices of $\gamma$ and $K$ ensure that (66) holds. $\qquad\square$

## J  Experiments

We now consider Fragile SGD with Minibatch SGD on quadratic optimization tasks with stochastic gradients. The working environment was emulated in Python 3.8 with one Intel(R) Xeon(R) Gold 6248 CPU @ 2.50GHz. The homogeneous optimization problem (1) is constructed in the following way. We take

$$f(x) = \frac{1}{2}x^\top \mathbf{A}x - b^\top x \quad \forall x \in \mathbb{R}^d,$$

$d = 1000,$

$$\mathbf{A} = \frac{1}{4}\begin{pmatrix} 2 & -1 & & 0 \\ -1 & \ddots & \ddots & \\ & \ddots & \ddots & -1 \\ 0 & & -1 & 2 \end{pmatrix} \in \mathbb{R}^{d\times d}, \quad \text{and} \quad b = \frac{1}{4}\begin{bmatrix} -1 \\ 0 \\ \vdots \\ 0 \end{bmatrix} \in \mathbb{R}^d.$$

Let us define $[x]_j$ as the $j^{\text{th}}$ index of a vector $x \in \mathbb{R}^d$. All $n$ workers calculate the stochastic gradients

$$[\nabla f(x;\xi)]_j := [\nabla f(x)]_j\left(1 + \mathbb{1}\left[j > \text{prog}(x)\right]\left(\frac{\xi}{p} - 1\right)\right) \quad \forall x \in \mathbb{R}^d, \forall i \in [n],$$

where $\xi \sim \text{Bernouilli}(p)$, $p \in (0,1]$. In our experiments, we take $p = 0.001$ and the starting point $x^0 = [\sqrt{d}, 0, \ldots, 0]^\top$. We emulate our setup by considering that the $i^{\text{th}}$ worker requires $h_i = 1$ second to calculate a stochastic gradient. And we assume that the workers have the structure of 2D-Mesh (see Figure 4a) and take $\rho_{i\to j} = \rho \in \{0.1, 1, 10\}$ seconds for all edges that connect workers in 2D-Mesh. We take $n = 100$. In all methods we fine-tune step sizes from the set $\{2^i \mid i \in [-20, 20]\}$. In Fragile SGD, we fine-tune the batch size $S$ from the set $\{10, 20, 40, 80, 120\}$.

The results are presented in Figures 5, 6, and 7. The plots are fully consisted with Table 1. One can see that when the communication is fast (Fig. 5), there is no big difference between the methods because both Fragile SGD and Minibatch SGD use all workers in the optimization steps. However, when we start decreasing the communication speed, we observe that Fragile SGD converges faster.

We looked deeper into the optimization processes of Fragile SGD in Figure 7 and observed that only 13 of 100 workers contribute to the optimization process for the batch size $S = 120$. Other workers are too far away from the pivot worker, and their contributions can only slow down optimization.

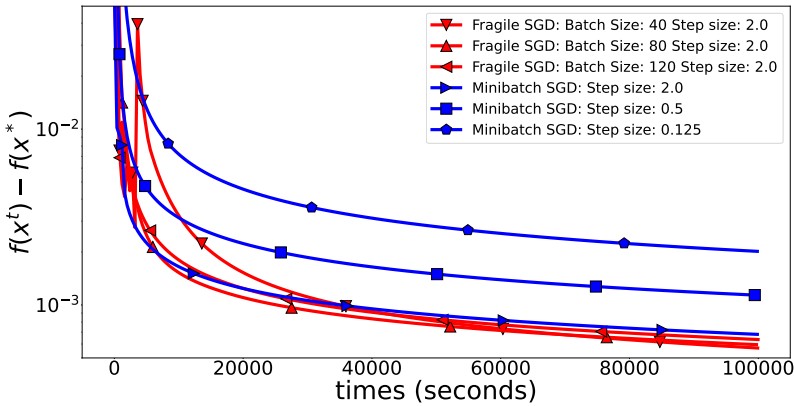

Figure 5: The communication time $\rho = 0.1$ seconds (Fast communication)

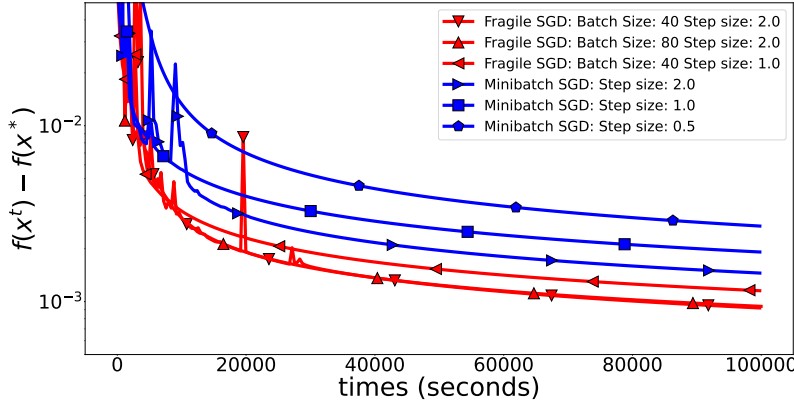

Figure 6: The communication time $\rho = 1$ seconds (Medium speed communication)

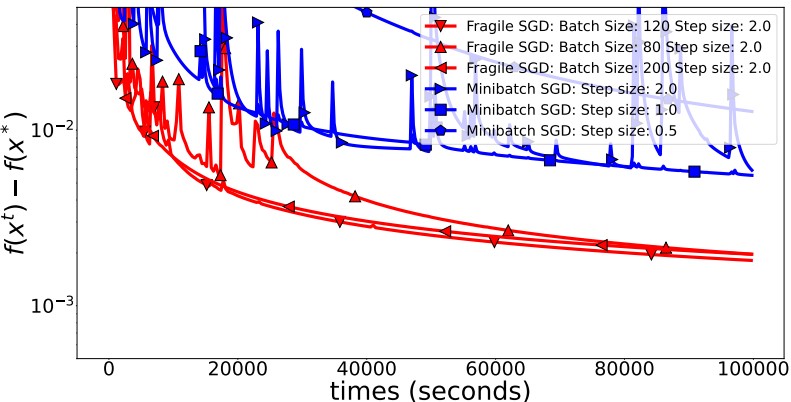

Figure 7: The communication time $\rho = 10$ seconds (Slow communication)

### J.1 Experiments with Logistic Regression: Fast vs Slow Communication

We now repeat the previous experiments but with logistic regression on *MNIST* dataset (LeCun et al., 2010) with 100 workers. We consider two regimes: fast and slow communication between workers. One can see that when the communication is fast, the gap between the methods is small, which is expected and compliant with the theory. However, Fragile SGD is much faster and has better test accuracy when the communication is slow.

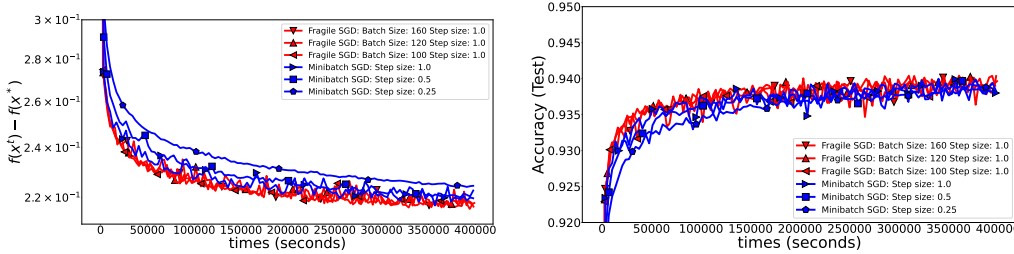

Figure 8: The communication time $\rho = 0.1$ seconds (Fast communication)

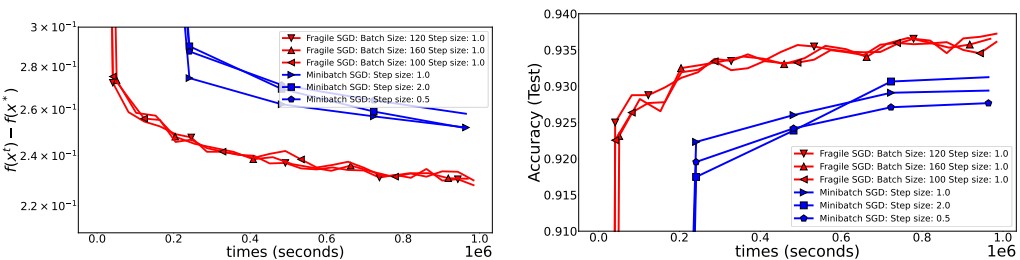

Figure 9: The communication time $\rho = 10$ seconds (Slow communication)

### J.2 Experiments with ResNet-18

We test algorithms on an image recognition task, *CIFAR10* (Krizhevsky et al., 2009), with the *ResNet-18* (He et al., 2016) deep neural network (the number of parameters $d \approx 10^7$). We use the torus structure and 9 workers. We run all methods with the step sizes $\{0.025, 0.25, 2.5\}$. Our findings from the low-scale experiments are also evident in the large-scale experiments. Fragile SGD converges faster than Minibatch SGD in terms of function values. When we compare accuracies on the test split of *MNIST*, the superiority of Fragile SGD is even more transparent.

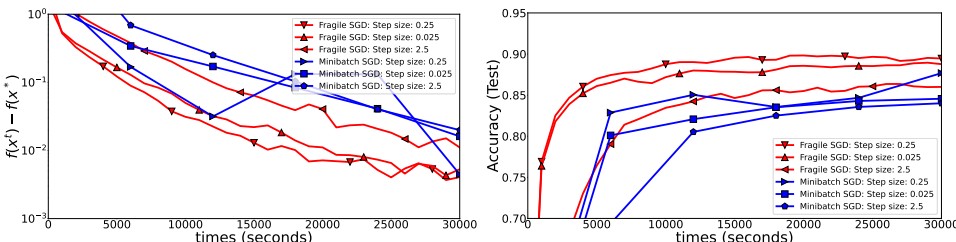

Figure 10: *ResNet-18* on *CIFAR10* dataset with 9 workers and the torus structure with the communication time $\rho = 1$ seconds (Medium communication)

