# OpenReview forum: "On the Optimal Time Complexities in Decentralized Stochastic Asynchronous Optimization"
_NeurIPS.cc/2024/Conference — NeurIPS 2024 poster_

### Official Review · Reviewer_qqc5 · 2024-07-12

**Soundness:** 3
**Presentation:** 4
**Contribution:** 4
**Rating:** 7
**Confidence:** 3

**Summary:**

The paper presents new theoretical advancements in decentralized stochastic asynchronous optimization. They introduce two new methods: Fragile SGD and Amelie SGD. Fragile SGD is designed for homogeneous setups, while Amelie SGD addresses heterogeneous setups. Both methods aim to achieve near-optimal time complexities under arbitrary heterogeneous computation and communication speeds. The paper establishes new lower bounds for time complexities in these setups and proves that the proposed methods match these bounds up to logarithmic factors.

**Strengths:**

The paper is well-written and organized, making significant theoretical contributions by establishing new lower bounds for time complexities in decentralized asynchronous optimization. The introduction and analysis of Fragile SGD and Amelie SGD add value to the field. Additionally, the proposed methods demonstrate robustness to dynamic computation and communication times.

**Weaknesses:**

1. Although the algorithms are interpretable, they may be challenging to implement in practical systems.

2. There is limited experimental validation of the proposed methods.

**Questions:**

I have no questions.

**Limitations:**

Limitations are addressed in the paper (selecting optimal pivot worker)

---

> ### Author Rebuttal · Authors · 2024-08-05
>
> Thank you for the positive evaluation of our work! Let us respond to the weaknesses:
>
> > Although the algorithms are interpretable, they may be challenging to implement in practical systems.
>
> We agree that our algorithms seem too lengthy, but the goal was to provide a very detailed listing to ensure that the algorithms can be implemented correctly and without challenges. We tried to give as many details as possible.
>
> > There is limited experimental validation of the proposed methods.
>
> The main target of the paper was to obtain the fundamental limits of decentralized optimization. Thus, we focused our attention on theoretical results and lower bounds, which we believe are important to understand the nature of decentralized optimization. In Section I, we run our algorithms to test our theory and ensure that the dependencies that appear in theory are reflected in practice. *Moreover, we prepared extra experiments with logistic regression and a neural network that the reviewer can find in the global rebuttal response, which we will add to the camera-ready version.*

---

> > ### Comment · Reviewer_qqc5 · 2024-08-12
> >
> > Thanks for the response.

---

### Official Review · Reviewer_YQkr · 2024-07-12

**Soundness:** 3
**Presentation:** 3
**Contribution:** 3
**Rating:** 6
**Confidence:** 3

**Summary:**

This paper considers the decentralizd optimization problem, where clients have heterogeneous computation and communication times. The paper proved a lower bound on the optimal physical time complexity (wherein the physical computation and communication times are take into consideration). It proposed Fragile SGD, an asynchronous decentralized first order method that matches this lower bound up to a logarithmic factor. It further considers the data heterogenous setting and proposed Amelie SGD that matches the lower bound in such a setting.

**Strengths:**

Unlike most other previous works, this paper takes into consideration the physical computation time and communication time of the clients in the decentralized network. The paper then proves a lower bound for decentralized optimization in terms of the physical time, and it proposed a nearly optimal algorithm Fragile SGD with matching physical running time. This is an interesting research setting, for which one could argue is more realistic, since whereas previous works on decentralized asynchrounous algorithms mostly considered the iteration complexity without worrying about the physical computation and communication time. In addition, unlike many other previous theoretical works on decentralized optimization where message passing is typically restricted to gossip type communications, this paper considers relayed communications where communication between any nodes on the graph are relayed on certain paths (in Fragile SGD, the paths are on certain spanning trees). These are all interesting additions to discussions around decentralized asynchronous algorithms.

**Weaknesses:**

1. This is also somewhat related to the strengths part. While the paper keeps referring to and comparing against some previous works (e.g. Even et al 2024), I do not think that there is a fair comparison to start with. Ultimately, the setting in which this paper is concerned is rather different from the previous works', where previous works typically only considers Gossip type communication and measures the convergence speed in terms of, number of iterations, communication rounds etc, but not the physical time of the whole process. Such a comparison is particular strange when the authors claimed that their method's convergence time does not depend on the spectral gap while previous works do -- this is obviously the case because previous works considers gossiping which is fundamentally constrained by the spectral gap while this paper allows any pair of clients to communicate via some paths on the graphs which is not directly constrained by the spectral gap (but can be otherwise more directly constrained by, say, the diameter of the graph).

2. The second point is also somewhat related. I appreciate it that the authors takes computation and communication times into consideration. This might be somewhat a more realistic setting than what some previous works considers on the surface, but I'm still not very convinced that such a setting is actually closer to reality than the others. One could argue that the works in this paper is to some extend overfitting on certain metrics which are not necessarily a good measurement of the methods' performances in reality, just like the other (seemingly) simpler setting previous works consider.

3. Fragile SGD requires a pivot worker which is, in some sense, at the center of the graph, to coordinate the updates. It also requires the knowledge of a spanning tree connecting the pivot worker to and from all other workers. Now suppose we take the pivot worker as the server, and set the communication time of all other workers as the distance from the worker to the pivot on the spanning tree, and we consider a the distributed asynchronous optimization problem with a centralized server. My understanding is that, conceptually speaking, the setting that the authors consider is closer to this hypothetical setting than to the more canonical decentralized setting that previous papers consider.

**Questions:**

see weaknesses.

**Limitations:**

see weaknesses.

---

> ### Author Rebuttal · Authors · 2024-08-05
>
> Thank you for your review! We now respond to the weaknesses:
>
> > 1. This is also somewhat related to the strengths part. While the paper keeps referring to and comparing against some previous works...
>
> Before we answer this comment, note that our new methods are optimal (up to log factor) for all possible parameters $h_i, \tau_{i \to j}, ...$, so if we want to show that some other method is worse, it is enough to show in one (practical) scenario (take $\tau_{i \to j} = 0$ for instance). Returning back to the comment, we totally agree that most of the previous methods designed their methods for a different setting, Gossip-type communication, but it does not mean that we can not compare the previous methods with our new methods. For instance, we consider Asynchronous SGD in the Gossip framework (Even et al 2024) in the scenario when the communication is free or negligible (Section 5.4). This scenario is equivalent to the setting when the Gossip matrix $w_{ij}$ is non-zero for all $i \neq j$ (in one step, one worker can send a vector to all other workers because communication is free). We explain that even in this scenario, when Asynchronous SGD is allowed for free to send one vector to other workers, it is worse than our method. We did similar work with other methods and showed that they are strongly worse (Sections 5.4, B.4, and C.7).
>
> In fact, our setting is more general than the Gossip communication. Indeed, recall that in the Gossip communication, a worker is allowed to get vectors $\{x_j\}$ from other workers through the operation $\sum_{j=1}^n w_{ij} x_j,$ where $w_{ij}$ either zero or not zero. This is equivalent to our setting for the case when the communication time $\rho_{ij} = \infty$ when $w_{ij}$ is zero, and $\rho_{ij} = 1$ if $w_{ij}$ is not zero, and worker $i$ sums the received vectors. But our setting is richer since we allow different communication and computation times, and allow workers to do with vectors whatever they like (not only to sum).
> We should probably add this paragraph to the paper. Thank you for the good comment!
>
> Finally, the Gossip framework codes the communication graph through the matrix $W = [w_{ij}]$. We propose to code graphs through the times $\rho_{i \to j}$ ($\tau_{i \to j}$). Our approach is closer to real scenarios because, as we explained previously, it includes the Gossip framework.
>
> > 2. The second point is also somewhat related. I appreciate it that the authors takes computation and communication times into consideration...
>
> In the previous comment, we explained that our time complexity framework is more general. As a consequence, it includes more realistic scenarios than the Gossip framework, for instance. We argue that the time complexity metric is one of the most natural metrics for parallel setting, unlike the iteration complexity metric. Our assumptions are generic and only assume that communication and computation take time, so it is not clear where we overfitted our methods. Of course, it is possible to add other time restrictions to take this setup closer to reality, but our work can be a starting point.
>
> > 3. Fragile SGD requires a pivot worker which is, in some sense, at the center of the graph, to coordinate the updates...
>
> Fragile SGD and Amelie SGD were designed to show that our lower bounds are tight (up to log factor). We do not deny the possibility of designing methods without pivot workers and spanning trees. However, any other method without these algorithmic components will not beat these methods due to the lower bounds. The fact that the methods distinguish one pivot worker does not contradict the decentralized setting. We agree that Fragile SGD and Amelie SGD, in some sense, mimic the centralized setup, but we are totally allowed to do it. Note that we can design similar methods in the Gossip setting, taking one pivot worker that aggregates all vectors from other workers using $W$ through several steps: in the first step, the pivot worker aggregates vectors from the neighbors $\sum_{j=1}^n w_{ij} x_j,$ in the second step, it aggregates the vector from the neighbors of the neighbors $\sum_{j=1}^n w_{ij} w_{jk} x_k,$ where $w_{jk} x_k$ is calculated in the first step in worker $j,$ and so on. This way, we can aggregate all information in one worker.

---

> > ### Comment · Reviewer_YQkr · 2024-08-11
> >
> > Thanks for the explanation. I do not have any more questions.

---

### Official Review · Reviewer_VUTZ · 2024-07-13

**Soundness:** 3
**Presentation:** 2
**Contribution:** 3
**Rating:** 6
**Confidence:** 2

**Summary:**

The paper addresses the decentralized stochastic asynchronous optimization setup. The authors establish new time complexity lower bounds for both homogeneous and heterogeneous setups, assuming bounded computation and communication speeds. They introduce two methods: Fragile SGD, a nearly optimal method, and Amelie SGD, an optimal method. Their methods leverage spanning trees to solve the problem. These methods achieve convergence under arbitrary heterogeneous computation and communication speeds and match the derived lower bounds, up to a logarithmic factor in the homogeneous setting.

**Strengths:**

Originality:
The innovative use of spanning trees to address the decentralized stochastic asynchronous optimization problem represents a significant departure from existing algorithms, introducing new theoretical contributions. Furthermore, the proposed algorithm exhibits greater robustness compared to previous methods, meaning that the impact of the worst-performing nodes on the overall update speed is minimized or even eliminated.
Significance:
The contributions of this paper to the field are significant. It offers a novel approach(spanning tree) for analyzing computational time complexity, which could be beneficial for the time complexity analysis of other decentralized stochastic asynchronous optimization algorithms. Additionally, the proposed method may assist in practical applications by providing insights into the selection of central nodes.
Quality:
The paper provides a detailed explanation of the prerequisite knowledge, a thorough proof process, and a comprehensive introduction to the algorithm's features. Additionally, it includes another algorithm and experimental results in the appendix, offering a well-rounded and complete presentation of the material.

**Weaknesses:**

Regarding the structure of the paper, I think there are several issues. First, one of the two algorithms mentioned in the abstract is relegated entirely to the appendix, which is inappropriate for the paper’s structure. Second, the experimental results are also exclusively placed in the appendix, which does not conform to standard layout. Finally, Section 6(Example: line or circle), which serves as the conclusion, fails to provide an effective summary of the entire paper and instead functions more as the concluding part of the theoretical analysis.

**Questions:**

In practical applications, we often cannot choose which node serves as the central node and are instead constrained by existing conditions. In such scenarios, will the proposed algorithm still maintain its current advantages?

**Limitations:**

I believe the authors have discussed the limitations of their method.

---

> ### Author Rebuttal · Authors · 2024-08-05
>
> Thank you for your time! Let us address the weaknesses:
>
> >  First, one of the two algorithms mentioned in the abstract is relegated entirely to the appendix, which is inappropriate for the paper’s structure.
>
> We agree. However, all algorithms, theorems, and discussions that are relegated to the appendix do not give, conceptually and significantly, new information and algorithmic insights. We tried to showcase the most important ideas in the main part. In the camera-ready version, having an extra page, we will provide more important details from the appendix.
>
> > Second, the experimental results are also exclusively placed in the appendix, which does not conform to standard layout.
>
> The main goal of this paper is to provide fundamental and theoretical limits of the decentralized setup. We tried to focus on theoretical questions in the main part. For instance, we believe adding details about our lower bounds is more important than experiments. Having said that, we will add experiment highlights on the extra page. Moreover, we prepared extra experiments with logistic regression and a neural network that the reviewer can find in the global rebuttal response, which we will add to the camera-ready version.
>
> > Finally, Section 6(Example: line or circle), which serves as the conclusion, fails to provide an effective summary of the entire paper and instead functions more as the concluding part of the theoretical analysis.
>
> Following many previous conference papers published at NeurIPS, the role of the conclusion section plays the Contributions section (Section 3). This section gives a comprehensive overview of the paper. Section 6(Example: line or circle) is not a conclusion. It provides an example with clearer formulas and intuition. We can add a conclusion section instead of Section 6, but it will sacrifice an important example illustrating our new time complexities.
>
> > In practical applications, we often cannot choose which node serves as the central node and are instead constrained by existing conditions. In such scenarios, will the proposed algorithm still maintain its current advantages?
>
> We answer this question in Section 5.3. Let us clarify it here. In general, one can choose any pivot worker and spanning trees, and the methods will converge (Theorem 5) with the time complexity (10) from the paper.
> But to get the best possible convergence time, one should use the rule from Corollary 1. Then we can guarantee the time complexity (11) from the paper, which is optimal (up to log factor). Comparing (10) and (11), one can see the difference is only in $\mu_{i \to j}$ and $\tau_{i \to j},$ which have the relation $\tau_{i \to j} \leq \mu_{i \to j}.$ The term $\mu_{i \to j}$ depends on how we choose a pivot worker and spanning trees in the algorithm. See also the discussion in Lines 181-183, 209-214.
>
> In order to find optimal spanning trees and an optimal pivot worker using the rule from Corollary 1, one has to know $\tau_{i \to j}$ and $h_i.$ In practice, one can run a *load testing* program that estimates $\tau_{i \to j}$ and $h_i.$ Then, one can substitute these estimations to the formulas to find optimal spanning trees and an optimal pivot worker.
>
> We believe that all the weaknesses pointed out by the reviewer can be addressed with an additional page in the camera-ready version of the paper. We hope that we have responded to all questions and weaknesses. If you have more questions, then please let us know.

---

> > ### Comment · Reviewer_VUTZ · 2024-08-13
> >
> > Thank you for addressing my concerns. I will adjust my score accordingly.

---

### Official Review · Reviewer_1dR2 · 2024-07-13

**Soundness:** 3
**Presentation:** 2
**Contribution:** 2
**Rating:** 4
**Confidence:** 4

**Summary:**

This paper examines the time complexity lower bounds in decentralized stochastic asynchronous optimization. It introduces two methods, Fragile SGD and Amelie SGD, which achieve near-optimal and optimal convergence, respectively. The paper also provides convergence analysis for various settings.

**Strengths:**

The topic of studying the exact communication costs is interesting since this should be a crucial problem in decentralized learning. This also brings us a new perspective on decentralized learning.

**Weaknesses:**

1. I believe it is crucial to  **provide empirical results** to demonstrate the effectiveness of the proposed method from two key aspects: the efficiency, which underscores the method’s effectiveness, and the final accuracy, which ensures overall performance guarantee. I would consider raising the score if solid empirical results were provided.

2. I think the proposed methods heavily relies on the pivot worker, which represents a different concept of convergence dependency compared to the spectral gap. If I understand correctly, the framework benefits from a tree-like structure; however, this structure necessitates that the pivot worker possesses significantly greater computation and communication capabilities, almost functioning as a server. For instance, as mentioned in Line 141, “A pivot worker aggregates all stochastic gradients.” Therefore, I think the proposed method acts as a compromise between the centralized “server-client” framework and a decentralized one. Could you please discuss more on the relationship between the pivot worker and the server?

**Questions:**

1. Could you provide empirical results for the proposed methods?

2. Could you provide more discussion about the relationship between the pivot worker and the server?

**Limitations:**

Most limitations are listed in the weaknesses section.

---

> ### Author Rebuttal · Authors · 2024-08-05
>
> Thank you for your time and review. In the following responses, we address the raised problems. **In particular, we prepared extra experiments to support our theoretical results, which the reviewer can find in the global rebuttal's PDF. We also have experiments in Section I of the paper.** Let us respond to the weaknesses and questions:
>
> > I believe it is crucial to provide empirical results to demonstrate the effectiveness of the proposed method from two key aspects:..
>
> > Could you provide empirical results for the proposed methods?
>
> In Section I of our paper, we have already conducted experiments with quadratic optimization problems and showed that our theoretical results align with numerical computations. Nevertheless, we have run extra experiments to support our results in the submission. Please consider our results in the pdf we submitted in the global rebuttal. There, we consider experiments with logistic regression and a neural network and show that our new method has the best convergence rate and accuracy in different settings. We will add highlights of the experiments to the extra page of the camera-ready version of the paper.
>
> We believe that we have addressed all concerns raised by the reviewer by conducting extra experiments, and hope that the reviewer will reconsider the score.
>
> > I think the proposed methods heavily relies on the pivot worker, which represents a different concept of convergence dependency compared to the spectral gap. ...
>
> > Could you provide more discussion about the relationship between the pivot worker and the server?
>
> We agree that our Fragile SGD and Amelie SGD, in some sense, mimic a centralized algorithm. But note that the goal of the paper is to find fundamental time limits of the decentralized optimization in the homogeneous and heterogeneous setup. The standard way to accomplish this goal is to prove a lower bound (Theorem 7) and find a method that attains this lower bound (Fragile SGD, Corollary 1). We have the freedom to design an optimal method in any way as long as this method satisfies the setup's constraints. We designed Fragile SGD and Amelies SGD, valid methods in the decentralized setup. One can design a method without pivot workers and spanning trees, but any other method will never get a time complexity better than our new methods due to the lower bounds.
>
> Let us give a comment on the statement "... the pivot worker possesses significantly greater computation and communication capabilities ..." This is not always true. For instance, consider the example from Section 6. In this example, almost all workers have the same computation and communication capabilities, but we chose the middle worker due to its relative position to other workers. At the same time, all workers are equally loaded with communications and computations. They all calculate stochastic gradients, send and receive the same amount of vectors per second (except the the first and the last worker since they have only one edge). The pivot worker runs an extra process, Process 0, that only aggregates vectors in $g^k.$ However, the aggregation is a negligible operation compared to stochastic gradient computations.

---

> > ### Comment · Reviewer_1dR2 · 2024-08-13
> >
> > Thanks to the authors for the rebuttal. The authors have mostly addressed my concerns in this paper, particularly regarding the experimental verification. I have raised my score. However, I still have a question about the pivot worker. If we consider that the pivot worker runs an extra aggregation process, this process seems very similar to the traditional aggregation process used in centralized methods. For instance, if there are six workers in total, the pivot worker needs to broadcast to the other five workers and receive $g^k$ from them, which significantly increases its communication overhead compared to the other workers. Additionally, the pivot worker runs two parallel processes, and if either of these processes experiences a delay or other system issue, the entire framework could be delayed. Therefore, I believe the proposed framework implicitly assumes that the pivot worker has significantly greater computational and communication capabilities.

---

> > > ### Author Response · Authors · 2024-08-13
> > > **Official Comment by Authors**
> > >
> > > Thank you for raising the score and for the respond.
> > >
> > > Let us clarify the role of the pivot worker.
> > >
> > > i)
> > >
> > > > For instance, if there are six workers in total, the pivot worker needs to broadcast to the other five workers and receive $g^k$ from them, which significantly increases its communication overhead compared to the other workers.
> > >
> > > It largely depends on the structure of a multigraph. If we consider the Line graph (Section 6) or the ND-Mesh (Section A.1), then it is *not* true that the pivot worker has to send vectors to all other workers. Consider Figure 3.a with the 2D-Mesh graph. In this picture, worker $13$ is a pivot worker. During the broadcasting operation, worker 13 sends a vector to workers $8, 12, 18, 14,$ then worker $8$ sends this vector to workers $7, 3, 9,$ worker $12$ sends to workers $17, 11, 7,$ and so on. It's a sequential process where everyone has a comparable load (the pivot worker sends to four workers, while other workers send to three workers). The same reasoning applies when the workers send stochastic gradients to the pivot worker; all workers send almost the same amount of information per second in the Line and the ND-Mesh graphs.
> > >
> > > ii)
> > > > Additionally, the pivot worker runs two parallel processes, and if either of these processes experiences a delay or other system issue, the entire framework could be delayed.
> > >
> > > Processes $0$ and $j^*$ from Algorithms 3 and 4 are *mathematical abstractions*. In a real implementation, it is possible to merge Process 0 and Process $j^*$ in a way that we move the logic from Process 0 to "function ReceiveVectorsFromPreviousWorkers" of Process $j^*.$ This approach allows us to eliminate the additional process in the pivot worker. In the paper, we decided to split the logic into two processes to enhance readability.
> > >
> > > Moreover, note that in real systems, when computers run gradient computations (e.g., with CUDA) and communicate vectors (e.g., via TCP or another standard), the number of running processes can be dozens. So, adding one more running process/thread is a negligible price (which we can avoid using the approach that we described in the previous paragraph).
> > >
> > > In total, we only require the pivot worker to run one extra sum operation of two vectors, which is negligible compared to stochastic gradient computations, where workers should run matrix multiplications, activations (sigmoid, ReLU, ....), convolutions, and other operations, which are clearly more complicated.
> > >
> > > iii)
> > >
> > > Consider the Star graph from Section A.2, where $n$ workers are only connected to one worker in the center (in total, $n + 1$ workers). In this case, the central worker can be a pivot worker. Indeed, in this scenario, the pivot/central worker sends and receives more information per second than other workers. But it's not a problem with our method; it's a problem with the setting. Any other reasonable method in this setting will require the pivot/central worker to work more in order to get fast convergence. For instance, if the pivot/central worker does not send and receive information from another worker, this is counterproductive. This other worker either will be idle or its computations will be ignored.
> > >
> > > Thank you for the response. We hope that we've clarified your last question. If you have more, please let us know.

---

> > > > ### Author Response · Authors · 2024-08-13
> > > > **Official Comment by Authors**
> > > >
> > > > We apologize for the additional comment, but we would like to include one more important remark about the paper.
> > > >
> > > > The goal of this paper is to find the *optimal theoretical time complexities*. To do so, we proved the lower bounds (which we believe are overlooked), and we think this is a non-trivial task that required us to develop new proof techniques that we discuss in detail in Sections 4 and D.3. Fragile SGD and Amelie SGD are practical and valid methods that do not violate the restrictions of the decentralized setup, and designed to show that these lower bounds are tight through our theorems. This is the standard approach in optimization (see [1, 2, 3]).
> > > >
> > > > We thank the reviewer for taking part in the discussion.
> > > >
> > > > [1] Nesterov, Y. (2018). Lectures on convex optimization, volume 137. Springer.
> > > >
> > > > [2] Carmon, Y., Duchi, J. C., Hinder, O., and Sidford, A. (2020). Lower bounds for finding stationary points i. Mathematical Programming, 184(1):71–120.
> > > >
> > > > [3] Woodworth, B. E., Wang, J., Smith, A., McMahan, B., and Srebro, N. (2018). Graph oracle models, lower bounds, and gaps for parallel stochastic optimization. Advances in Neural Information Processing Systems, 31.

---

### Author Rebuttal · Authors · 2024-08-05

We thank the reviewers and the AC for their time and effort. **In the attached PDF**, we added extra experiments with logistic regression and a neural network to support our theoretical claims. These experiments, together with the experiments from Section I, provide solid evidence that our method is efficient and generalizes well (has good accuracy on test splits of datasets).

We want to emphasize that this is the first work to provide *optimal theoretical time complexities* for the challenging decentralized setting with heterogeneous computation and communication times. It was not a trivial task because it required us to develop new proof techniques to prove the lower bounds (see the descriptions in Section 4). We discovered many phenomena, such as *not all workers* should work to achieve optimality in the homogeneous setting (see Section 5.2). Moreover, we derive time complexities with non-trivial dependencies for Line, Torus, and Star graphs, which were unknown in the literature. We believe that these theoretical results, which include lower bounds and optimal methods, are of significant importance to the community working on decentralized methods.

Thank you again! If you have more questions, we will be happy to provide more details.

---

### Decision · Program_Chairs · 2024-09-25

**Decision:**

Accept (poster)

**Comment:**

This paper presents an innovative application of spanning trees in decentralized stochastic asynchronous optimization, contributing significantly to the theoretical foundations of the field. The proposed algorithm improves robustness by mitigating the influence of the worst-performing nodes and provides valuable insights into both computational time complexity and practical applications. The paper is well-written, offering detailed explanations, rigorous proofs, and comprehensive supplementary materials. One reviewer expressed a concern regarding the role of the pivot worker, which the authors satisfactorily addressed during the rebuttal phase. Therefore, we recommend accepting this paper.